# A Theoretical Study of The Effects of Adversarial Attacks on Sparse Regression

**Deepak Maurya**                                                                  *dmaurya@purdue.edu*
*Department of Computer Science, Purdue University*

**Jean Honorio**                                                                  *jhonorio@unimelb.edu.au*
*School of Computing and Information Systems, The University of Melbourne, and*
*ARC Training Centre in Optimisation Technologies, Integrated Methodologies, and Applications (OPTIMA)*

**Reviewed on OpenReview:** *https://openreview.net/forum?id=DaDBtnWcy9*

## Abstract

This paper analyzes $\ell_1$ regularized linear regression under the challenging scenario of having only adversarially corrupted data for training. Firstly, we prove existing deterministic adversarial attacks (e.g., FGSM and variants) focusing on maximizing the loss function can be easily handled with a few samples for support recovery. Hence, we consider a more general, challenging stochastic adversary which can be conditionally dependent on uncorrupted data and show existing models attacking support (Goodfellow et al., 2015; Madry et al., 2018) or Huber model (Prasad et al., 2020) are particular cases of our adversarial model. This enables us to show the counter-intuitive result that an adversary can influence sample complexity by corrupting the "irrelevant features", i.e., non-support. Secondly, as any adversarially robust algorithm has limitations, our theoretical analysis identifies that the dependence (covariance) between adversarial perturbations and uncorrupted data plays a critical role in defining the regimes under which this challenging adversary or Lasso can dominate over each other. Thirdly, we derive a necessary condition for support recovery for any algorithm (not restrictive to Lasso), which corroborates our theoretical findings for Lasso. Fourthly, we identify the fundamental limits and address critical scientific questions of which parameters (i.e., mutual incoherence, the maximum and minimum eigenvalue of the covariance matrix, and the budget of adversarial perturbations) play a role in the high or low probability of success of the Lasso algorithm. Also, the derived sample complexity is logarithmic with respect to the size of the regression parameter vector. Our theoretical claims are validated by empirical analysis.

## 1 Introduction

A well-known instance of the failure of machine learning (ML) models is when they are confronted with adversarial attacks. The vulnerability of ML models to possibly small perturbations imperceptible to the human eye such as one-pixel attacks (Su et al., 2019) may produce inaccurate predictions with high confidence (Szegedy et al., 2014; Goodfellow et al., 2015; Madry et al., 2018). Hence, demystifying empirical failure with theoretical analysis to design certifiable learning algorithms has been an active area of research recently. In this work, we consider the support recovery problem under adversarial attacks and theoretically analyze the behavior of the widely accepted Lasso algorithm.

We begin by providing theoretical evidence of the limitations of existing adversarial attack models in the context of the support recovery problem. We propose a more comprehensive and demanding adversarial model that addresses these limitations. Existing gradient descent-based methods, such as FGSM (Goodfellow et al., 2015) and various others (Madry et al., 2018; Szegedy et al., 2014; Xing et al., 2021; Yin et al., 2019; Awasthi et al., 2020; Qin et al., 2021) target the support (the entries corresponding to non-zero coefficients)

on a per-sample basis. These methods primarily focus on maximizing the loss function rather than making parameter estimation more challenging. Attacks that exclusively target the support can be easily countered by inspecting the mean with just $\Omega(\log(p))$ samples, as demonstrated in Section 2.1 and 2.2. To ensure a non-trivial and intriguing problem, we introduce a more general and challenging adversary.

In this work, we consider a zero-mean sub-Gaussian adversary that can be *conditionally dependent* on the uncorrupted data, aiming to maximize the complexity of the learning process (for Lasso). Our adversary is more challenging as compared to existing attacks (Goodfellow et al., 2015; Szegedy et al., 2014; Xing et al., 2021; Yin et al., 2019; Awasthi et al., 2020; Qin et al., 2021), as it demands a larger number of samples for support recovery. Specifically, as proved in Theorem 10, our adversary demands $\Omega\left(k^2 \log(p)\right)$ samples, whereas existing attacks only need $\Omega(\log(p))$ samples (in Section 2.2). Here, $k$ denotes the number of elements in the support. Furthermore, it is worth noting that under a specific condition, our adversary can be so powerful that achieving successful support recovery becomes impossible, even with infinite samples.

The attack in existing methods is just one possible attack model that fits our assumptions; hence, our adversarial model is more general. For instance, approaches by Goodfellow et al. (2015); Madry et al. (2018); Szegedy et al. (2014); Xing et al. (2021); Yin et al. (2019); Schmidt et al. (2018); Awasthi et al. (2020); Qin et al. (2021) consider deterministic attacks within an $\epsilon$-ball, which is sub-Gaussian since any bounded random variable is known to be sub-Gaussian. Also, note that stochastic attacks are more general as any deterministic value is a random variable with a Dirac delta probability density function. Furthermore, it is worth noting that the $\alpha$-Huber model (Prasad et al., 2020; Diakonikolas et al., 2019) corrupting only $\alpha < 1$ fraction of samples is a particular case of our adversarial model.

In the presence of a more challenging adversary, our analysis (as demonstrated in Lemma 12) reveals an intriguing aspect: the adversary can impact the sample complexity by manipulating not only the "relevant features" or support (i.e., entries corresponding to non-zero coefficients of the regression parameter vector) but also the "irrelevant features" or non-support (i.e., entries corresponding to zero coefficients). This is a counter-intuitive result as the existing methods (Madry et al., 2018; Szegedy et al., 2014; Xing et al., 2021; Yin et al., 2019; Awasthi et al., 2020; Qin et al., 2021) attack only the support (Lemma 6). We also address critical scientific questions of which parameters (i.e., mutual incoherence (after Eq (8)), the maximum and minimum eigenvalue (Lemma 19) of the covariance matrix, and the budget of adversarial perturbations (Eq (15))) play a role in the high or low probability of success of Lasso. The adversary can also increase the sample complexity by designing the covariance matrix of perturbations in the support such that its eigenvector corresponding to the maximum eigenvalue is parallel to the regression parameter vector (discussed after Eq (15)).

We derive a necessary condition for support recovery that applies to any algorithm, extending beyond the scope of Lasso. Notably, we observe that this condition aligns with our separate analysis of the Lasso algorithm, strengthening our findings in that context. With slight abuse of terminology, both conditions essentially indicate that support recovery is impossible when the absolute value of the coefficient in the parameter vector (for an entry in the support) is not large enough as compared to the entries in the covariance matrix between the adversarial perturbation and the uncorrupted data . This aspect of our main result in Theorem 10 is discussed in Section 3.2 and verified empirically. Please refer to Figure 1 for a demonstration of possible and impossible support recovery scenarios in the cases of $b = 0$ and $b \neq 0$ respectively. For more detailed experiments on the three cases of the regularization parameter, $\lambda$ in Theorem 10, characterizing possible and impossible support recovery, please refer to Appendix E.4.

Some initial attempts have been made in the sparse regression literature (Herman & Strohmer, 2010), but we believe that the particular theoretical framework did not allow prior work to make important findings such as noting that when adversarial perturbations correlate with uncorrupted data, then support recovery is impossible. In addition, there is prior literature in robust optimization (Bertsimas & Den Hertog, 2020), where there is uncertainty in the input coefficients of an optimization problem. Their goal is to provide guarantees that the unperturbed optimal solution is still feasible (although not optimal) in the problem with perturbed inputs. Bertsimas & Den Hertog (2020) treats canonical problems such as linear programming, for instance. While we also focus on an optimization problem, Bertsimas & Den Hertog (2020) provides a

general treatment and lacks the focus on machine learning aspects, e.g., support recovery and adversarial perturbations.

**Our Contributions.** Our key contributions are summarized below:

- **A challenging adversary:** We show the limitations of existing adversarial models and use a more general and challenging adversary (in Section 2.1 and 2.2). In this regard, our work offers theoretical insights into the reasons behind the failure of existing adversarial training approaches, even for seemingly simple problems such as sparse regression.

- **Necessary condition:** We derive the necessary condition (in Section 2.4) for the support recovery problem, which is applicable to any method and not limited to Lasso. Further, our independent analysis for Lasso derives similar requirements (in Section 3).

- **Identifying fundamental limitations:** While adversarially robust algorithms have their limitations, our theoretical analysis (in Section 3.2) reveals that the dependence (covariance) between adversarial perturbations and uncorrupted data relative to the magnitude of coefficients in the parameter vector plays a crucial role in determining the regimes in which either the challenging adversary or the Lasso algorithm can outperform each other.

- **Sample complexity:** We derive a sample complexity of order $\Omega\left(k^2 \log(p)\right)$ (Theorem 10) for support recovery under adversarial attacks, which is known to be $\Omega(k \log(p))$ for the non-adversarial regime (Wainwright, 2009). If we assume the adversarial perturbation to be Gaussian, the sample complexity improves to $\Omega\left(k \log(p)\right)$ (Appendix D.10). This improved complexity is proven to be minimax optimal, indicating that further improvement is not possible.

- **Empirical validation:** We verify our theoretical findings via experiments on synthetic and real-world data where the adversarial perturbation can be dependent on the uncorrupted data. This identifies the regimes when Lasso (Fig. 1(a)) or the adversary (Fig. 1(b)) can dominate over each other (Appendix E.4).

- **Theoretical tools:** Our contribution can be seen as a first step towards the study of learning from adversarial training data. As a byproduct, we also obtain several technical results related to a new concentration inequality (Theorem 14), necessary condition (Lemma 9) which could be useful for other problems.

While we focus on sub-Gaussian data, our proofs can be extended to heavy-tailed data. Before proceeding further, we introduce the notations used in this paper.

**Notation.** We use a lowercase letter, e.g., $a$ to denote a scalar, a lowercase bold letter such as $\mathbf{a}$ to denote a vector, and an uppercase bold letter such as $\mathbf{A}$ to denote a matrix. A vector $\mathbf{1}_m$ or $\mathbf{0}_m$ represents a vector of ones or zeros respectively, of size $m$. We denote a set with a calligraphic letter, e.g., $\mathcal{P}$. Also, $[n]$ denotes the set $\{1, 2, \ldots, n\}$. For a vector, $\mathbf{a}_i$ denotes the $i^{\text{th}}$ entry of the vector $\mathbf{a}$. For a matrix $\mathbf{A} \in \mathbb{R}^{p \times q}$, we represent the sub-matrix with rows $\mathcal{P} \subseteq [p]$ and columns $\mathcal{Q} \subseteq [q]$ as $\mathbf{A}_{\mathcal{P}\mathcal{Q}} \in \mathbb{R}^{|\mathcal{P}| \times |\mathcal{Q}|}$.

For a vector vector $\mathbf{a} \in \mathbb{R}^m$, we denote the $\ell_p$ norm as $||\mathbf{a}||_p = \left(\sum_{i=1}^m |\mathbf{a}_i|^p\right)^{\frac{1}{p}}$. Similarly $||\mathbf{A}||_\infty$ denotes the entrywise $\ell_\infty$ norm of a matrix $\mathbf{A}$ and $||\mathbf{A}||_2$ denotes the spectral norm. We denote the induced $\ell_\infty$ norm using $|||\mathbf{A}|||_\infty = \max_{i \in [m]} ||\mathbf{a}_i||_1$ for a matrix $\mathbf{A} \in \mathbb{R}^{m \times k}$, where $\mathbf{a}_i$ denotes its $i^{\text{th}}$ row. Minimum and maximum eigenvalues of a matrix $\mathbf{A}$ are denoted by $\lambda_{\min}(\mathbf{A})$, $\lambda_{\max}(\mathbf{A})$. A function $f(m) = \Omega(g(m))$ implies that there exists a constant $c_1$ such that $f(m) \geq c_1 g(m), \forall m \geq m_0$. Similarly, $f(m) = \mathcal{O}(g(m))$ denotes that there exists a constant $c_2$ such that $f(m) \leq c_2 g(m), \forall m \geq m_0$. For a vector $\mathbf{w} \in \mathbb{R}^p$, $\mathcal{S}(\mathbf{w}) = \{i \in [p], \mathbf{w}_i \neq 0\}$ denotes the support of $\mathbf{w}$ and similarly $\mathcal{S}^c(\mathbf{w}) = [p] \setminus \mathcal{S}(\mathbf{w})$ denotes the non-support.

## 2 A Novel Generative Model For Adversarial Training Data

In this section, we propose a novel generative model for adversarial training. Let $\left(\mathbf{x}^{\star(j)}, y^{\star(j)}\right)$ denote the $j^{th}$ *noise-free* sample. In our model, we assume the adversary has attacked the system and disturbed the noise-free features $\mathbf{x}^{\star(j)}$ as shown below:

$$y^{\star(j)} = \mathbf{w}^{\star\intercal}\mathbf{x}^{\star(j)}, \qquad y^{(j)} = y^{\star(j)} + e^{(j)}, \qquad \mathbf{x}^{(j)} = \mathbf{x}^{\star(j)} + \mathbf{\Delta}(\mathbf{x}^{\star(j)}) \tag{1}$$

where $y^{\star(j)} \in \mathbb{R}$, $\mathbf{x}^{\star(j)} \in \mathbb{R}^p$, $\mathbf{w}^\star \in \mathbb{R}^p$, and $\boldsymbol{\Delta}(\mathbf{x}^{\star(j)})$ is the adversarial perturbation which can be any sub-Gaussian random variable *possibly dependent* on $\mathbf{x}^{\star(j)}$ (See Appendix E.3 for an example). Let $\mathcal{S} = \mathcal{S}(\mathbf{w}^\star)$ denote the support of $\mathbf{w}^\star$, i.e., the set of indices corresponding to non-zero entries of $\mathbf{w}^\star$. Let $k = |\mathcal{S}|$ denote the cardinality of the support, and hence $p - k = |\mathcal{S}^c|$. We argue that our proposed model is more general than other existing models in the literature, including the Huber model in Lemma 3. Our theoretical and empirical analysis (Appendix E.4) draws interesting inferences using this model. We assume only adversarially corrupted data, that is, $n$ independent samples of $\{y^{(j)}, \mathbf{x}^{(j)}\}$ for $j \in [n]$ are available for training, which makes the problem challenging as compared to existing works which assume availability of uncorrupted features $\mathbf{x}^{\star(j)}$. Let $\mathbf{X} \in \mathbb{R}^{n \times p}$ be the collection of $n$ samples $\mathbf{x}^{(j)}$ for $j \in [n]$. For brevity, we may drop the superscript $(j)$ later. Let the population covariance matrix of $\mathbf{X}^\star$ be denoted by $\boldsymbol{\Sigma}^{\mathbf{x}^\star}$. We assume that $\mathbf{x}_i^\star / \sqrt{\boldsymbol{\Sigma}_{ii}^{\mathbf{x}^\star}}$ is a zero-mean sub-Gaussian random variable with variance proxy parameter $\sigma^2$ for $i \in [p]$. We assume that $e$ is a zero-mean sub-Gaussian random variable with variance proxy $\sigma_e^2$.

Adversarial perturbations are typically bounded by some budget for each sample (Zhai et al., 2019; Balda et al., 2019; Yin et al., 2019; Cohen et al., 2019) so that the underlying model is learnable. We model this fixed budget as a parameter in our analysis, and hence, we provide performance guarantees as a function of this parameter. We choose adversarial perturbations to be sub-Gaussian random variables, which generalizes the regime of bounded perturbations. Adversarial perturbations can also be dependent on the uncorrupted regressors, $\mathbf{x}^\star$ for each sample. To clarify, $\boldsymbol{\Delta}(\mathbf{x}^{\star(i)})$ can be dependent on $\mathbf{x}^{\star(i)}$, where $i$ denotes the $i^{\text{th}}$ sample, but $\boldsymbol{\Delta}(\mathbf{x}^{\star(i)})$ is independent of $\mathbf{x}^{\star(j)}$, if $i \neq j$, where $j$ denotes another sample. Note that the adversary does not have control over the uncorrupted regressors $\mathbf{x}^\star$, but has access to $\mathbf{x}^\star$, which can be used to design $\boldsymbol{\Delta}(\mathbf{x}^\star)$ in an arbitrary manner (that may make the support recovery problem more challenging).

**Assumption 1.** *(Adversarial Perturbation)* $\boldsymbol{\Delta}(\mathbf{x}^\star)$ *is a zero-mean sub-Gaussian random vector with parameters* $(\boldsymbol{\Sigma}^{\boldsymbol{\Delta}}, r^2)$, *which can be* conditionally dependent *on uncorrupted data. Here* $\boldsymbol{\Sigma}^{\boldsymbol{\Delta}}$ *is the population covariance matrix of the adversarial perturbation* $(\boldsymbol{\Delta}(\mathbf{x}^\star))$ *and* $r^2$ *is the variance proxy parameter for some* $r \in \mathbb{R}$.

In the above definition, we require that $\mathbb{E}\left(\exp\left(\boldsymbol{\alpha}^\intercal \boldsymbol{\Delta}(\mathbf{x}^\star)\right)\right) \leq \exp\left(\boldsymbol{\alpha}^\intercal \boldsymbol{\Sigma}^{\boldsymbol{\Delta}} \boldsymbol{\alpha} r^2 / 2\right)$, for all $\boldsymbol{\alpha} \in \mathbb{R}^p$ which is akin to the classical definition (Hsu et al., 2012) as discussed briefly in Appendix A.1.

## 2.1 Existing attacks are special cases of our adversarial model

Note that a random variable is more general than a deterministic one, i.e., a deterministic quantity is a random variable with a Dirac delta probability density function. Hence, existing deterministic adversarial attacks (Goodfellow et al., 2015; Madry et al., 2018; Szegedy et al., 2014; Xing et al., 2021; Yin et al., 2019; Awasthi et al., 2020; Qin et al., 2021) are a special case of our adversarial model specified in Assumption 1.

**Lemma 2.** *Existing adversarial attacks (Goodfellow et al., 2015; Madry et al., 2018; Szegedy et al., 2014; Xing et al., 2021; Yin et al., 2019; Awasthi et al., 2020; Qin et al., 2021) are a special case of our adversarial model specified in Assumption 1.*

The proof of the above lemma uses the fact that the aforementioned attacks use a vector $\boldsymbol{\Delta}(\mathbf{x}^\star)$ with bounded norm, and then shows that any vector with a bounded norm is a sub-Gaussian random vector.

**Lemma 3.** *The $\alpha$-Huber-model (Prasad et al., 2020; Diakonikolas et al., 2019) corrupting only $\alpha < 1$ fraction of samples is a special case of our adversarial model specified in Assumption 1.*

The proof of the above lemma designs an adversarial attack $\boldsymbol{\Delta}(\mathbf{x}^\star)$ that returns $\mathbf{0}$ with probability $1 - \alpha$ and a carefully defined perturbation that depends on $\mathbf{x}^\star$ with probability $\alpha$.

**Remark 4.** *The class of sub-Gaussian variates includes for instance some unbounded random variables (e.g., Gaussian variables), any bounded random variable (e.g. Bernoulli, multinomial, uniform), any random variable with strictly log-concave density, and any finite mixture of sub-Gaussian variables. Hence, our adversarial model is more general.*

## 2.2 Our adversarial attack is more challenging than existing attacks

It may look obvious for an adversary to spend its fixed budget for a sample to disturb the support entries $(\mathbf{x}_{\mathcal{S}})$ only, since $y^{\star(j)} = \mathbf{w}^{\star\intercal}\mathbf{x}^{\star(j)} = \mathbf{w}_{\mathcal{S}}^{\star\intercal}\mathbf{x}_{\mathcal{S}}^{\star(j)}, \forall j$. We assume the adversary attacks the non-support entries $(\mathbf{x}_{\mathcal{S}^c})$ as well, to make the learning task of estimating $\mathcal{S}$ and $\mathcal{S}^c$ tougher, which is formalized in the following lemma.

**Lemma 5.** *If the adversary attacks only the support entries ($\mathbf{x}_{\mathcal{S}}$) or the non-support entries ($\mathbf{x}_{\mathcal{S}^c}$) with non-zero-mean adversarial perturbation, then the learner can guess the support trivially with probability at least $1 - \mathcal{O}\left(1/p\right)$ if $n = \Omega\left(\log(p)\right)$.*

The proof of the above lemma relies on the fact that the sample mean of feature vector $\mathbf{x}^{(j)}$ for $j \in [n]$ will be close to the population mean. If only the support is attacked, the sample mean is away from zero for entries in the support, and is zero for entries in the non-support. Hence, the learner can guess the support by just computing the sample mean.

**Lemma 6.** *The support can be trivially estimated in $n = \Omega\left(\log(p)\right)$ samples with high probability under existing adversarial attacks (Goodfellow et al., 2015; Madry et al., 2018; Szegedy et al., 2014; Xing et al., 2021; Yin et al., 2019; Awasthi et al., 2020; Qin et al., 2021) aiming to maximize a per-sample loss function.*

The proof of the above lemma first shows that the aforementioned methods attack only the support, and then uses Lemma 5 to make the final claim.

In order to avoid such a trivial case, we consider the zero-mean in Assumption 1.

## 2.3 Other assumptions for support recovery

In this section, we define a few quantities used in our analysis later. Let the sample covariance matrix of $\mathbf{X}$ be denoted by $\hat{\boldsymbol{\Sigma}}^{\mathbf{x}} = \frac{1}{n}\mathbf{X}^{\intercal}\mathbf{X}$ and the population covariance matrix be denoted by $\boldsymbol{\Sigma}^{\mathbf{x}}$. For the uniqueness of the solution to the problem stated in Eq. (2), we need a submatrix of the sample covariance matrix to be positive definite. But as $\mathbf{X}$ is assumed to be random, we assume the population covariance matrix of $\mathbf{X}^{\star}$ to be positive definite as done in prior literature (Wainwright, 2009; Ravikumar et al., 2011; 2010; 2009; Daneshmand et al., 2014).

**Assumption 7.** *(Positive Definiteness) The minimum eigenvalue of the population covariance matrix of $\mathbf{x}^{\star}$ fulfills $\lambda_{\min}\left(\boldsymbol{\Sigma}_{\mathcal{S}\mathcal{S}}^{\mathbf{x}^{\star}}\right) = C_{\min} > 0$.*

Let the population covariance matrix between adversarial perturbations $\boldsymbol{\Delta}(\mathbf{x}^{\star})$ and measurements $\mathbf{x}^{\star}$ be denoted by $\boldsymbol{\Sigma}^{\boldsymbol{\Delta}\mathbf{x}^{\star}}$. Similarly, the minimum eigenvalue of the population covariance matrix of adversarial perturbations in the support $\mathcal{S}$, denoted by $\boldsymbol{\Sigma}_{\mathcal{S}\mathcal{S}}^{\boldsymbol{\Delta}}$, is $D_{\min} \geq 0$. The minimum eigenvalue of $[\boldsymbol{\Sigma}^{\boldsymbol{\Delta}\mathbf{x}^{\star}} + \boldsymbol{\Sigma}^{\mathbf{x}^{\star}\boldsymbol{\Delta}}]_{\mathcal{S}\mathcal{S}}/2$ is denoted by $F_{\min}$, which influences the sample complexity as discussed in Lemma 13. Similarly, the maximum eigenvalues of matrices $\{\boldsymbol{\Sigma}^{\mathbf{x}}, \boldsymbol{\Sigma}^{\boldsymbol{\Delta}}, \boldsymbol{\Sigma}^{\boldsymbol{\Delta}\mathbf{x}^{\star}}\}$ are denoted by $\{C_{\max}, D_{\max}, F_{\max}\}$, whose influence on sample complexity can be seen in Lemma 15. Further, we make the assumption of mutual incoherence, which implies that the regressors in the non-support $\mathcal{S}^c$ do not have a strong correlation with regressors in the support $\mathcal{S}$.

**Assumption 8.** *(Mutual Incoherence) For some $\gamma \in (0, 1]$, $\left|\left|\left|\boldsymbol{\Sigma}_{\mathcal{S}^c\mathcal{S}}^{\mathbf{x}}\left(\boldsymbol{\Sigma}_{\mathcal{S}\mathcal{S}}^{\mathbf{x}}\right)^{-1}\right|\right|\right|_{\infty} \leq 1 - \gamma$.*

This assumption is not restrictive (see Lemma 18) and has been used in various works related to support recovery (Wainwright, 2009; Ravikumar et al., 2011; 2010; 2009; Daneshmand et al., 2014). In addition, we also assume $\left|\left|\left|\left(\boldsymbol{\Sigma}_{\mathcal{S}\mathcal{S}}^{\mathbf{x}}\right)^{-1}\right|\right|\right|_{\infty} = G_{\max} \leq \infty$ and $|||\boldsymbol{\Sigma}_{\mathcal{S}^c\mathcal{S}}^{\mathbf{x}}|||_{\infty} = H_{\max} \leq \infty$. We refer to $(r, \gamma, F_{\min}, F_{\max}, D_{\min}, D_{\max}, G_{\max}, H_{\max})$ as the budget given to the adversary for corrupting each sample.

## 2.4 Necessary condition for support recovery

We derive the following necessary condition for support recovery relating the adversarial budget to the model parameters. Note that the following requirement is for all methods and is not limited to Lasso.

**Lemma 9.** *The condition $\frac{16b}{15\gamma}\left(1+\frac{\gamma}{4}\right)\frac{3G_{\max}}{2} \leq \min_{i\in\mathcal{S}}|\mathbf{w}_i^\star|$, where $b = \left\|\mathbf{\Sigma}_{[p]\mathcal{S}}^{\mathbf{x}^\star\mathbf{\Delta}}\,\mathbf{w}_{\mathcal{S}}^\star\right\|_\infty$ is necessary for support recovery, even in the population regime, where (intuitively speaking) the learner has access to an infinite number of samples.*

In the above lemma, we prove the necessary condition by considering a case where support recovery is impossible if the required condition is disobeyed. The proof carefully constructs a specific case where the uncorrupted data as well as the adversarial perturbations are both Gaussian, and then uses the fact that Gaussian distributions are fully identifiable from its first and second order moments to argue for impossibility.

With slight abuse of terminology, the above lemma implies that the coefficients in the parameter vector must be sufficiently large compared to the adversarial budget for successful detection. With a brief discussion of our assumptions in this section, we present our main theoretical result in the next section.

## 3 Our Main Result

In this section, we present and discuss our main theorem that implies a sample complexity of $n = \Omega\left(k^2\log(p)\right)$ for correct support recovery with high probability (except when there is correlation between the adversarial perturbations and uncorrupted data). We provide formal and intuitive implications of the proposed theorem first and further present its proof.

The Lasso problem under the adversarial setting can be stated as:

$$\hat{\mathbf{w}} = \arg\min_{\mathbf{w}\in\mathbb{R}^p} l(\mathbf{w}) + \lambda||\mathbf{w}||_1 = \arg\min_{\mathbf{w}\in\mathbb{R}^p}\frac{1}{2n}||\mathbf{y}-\mathbf{X}\mathbf{w}||_2^2 + \lambda||\mathbf{w}||_1 \tag{2}$$

Our main theoretical result for the above problem is as follows.

**Theorem 10.** *If $n = \Omega(k^2\log(p))$, Assumption 1, 7, and 8 hold, and*

$$\lambda \geq \max\left\{\frac{16b}{\gamma}, \frac{q_1\sigma_e}{\gamma}\sqrt{\frac{2\log(p)}{n}}, \frac{16q}{\gamma}\sqrt{\frac{4\log(p)}{n}}\right\}$$

*where $q = r\sqrt{\mathbf{w}_{\mathcal{S}}^{\star\mathsf{T}}\mathbf{\Sigma}_{\mathcal{S}\mathcal{S}}^{\mathbf{\Delta}}\,\mathbf{w}_{\mathcal{S}}^\star}\max_{i\in[p]}\left(\sigma\sqrt{\mathbf{\Sigma}_{ii}^{\mathbf{x}}}+r\sqrt{\mathbf{\Sigma}_{ii}^{\mathbf{\Delta}}}\right)$, $b = \left\|\mathbf{\Sigma}_{[p]\mathcal{S}}^{\mathbf{x}^\star\mathbf{\Delta}}\,\mathbf{w}_{\mathcal{S}}^\star\right\|_\infty$, $q_1^2 = 3\left(C_{\max}+2F_{\max}+D_{\max}\right)$, then with probability of at least $1-\mathcal{O}\left(\frac{1}{p}\right)$, we have:*

1. *The true support is recovered, i.e., $\mathcal{S}(\hat{\mathbf{w}}) \subseteq \mathcal{S}(\mathbf{w}^\star)$ or equivalently $\hat{\mathbf{w}}_i = 0, \forall i \notin \mathcal{S}(\mathbf{w}^\star)$*

2. *$\hat{\mathbf{w}}_{\mathcal{S}}$ is the unique solution for the Lasso problem stated in Eq. (2).*

3. *The estimated parameter vector satisfies*

$$\|\hat{\mathbf{w}}_{\mathcal{S}}-\mathbf{w}_{\mathcal{S}}^\star\|_\infty \leq \lambda\left(1+\gamma/4\right)3G_{\max}/2 = f(\lambda) \tag{3}$$

4. *$\hat{\mathbf{w}}_i \neq 0$ and furthermore $\text{sign}(\hat{\mathbf{w}}_i) = \text{sign}(\mathbf{w}_i^\star), \forall i \in \mathcal{S}(\mathbf{w}^\star)$ if $\min_{i\in\mathcal{S}}|\mathbf{w}_i^\star| \geq 2f\left(\lambda\right)$.*

5. *Additionally if $\mathbf{\Sigma}_{[p]\mathcal{S}}^{\mathbf{x}^\star\mathbf{\Delta}} = \mathbf{0}_{p\times k}$, and therefore $b = 0$, then $\lambda \geq \mathcal{O}\left(\sqrt{\log(p)/n}\right)$, statement 1, 2, and 4 still hold true and statement 3 is modified to*

$$\|\hat{\mathbf{w}}_{\mathcal{S}}-\mathbf{w}_{\mathcal{S}}^\star\|_\infty \leq f(\lambda) = \mathcal{O}\left(\sqrt{\log(p)/n}\right)$$

The first two claims of the Theorem 10 imply that we can uniquely recover the true support with high probability, assuming we have a sufficient number of samples if we choose a regularization parameter $\lambda$ greater than a certain threshold. Note that choosing a very large value of $\lambda$ is not desirable as it would also increase the upper bound for $||\hat{\mathbf{w}}_{\mathcal{S}}-\mathbf{w}_{\mathcal{S}}^\star||_\infty$, as per the above theorem (Eq. (3)). The fourth claim of the theorem states that the minimum magnitude among the support entries of the regression parameter vector should be greater than a certain function of $\lambda$ for correct sign recovery.

**Intuition.** Note that the adversary can increase the lower bound of $\lambda$. For example, if $b \neq 0$, then $\lambda = {}^{16b}/\gamma$ even for $n \to \infty$, resulting in a higher value of $f(\lambda)$ in Eq. (3). This may lead to the violation of the requirement in statement 4, $\min_{i \in \mathcal{S}} |\mathbf{w}_i^\star| \geq 2f(\lambda)$, if the smallest entry in $\mathbf{w}_{\mathcal{S}}^\star$ is not large enough. Support recovery may not be possible in such a case due to a low probability of non-support recovery $\mathbb{P}[\forall i \notin \mathcal{S}, \hat{w}_i = 0]$ (See Appendix E.4).

We have formally proved a similar requirement in Lemma 9 for all methods not limited to Lasso. But it is worth noting that substituting $\lambda = \frac{16b}{\gamma}$ in Eq. (3) leads to the requirement stated in Lemma 9. The term $\min_{i \in \mathcal{S}} |\mathbf{w}_i^\star|$ can be seen in statement 4 of Theorem 10. Hence, our analysis for Lasso relates to the necessary condition for all methods.

In statement 5 with $b = 0$, the situation mentioned above can be avoided by increasing the value of $n$ appropriately despite any efforts from the adversary. This helps us to identify different regimes under which we can provide theoretical guarantees of Lasso for successful support recovery under adversarial attacks and also the case which may be favorable to the adversary (See Section 3.2).

**Proof Sketch.** We use a constructive proof technique: primal-dual witness (PDW) method (Wainwright, 2009; Ravikumar et al., 2011; 2010; 2009; Daneshmand et al., 2014) to prove Theorem 10. The proof outline is summarized below:

- The PDW framework starts by allowing us to find sufficient conditions to estimate the elements of the non-support ($\mathcal{S}^c$) first by ensuring strict dual feasibility (Section 3.1). This step ensures that we correctly recover the zeroes, i.e., $\hat{\mathbf{w}}_i = 0$ for all $i \notin \mathcal{S}(\mathbf{w}^\star)$. This establishes the first claim of exact support recovery in Theorem 10.
- We then derive the sufficient conditions for uniqueness of $\hat{\mathbf{w}}_{\mathcal{S}}$ (Section C.1), which proves the second claim of Theorem 10.
- The goodness of the estimated parameter vector is proved in Section C.2 by deriving an upper bound on $||\hat{\mathbf{w}}_{\mathcal{S}} - \mathbf{w}_{\mathcal{S}}^\star||_\infty$ to justify the third claim of Theorem 10.
- Armed with the second and third claim, we prove the fourth claim of Theorem 10 for correctly recovering the non-zeros, i.e., $\hat{\mathbf{w}}_i \neq 0$ for all $i \in \mathcal{S}(\mathbf{w}^\star)$.

**Remark 11.** *For clarity of exposition, we focus on sub-Gaussian data. Our proofs can be extended to heavy-tailed data by using m-order moment concentration instead of sub-Gaussian concentration, as done in (Ravikumar et al., 2011) for another machine learning problem. This will lead to a sample complexity polynomial in p, instead of logarithmic in p.*

## 3.1 Exact Support Recovery

In this subsection, we verify the stationarity, complementary slackness, and strict dual feasibility conditions for the optimal solution $\hat{\mathbf{w}}_{\mathcal{S}}$. The stationarity condition is (See Appendix D.1):

$$\nabla l((\hat{\mathbf{w}}_{\mathcal{S}}, \mathbf{0})) + \lambda \hat{\mathbf{z}} = \mathbf{0}_{p \times 1} \tag{4}$$

where $\hat{\mathbf{z}} \in \partial ||\hat{\mathbf{w}}||_1$ belongs to the sub-differential set of the $\ell_1$ norm at $\hat{\mathbf{w}}$. In the context of the primal-dual witness framework (Ravikumar et al., 2010; 2011; 2009; Daneshmand et al., 2014), $\hat{\mathbf{w}}_{\mathcal{S}}$ and $\hat{\mathbf{z}}$ are referred as the primal and dual variables respectively. As $\hat{\mathbf{z}}$ belongs to the sub-differential set of the $\ell_1$ norm, we can claim that $||\hat{\mathbf{z}}||_\infty \leq 1$ by norm duality but for strict dual feasibility we need $||\hat{\mathbf{z}}_{\mathcal{S}^c}||_\infty < 1$ as stated in Lemma 1 of (Wainwright, 2009). In order to ensure this condition, we need to first derive $\hat{\mathbf{z}}_{\mathcal{S}^c}$ from the first order stationary condition in Eq. (4) which is a $p-$dimensional vector equation and can be written for elements in $\mathcal{S}$ and $\mathcal{S}^c$ separately to derive $\hat{\mathbf{z}}_{\mathcal{S}^c}$. The final expression is (See Appendix D.1 for more details):

$$\hat{\mathbf{z}}_{\mathcal{S}^c} = \hat{\mathbf{z}}_{\mathcal{S}^c_{t_1}} + \hat{\mathbf{z}}_{\mathcal{S}^c_{t_2}}, \quad \hat{\mathbf{z}}_{\mathcal{S}^c_{t_1}} = \mathbf{X}_{\mathcal{S}^c}^\intercal \mathbf{X}_{\mathcal{S}} \left( \mathbf{X}_{\mathcal{S}}^\intercal \mathbf{X}_{\mathcal{S}} \right)^{-1} \hat{\mathbf{z}}_{\mathcal{S}} \tag{5}$$

$$\hat{\mathbf{z}}_{\mathcal{S}^c_{t_2}} = \mathbf{X}_{\mathcal{S}^c}^\intercal \left( \mathbf{P}/\lambda n \right) \left( \mathbf{e} - \boldsymbol{\Delta} \left( \mathbf{X}_{\mathcal{S}}^\star \right) \mathbf{w}_{\mathcal{S}}^\star \right), \quad \text{where} \quad \mathbf{P} = \left( \mathbf{I}_n - \mathbf{X}_{\mathcal{S}} \left( \mathbf{X}_{\mathcal{S}}^\intercal \mathbf{X}_{\mathcal{S}} \right)^{-1} \mathbf{X}_{\mathcal{S}}^\intercal \right) \tag{6}$$

where $\mathbf{I}_n$ represents an identity matrix of dimension $n \times n$, $\mathbf{\Delta}(\mathbf{X})$ is $n \times p$ matrix containing $n$ samples of adversarial perturbation. We have decomposed $\hat{\mathbf{z}}_{\mathcal{S}^c}$ in two terms $\hat{\mathbf{z}}_{\mathcal{S}^c{}_{t_1}}$ and $\hat{\mathbf{z}}_{\mathcal{S}^c{}_{t_2}}$ to bound them separately in next two sub-sections.

### 3.1.1 Analyzing adversarial attack on support ($\mathcal{S}$) and non-support ($\mathcal{S}^c$)

Let $\mathbf{R} = \frac{1}{n}\mathbf{X}_{\mathcal{S}^c}^{\intercal}\mathbf{X}_{\mathcal{S}}$ and $\mathbf{Q} = \frac{1}{n}\mathbf{X}_{\mathcal{S}}^{\intercal}\mathbf{X}_{\mathcal{S}}$, and hence $\mathbb{E}[\mathbf{R}] = \mathbf{\Sigma}_{\mathcal{S}^c\mathcal{S}}^{\mathbf{x}}$, $\mathbb{E}[\mathbf{Q}] = \mathbf{\Sigma}_{\mathcal{S}\mathcal{S}}^{\mathbf{x}}$. The simplified expression obtained after simplification for $\hat{\mathbf{z}}_{\mathcal{S}^c{}_{t_1}}$ in Eq. (5) is (See Appendix D.2):

$$\left|\left|\hat{\mathbf{z}}_{\mathcal{S}^c{}_{t_1}}\right|\right|_{\infty} \leq \left|\left|\left|\mathbf{X}_{\mathcal{S}^c}^{\intercal}\mathbf{X}_{\mathcal{S}}\left(\mathbf{X}_{\mathcal{S}}^{\intercal}\mathbf{X}_{\mathcal{S}}\right)^{-1}\right|\right|\right|_{\infty} \leq m_1 + m_2 + m_3 + m_4 \tag{7}$$

where $m_1$, $m_2$, $m_3$, and $m_4$ are defined as:

$$m_1 = \left|\left|\left|(\mathbf{R} - \mathbb{E}[\mathbf{R}])\left(\mathbb{E}[\mathbf{Q}]\right)^{-1}\right|\right|\right|_{\infty}, \quad m_2 = \left|\left|\left|\mathbb{E}[\mathbf{R}]\left(\mathbf{Q}^{-1} - \left(\mathbb{E}[\mathbf{Q}]\right)^{-1}\right)\right|\right|\right|_{\infty}$$

$$m_3 = \left|\left|\left|(\mathbf{R} - \mathbb{E}[\mathbf{R}])\left(\mathbf{Q}^{-1} - \left(\mathbb{E}[\mathbf{Q}]\right)^{-1}\right)\right|\right|\right|_{\infty}, \quad m_4 = \left|\left|\left|\mathbb{E}[\mathbf{R}]\left(\mathbb{E}[\mathbf{Q}]\right)^{-1}\right|\right|\right|_{\infty}$$

This carefully constructed decomposition of $\hat{\mathbf{z}}_{\mathcal{S}^c{}_{t_1}}$ has given us the freedom to study the effect of the adversarial perturbation on the non-support entries and the support entries by analyzing $m_1$ and $m_2$ respectively. The term $m_4$ in Eq. (7) can be bounded using the mutual incoherence Assumption 8. We propose Lemma 12 and Lemma 13 to bound the terms $m_1$, $m_2$, and $m_3$ as discussed below.

The variable $m_1$ in Eq. (7) is a function of the adversarial perturbation in $\mathcal{S}^c$ for a fixed value of $\mathbb{E}[\mathbf{Q}] = \mathbf{\Sigma}_{\mathcal{S}\mathcal{S}}^{\mathbf{x}}$. To bound $m_1$, we use the sub-multiplicative property of norms:

$$m_1 \leq |||\mathbf{X}_{\mathcal{S}^c}^{\intercal}\mathbf{X}_{\mathcal{S}}/n - \mathbf{\Sigma}_{\mathcal{S}^c\mathcal{S}}^{\mathbf{x}}|||_{\infty}\left|\left|\left|(\mathbf{\Sigma}_{\mathcal{S}\mathcal{S}}^{\mathbf{x}})^{-1}\right|\right|\right|_{\infty}$$

**Lemma 12.** *If* $n = \Omega\left(k^2\xi^2\log(p)/\delta^2\right)$ *and* $0 \leq \delta \leq 32\xi k$, *then* $\left|\left|\left|\frac{1}{n}\mathbf{X}_{\mathcal{S}^c}^{\intercal}\mathbf{X}_{\mathcal{S}} - \mathbf{\Sigma}_{\mathcal{S}^c\mathcal{S}}^{\mathbf{x}}\right|\right|\right|_{\infty} \leq \delta$ *with probability at least* $1 - \mathcal{O}(1/p)$, *where* $\xi = \max_{i \in \mathcal{S}}\left(\sigma\sqrt{\mathbf{\Sigma}_{ii}^{\mathbf{x}}} + r\sqrt{\mathbf{\Sigma}_{ii}^{\mathbf{\Delta}}}\right)\max_{j \in \mathcal{S}^c}\left(\sigma\sqrt{\mathbf{\Sigma}_{jj}^{\mathbf{x}}} + r\sqrt{\mathbf{\Sigma}_{jj}^{\mathbf{\Delta}}}\right)$.

The proof of the above lemma relies on properties of norms and sub-Gaussian distributions, union bound, and sub-exponential tail bounds. We can claim the following by substituting $\delta = \gamma/\left|\left|\left|16(\mathbf{\Sigma}_{\mathcal{S}\mathcal{S}}^{\mathbf{x}})^{-1}\right|\right|\right|_{\infty}$ in Lemma 12:

$$m_1 = \left|\left|\left|(\mathbf{R} - \mathbb{E}[\mathbf{R}])\left(\mathbb{E}[\mathbf{Q}]\right)^{-1}\right|\right|\right|_{\infty} \leq \gamma/16 \tag{8}$$

if $n = \Omega\left(k^2\xi^2 G_{\max}^2\log(p)/\gamma^2\right)$.

**Intuition.** It should be noted that the value of $\delta$ chosen to analyze the adversarial perturbation in $\mathcal{S}^c$ is a function of the adversarial perturbation in $\mathcal{S}$ as $\mathbf{\Sigma}_{\mathcal{S}\mathcal{S}}^{\mathbf{x}} = \mathbf{\Sigma}_{\mathcal{S}\mathcal{S}}^{\mathbf{x}^\star} + [\mathbf{\Sigma}^{\mathbf{\Delta}\mathbf{x}^\star} + \mathbf{\Sigma}^{\mathbf{x}^\star\mathbf{\Delta}}]_{\mathcal{S}\mathcal{S}} + \mathbf{\Sigma}_{\mathcal{S}\mathcal{S}}^{\mathbf{\Delta}}$. Hence if the adversarial perturbation in $\mathcal{S}$ is designed such that $G_{\max} = \left|\left|\left|(\mathbf{\Sigma}_{\mathcal{S}\mathcal{S}}^{\mathbf{x}})^{-1}\right|\right|\right|_{\infty}$ increases, then the learner is forced to choose a smaller value of $\delta$ in Lemma 12 for the adversarial perturbation in $\mathcal{S}^c$ which increases the sample complexity. This also demonstrates the counterintuitive point that an adversary can influence the sample complexity by an attack on the non-support (See Lemma 6). The next step is to bound the term $m_2$ in Eq. (7) using the following lemma.

**Lemma 13.** *If* $n = \Omega\left(k^2\log(p)/\delta^2(C_{\min}+2F_{\min}+D_{\min})^4\right)$, *then with probability of at least* $1 - \mathcal{O}(1/p)$ *we have* $\left|\left|\left|\left(\frac{1}{n}\mathbf{X}_{\mathcal{S}}^{\intercal}\mathbf{X}_{\mathcal{S}}\right)^{-1} - (\mathbf{\Sigma}_{\mathcal{S}\mathcal{S}}^{\mathbf{x}})^{-1}\right|\right|\right|_{\infty} \leq \delta$.

The proof of the above lemma relies on properties of norms, sub-Gaussian tail bounds, and also the bound for $\left|\left|\frac{1}{n}\mathbf{X}_{\mathcal{S}}^{\star\intercal}\mathbf{\Delta}(\mathbf{X}_{\mathcal{S}}^{\star}) - \mathbf{\Sigma}_{\mathcal{S}\mathcal{S}}^{\mathbf{\Delta}\mathbf{x}^\star}\right|\right|_2$ derived in the following theorem.

**Theorem 14.** *For* $0 < \delta < \frac{32r\sigma ab_k}{n}$, $\mathbf{\Delta}(\mathbf{X}_{\mathcal{S}}^{\star}), \mathbf{X}_{\mathcal{S}}^{\star} \in \mathbb{R}^{n \times k}$, $a^2 = \max_{j \in \mathcal{S}}\mathbf{\Sigma}_{jj}, b_k^2 = k\sum_{i \in \mathcal{S}}\left(\mathbf{\Sigma}_{ii}^{\mathbf{\Delta}}\right)^2$, *we have:*

$$\mathbb{P}\left[\left|\left|\frac{1}{n}\mathbf{X}_{\mathcal{S}}^{\star\intercal}\mathbf{\Delta}(\mathbf{X}_{\mathcal{S}}^{\star}) - \mathbf{\Sigma}_{\mathcal{S}\mathcal{S}}^{\mathbf{\Delta}\mathbf{x}^\star}\right|\right|_2 \geq \delta\right] \leq 4e^{\frac{-n\delta^2}{256r^2\sigma^2 ab_k}} \tag{9}$$

The proof of this theorem is interesting as $\mathbf{X}_{\mathcal{S}}^{\star\intercal}\boldsymbol{\Delta}\left(\mathbf{X}_{\mathcal{S}}^{\star}\right)$ is a non-symmetric matrix, which makes the problem slightly challenging as compared to symmetric matrices. Hence we use Lemma 20 to transform a non-symmetric matrix to a symmetric matrix without changing its spectral norm. Further, we use Lemma 6.12 from (Wainwright, 2019), which states for a random symmetric matrix $\mathbf{M}$

$$\mathbb{P}\left[||\mathbf{M}||_2 \geq \delta\right] \leq 2\mathrm{tr}\left(\Psi_{\mathbf{M}}(\lambda)\right)e^{-\lambda\delta} \tag{10}$$

where $\delta > 0$, $\Psi_{\mathbf{M}}(\lambda)$ denotes the moment generating function (MGF) and tr represents the trace. Thus, to derive the bound of $||\mathbf{M}||_2$, we bound the trace of the MGF of $\mathbf{M}$. Using the properties of sub-Gaussian and sub-exponential distributions, we derive a matrix $\mathbf{V} \in \mathbb{R}^{2k \times 2k}$ such that

$$\Psi_{\mathbf{M}}(\lambda) \preccurlyeq e^{\frac{\lambda^2 \mathbf{V}}{2}}$$

Hence, in order to bound the trace of $\Psi_{\mathbf{M}}(\lambda)$, we focus on the eigenvalues of $\mathbf{V}$. We observe that $\mathbf{V}$ is a matrix with only two non-zero eigenvalues. This helps us to derive improved bounds as compared to the case of all $2k$ eigenvalues being non-zero. More details can be seen in Appendix D.5.

Returning to deriving bound of $m_2$ in Eq. (7), we substitute $\delta = \gamma/8|||\boldsymbol{\Sigma}_{\mathcal{S}^c\mathcal{S}}^{\mathbf{x}}|||_\infty$ in Lemma 13:

$$m_2 = \left|\left|\left|\mathbb{E}\left[\mathbf{R}\right]\left(\mathbf{Q}^{-1} - \left(\mathbb{E}\left[\mathbf{Q}\right]\right)^{-1}\right)\right|\right|\right|_\infty \leq \gamma/8 \tag{11}$$

if $n = \Omega\left(k^2\log(p)H_{\max}^2/\gamma^2(C_{\min}+2F_{\min}+D_{\min})^4\right)$.

**Intuition.** This analysis provides insight into the nature of the dependence of the sample complexity on the dimensions ($k$ and $p$), and also other parameters such as mutual incoherence ($\gamma$), minimum eigenvalues $\{C_{\min}, D_{\min}, F_{\min}\}$, and other constants like $H_{\max}$.

Further, we proceed to bound the term $m_3$ in Eq. (7) by using the sub-multiplicative property of norms and substituting $\delta = \sqrt{\gamma}/4$ in Lemma 12 and Lemma 13 to claim:

$$m_3 = |||(\mathbf{R} - \mathbb{E}\left[\mathbf{R}\right])|||_\infty \left|\left|\left|\left(\mathbf{Q}^{-1} - \left(\mathbb{E}\left[\mathbf{Q}\right]\right)^{-1}\right)\right|\right|\right|_\infty \leq \gamma/16$$

if $n = \Omega\left(k^2\log(p)\right)$. We substitute Eq. (8), Eq. (11), and the above equation in Eq. (7) to arrive at:

$$\left|\left|\hat{\mathbf{z}}_{\mathcal{S}^c_{t_1}}\right|\right|_\infty \leq 1 - \gamma + \gamma/16 + \gamma/8 + \gamma/16 = 1 - 3\gamma/4 \tag{12}$$

In this sub-section, we derived an upper bound for the infinity norm of $\hat{\mathbf{z}}_{\mathcal{S}^c_{t_1}}$ which will be used later to bound the infinity norm of $\hat{\mathbf{z}}_{\mathcal{S}^c}$ defined in Eq. (5) to ensure strict dual feasibility. More importantly, our analysis also sheds light on the dependence of the sample complexity on parameters such as mutual incoherence, minimum eigenvalue, and other constants like $G_{\max}$ and $H_{\max}$, which helps us to study critical scientific limitations or behavior of the Lasso algorithm under adversarial attacks.

### 3.1.2 Choosing regularization parameter ($\lambda$)

In this subsection, we continue the discussion on strict dual feasibility and focus on how the adversary affects the regularization parameter. We start from $\hat{\mathbf{z}}_{\mathcal{S}^c_{t_2}}$ in Eq. (6), which is a $(p - k)$ dimensional random vector. Using properties of norms, whose details are mentioned in Appendix D.6 and using the bound derived in Eq. (12), we arrive at:

$$\left\|\hat{\mathbf{z}}_{\mathcal{S}^c_{t_2}}\right\|_\infty \leq \frac{1}{\lambda}\left\|\frac{1}{n}\mathbf{X}_{\mathcal{S}^c}^\intercal\mathbf{P}\mathbf{e}\right\|_\infty + \frac{1}{\lambda}\left\|\frac{1}{n}\mathbf{X}_{\mathcal{S}^c}^\intercal\boldsymbol{\Delta}\left(\mathbf{X}_{\mathcal{S}}^\star\right)\mathbf{w}_{\mathcal{S}}^\star\right\|_\infty + \frac{1}{\lambda}\left(1 - \frac{3\gamma}{4}\right)\left\|\frac{1}{n}\mathbf{X}_{\mathcal{S}}^\intercal\boldsymbol{\Delta}\left(\mathbf{X}_{\mathcal{S}}^\star\right)\mathbf{w}_{\mathcal{S}}^\star\right\|_\infty \tag{13}$$

**Intuition.** This decomposition enables us to analyze the effect of the adversarial perturbation on various model parameters. For example, the first term on the RHS of the above equation is concerned with the interaction of adversarial perturbations in $\mathcal{S}^c$ with $\mathbf{e}$. This term can be bounded by choosing an appropriate value of $\lambda$, as shown in the following lemma.

**Lemma 15.** *If $\lambda = \lambda_1 \geq \frac{q_1 \sigma_e}{\gamma} \sqrt{\frac{2 \log(p)}{n}}$, where constant $q_1^2 = 3 \left(C_{\max} + 2F_{\max} + D_{\max}\right)$, then $\left\|\mathbf{X}_{\mathcal{S}^c}^{\mathsf{T}} \mathbf{P} \mathbf{e} / n\lambda\right\|_{\infty} \leq \gamma/8$ with probability of at least $1 - \mathcal{O}\left(1/p\right)$.*

Similarly, the second term on the RHS of Eq. (13) signifies the interaction of adversarial perturbations with $\mathbf{X}_{\mathcal{S}^c}^{\star}$. It can be bounded by choosing a suitable value of $\lambda$, as shown in the following lemma.

**Lemma 16.** *If $\lambda = \lambda_2 \geq \frac{16}{\gamma} \max\left\{b_2, q_2 \sqrt{4 \log(p)/n}\right\}$, then $\left\|\mathbf{X}_{\mathcal{S}^c}^{\mathsf{T}} \mathbf{\Delta}(\mathbf{X}_{\mathcal{S}}^{\star}) \mathbf{w}_{\mathcal{S}}^{\star} / n\lambda\right\|_{\infty} \leq \gamma/8$ with probability of at least $1 - \mathcal{O}\left(1/p\right)$, where $b_2 = \left\|\mathbf{\Sigma}_{[p]\mathcal{S}}^{\mathbf{x}^{\star}\mathbf{\Delta}} \mathbf{w}_{\mathcal{S}}^{\star}\right\|_{\infty}$ and $q_2 = r\sqrt{\mathbf{w}_{\mathcal{S}}^{\star\mathsf{T}} \mathbf{\Sigma}_{\mathcal{SS}}^{\mathbf{\Delta}} \mathbf{w}_{\mathcal{S}}^{\star}} \max\limits_{i \in \mathcal{S}^c} \left(\sigma\sqrt{\mathbf{\Sigma}_{ii}^{\mathbf{x}}} + r\sqrt{\mathbf{\Sigma}_{ii}^{\mathbf{\Delta}}}\right)$.*

Similarly, the third term in RHS of Eq. (13) is concerned with the interaction of adversarial perturbations in $\mathbf{X}_{\mathcal{S}}^{\star}$. It can be bounded by selecting a suitable value of $\lambda$, presented in Lemma 24 in Appendix D. Substituting the bounds derived in Lemma 15, Lemma 16, and Lemma 24 in Eq. (13), we obtain:

$$\left\|\hat{\mathbf{z}}_{\mathcal{S}^c_{t_2}}\right\|_{\infty} \leq \gamma/8 + \gamma/8 + \gamma/8 = 3\gamma/8 \tag{14}$$

It should be noted that this bound is derived under some lower bound constraint on the regularization parameter. The lower bound can be obtained by taking the maximum of $\lambda_1, \lambda_2$, and $\lambda_3$ presented in Lemma 15, Lemma 16, and Lemma 24 respectively in Appendix D:

$$\lambda \geq \max\left\{\lambda_1, \lambda_2, \lambda_3\right\} = \max\left\{\frac{16b}{\gamma}, \frac{q_1 \sigma_e}{\gamma}\sqrt{\frac{2\log(p)}{n}}, \frac{16q}{\gamma}\sqrt{\frac{4\log(p)}{n}}\right\} \tag{15}$$

where $q = r\sqrt{\mathbf{w}_{\mathcal{S}}^{\star\mathsf{T}} \mathbf{\Sigma}_{\mathcal{SS}}^{\mathbf{\Delta}} \mathbf{w}_{\mathcal{S}}^{\star}} \max\limits_{i \in [p]} \left(\sigma\sqrt{\mathbf{\Sigma}_{ii}^{\mathbf{x}}} + r\sqrt{\mathbf{\Sigma}_{ii}^{\mathbf{\Delta}}}\right)$ and $b = \left\|\mathbf{\Sigma}_{[p]\mathcal{S}}^{\mathbf{x}^{\star}\mathbf{\Delta}} \mathbf{w}_{\mathcal{S}}^{\star}\right\|_{\infty}$. This completes the lower bound proof of $\lambda$ used in Theorem 10.

**Intuition.** Note that for a fixed budget, the adversary can increase the lower bound of $\lambda$ by designing $\mathbf{\Sigma}_{\mathcal{SS}}^{\mathbf{\Delta}}$ such that the eigenvector corresponding to the maximum eigenvalue of $\mathbf{\Sigma}_{\mathcal{SS}}^{\mathbf{\Delta}}$ is parallel to $\mathbf{w}_{\mathcal{S}}^{\star}$ to increase $q$ in Eq. (15). A higher value of the lower bound of $\lambda$ implies more penalization on the regression parameter vector, which might make the learning algorithm to incorrectly estimate the small non-zero parameters in $\mathcal{S}$ as zero.

Returning to the strict dual feasibility condition, the bound for $\hat{\mathbf{z}}_{\mathcal{S}^c}$ defined in Eq. (5) is derived by using the bound for $\hat{\mathbf{z}}_{\mathcal{S}^c_{t_2}}$ in Eq. (14) and the bound for $\hat{\mathbf{z}}_{\mathcal{S}^c_{t_1}}$ in Eq. (12) as follows:

$$||\hat{\mathbf{z}}_{\mathcal{S}^c}||_{\infty} \leq 1 - \frac{3\gamma}{4} + \frac{3\gamma}{8} = 1 - \frac{3\gamma}{8} < 1 \tag{16}$$

In this sub-section, we have verified the strict dual feasibility condition by proving that $||\hat{\mathbf{z}}_{\mathcal{S}^c}||_{\infty} < 1$ as $\gamma > 0$ in the above equation. This ensures that KKT conditions are met, which proves the first claim of Theorem 10, i.e., $\mathcal{S}(\hat{\mathbf{w}}) \subseteq \mathcal{S}(\mathbf{w}^{\star})$. It should be noted that we derive the lower bound constraint on $\lambda$ in order to provide theoretical guarantees. For practical purposes, we choose $\lambda = \mathcal{O}\left(\sqrt{\log(p)/n}\right)$ as done in the sparse regression literature (Wainwright, 2009; Ravikumar et al., 2009; 2010; 2011; Daneshmand et al., 2014).

The remaining claims on uniqueness and upper bound of $\|\hat{\mathbf{w}}_{\mathcal{S}} - \mathbf{w}_{\mathcal{S}}^{\star}\|_{\infty}$ are discussed in Appendix C.1 and Appendix C.2 respectively. In the next section, we discuss the interesting implications of Theorem 10 regarding the regimes when the adversary and Lasso can dominate over each other.

## 3.2 Adversary vs Lasso Territory

If $\lambda = \max\left\{\lambda_1, \lambda_2, \lambda_3\right\} = 16b/\gamma$, then we need the following condition as per Theorem 10:

$$\min_{i \in \mathcal{S}} |\mathbf{w}_i^{\star}| \geq 2f(\lambda) = 12G_{\max}\left(1 + 4/\gamma\right)b \tag{17}$$

This requirement on the lower bound of the absolute value of parameters in the support basically states that these coefficients should have significant values for detection. If the adversary is given more budget and

designs a large value of $b = \left\| \boldsymbol{\Sigma}_{[p]\mathcal{S}}^{\mathbf{x}^\star \boldsymbol{\Delta}} \mathbf{w}_{\mathcal{S}}^\star \right\|_\infty$ to break the above requirement (Eq. (17)), then we may not be able to provide theoretical guarantees for successful support recovery as proved in Lemma 9. Our theoretical analysis has identified the critical condition under which the adversary can design malicious attacks such that the Lasso algorithm may not have a high probability of successful support recovery (See Fig. 1(b)).

Consider the case with $\boldsymbol{\Sigma}_{[p]\mathcal{S}}^{\mathbf{x}^\star \boldsymbol{\Delta}} = \mathbf{0}_{p \times k}$, and hence $b = 0$. Therefore $\lambda = \max\{\lambda_1, \lambda_2, \lambda_3\} = \max\{\lambda_2, \lambda_3\} = \mathcal{O}\left(\sqrt{\log(p)/n}\right)$, then the previous requirement becomes:

$$\min_{i \in \mathcal{S}} |\mathbf{w}_i^\star| \geq 2f(\lambda) = \mathcal{O}\left(\sqrt{\log(p)/n}\right) \tag{18}$$

This condition can be easily fulfilled by increasing the value of $n$ sufficiently high depending on the value of $\min_{i \in \mathcal{S}} |\mathbf{w}_i^\star|$, and hence, theoretical guarantees can be established. The adversary can still try to break the above condition by increasing the value of $q_1$ or $q$ in $\lambda_2$ or $\lambda_3$ respectively, but the user can increase the sample size ($n$) accordingly as derived in various lemmas to ensure a high probability of success. Note that the condition $\min_{i \in \mathcal{S}} |\mathbf{w}_i^\star| \geq \left\| \boldsymbol{\Sigma}_{[p]\mathcal{S}}^{\mathbf{x}^\star \boldsymbol{\Delta}} \mathbf{w}_{\mathcal{S}}^\star \right\|_\infty = 0$ specified in Lemma 9 holds true for this case.

In this subsection, we completed the proof of Theorem 10 and discussed the critical regimes which may be favorable to the adversary or the learning algorithm. We also discussed the counter-intuitive result of how the adversarial perturbation in $\mathcal{S}^c$ can affect the guarantees for $\hat{\mathbf{w}}_\mathcal{S}$ indirectly by influencing the lower bound on the regularization parameter.

## 4 Experiments

In this section, we validate our proposed theoretical claims with empirical analysis on synthetic data and real-world data. Please refer to Appendix E for more details.

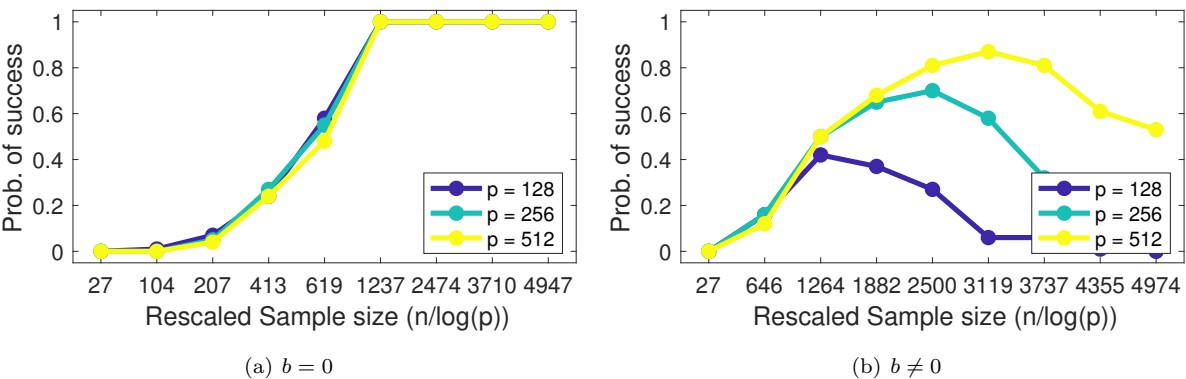

(a) $b = 0$          (b) $b \neq 0$

Figure 1: Probability of support recovery vs rescaled sample size.

**Synthetic data.** To verify the sample complexity result of $n = \Omega(\log(p))$ for fixed $k$, we repeat the support recovery experiment 200 times for a particular value of $(n, p)$. We report the empirical probability of successful support recovery. We perform this experiment for $p \in \{128, 256, 512\}$ and vary $n$ such that $\frac{n}{\log(p)} \in (25, 1250)$. The results presented in Figure 1(a) for $b = 0$ show that $\frac{n}{\log(p)}$ is not a function of $p$ (hence a constant), as the plots are overlapping (See Appendix E.1 for details regarding the data and adversarial perturbation generation.). For $b \neq 0$, Figure 1(b) shows that perfect support recovery is not achievable due to the dependence between adversarial perturbations and uncorrupted data (See Appendix E.4 for more details).

**Real-world data.** We used the BlogFeedback dataset (Buza, 2014) which contains 52397 samples and 276 features. We first recover the "true" support with the given data and further estimate the support from

adversarially corrupted data, which is generated by adding adversarial perturbations to all the features. The Lasso algorithm recovers the support with F1-score of 0.94.

## 5    Concluding Remarks

Sparsity, support recovery and weight recovery are tightly related concepts. In our paper, support relates to the zero/nonzero weights in the sparse regression vector (nonzero entries corresponding to relevant features for prediction) and weight recovery relates to estimating a vector that is close (in $\ell_\infty$ distance) to the true regression vector (See e.g., Theorem 10, Eq. (3)).

Our initial analysis for sparse regression already highlights some fundamental issues that not only pertain to sample complexity, e.g., impossibility. Our initial results could be later extended to other machine learning problems where sparsity as well as support and weight recovery is relevant. This includes for instance, Gaussian graphical models (Ravikumar et al., 2011), Ising models (Ravikumar et al., 2010), non-parametric regression (Ravikumar et al., 2009) and diffusion networks (Daneshmand et al., 2014).

Besides the aforementioned models, we believe our results could potentially motivate future work on neural networks. Indeed, there has been a recent interest on sparse neural networks (Liu & Wang, 2023) as well as weight recovery for two-layer neural networks (Bakshi et al., 2019).

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
