# A  Proofs of necessity for the assumptions made in Section 2.3

## A.1  Variance proxy parameter for adversarial perturbation

We derive the variance proxy parameter of a sub-Gaussian vector $\mathbf{\Delta}(\mathbf{x}^\star)$. This is done by first starting with a general random sub-Gaussian vector, $\mathbf{z}$ with variance proxy parameter $r^2$ and identity covariance matrix. The definition of sub-Gaussian vectors in (Hsu et al., 2012) states that for all $\mathbf{v} \in \mathbb{R}^p$:

$$\mathbb{E}\left[\exp\left(\mathbf{v}^\mathsf{T}\mathbf{z}\right)\right] \leq \exp\left(\frac{\|\mathbf{v}\|_2^2\, r^2}{2}\right) \tag{19}$$

Without loss of generality, we define $\mathbf{\Delta}(\mathbf{x}^\star) = (\mathbf{\Sigma}^{\mathbf{\Delta}})^{1/2}\mathbf{z}$, where $\mathbf{\Sigma}^{\mathbf{\Delta}}$ is the covariance matrix of $\mathbf{\Delta}(\mathbf{x}^\star)$. Substituting $\mathbf{z} = \mathbf{\Sigma}^{\mathbf{\Delta}^{-1/2}}\mathbf{\Delta}(\mathbf{x}^\star)$ in the above equation:

$$\mathbb{E}\left[\exp\left(\mathbf{v}^\mathsf{T}(\mathbf{\Sigma}^{\mathbf{\Delta}})^{-1/2}\mathbf{\Delta}(\mathbf{x}^\star)\right)\right] \leq \exp\left(\frac{\|\mathbf{v}\|_2^2\, r^2}{2}\right)$$

Substituting $\boldsymbol{\alpha}^\mathsf{T} = \mathbf{v}^\mathsf{T}(\mathbf{\Sigma}^{\mathbf{\Delta}})^{-1/2}$ in the above equation

$$\mathbb{E}\left[\exp\left(\boldsymbol{\alpha}^\mathsf{T}\mathbf{\Delta}(\mathbf{x}^\star)\right)\right] \leq \exp\left(\frac{\boldsymbol{\alpha}^\mathsf{T}\mathbf{\Sigma}^{\mathbf{\Delta}}\boldsymbol{\alpha}r^2}{2}\right) \tag{20}$$

which holds for all $\boldsymbol{\alpha} \in \mathbb{R}^p$.

## A.2  Existing adversarial attacks are a special case of our adversarial model

**Lemma 2.** Existing adversarial attacks (Goodfellow et al., 2015; Madry et al., 2018; Szegedy et al., 2014; Xing et al., 2021; Yin et al., 2019; Awasthi et al., 2020; Qin et al., 2021) are a special case of our adversarial model specified in Assumption 1.

*Proof.* Existing adversarial attacks (Goodfellow et al., 2015; Madry et al., 2018; Szegedy et al., 2014; Xing et al., 2021; Yin et al., 2019; Awasthi et al., 2020; Qin et al., 2021) are deterministic and bounded as they maximize the loss function within some $\epsilon$ norm ball and hence are a special case of our adversarial model specified in Assumption 1.

As seen in Lemma 6, the solution to the following maximization problem

$$\mathbf{\Delta}_{\mathrm{opt}}\left(\mathbf{x}^{\star(j)}\right) = \operatorname*{arg\,max}_{\|\mathbf{\Delta}\| \leq \epsilon}\left(\mathbf{w}^{\star\mathsf{T}}\left(\mathbf{x}^{\star(j)} + \mathbf{\Delta}\right) - y^{\star(j)}\right)^2$$

for any general norm constraint on $\mathbf{\Delta}$ is given by

$$\mathbf{\Delta}_{\mathrm{opt}}\left(\mathbf{x}^{\star(j)}\right) = \begin{cases} \pm\epsilon\frac{\mathbf{v}}{\|\mathbf{v}\|}, & \mathbf{w}_{\mathcal{S}}^{\star\mathsf{T}}\mathbf{x}^{(i)} - y^{(i)} = 0 \\ \operatorname{sign}(\mathbf{w}_{\mathcal{S}}^{\star\mathsf{T}}\mathbf{x}^{(i)} - y^{(i)})\epsilon\frac{\mathbf{v}}{\|\mathbf{v}\|} & \mathbf{w}_{\mathcal{S}}^{\star\mathsf{T}}\mathbf{x}^{(i)} - y^{(i)} \neq 0 \end{cases}$$

where $\mathbf{v} \in \partial\|\mathbf{w}_{\mathcal{S}}^\star\|_*$. It is obvious that the optima satisfies $\|\mathbf{\Delta}\| \leq \epsilon$ and hence is bounded.

Any vector with a bounded norm is a sub-Gaussian random vector, as shown below. Also, note that a random variable is more general than a deterministic one, i.e., a deterministic quantity is a random variable with a Dirac delta probability density function.

Assume $\mathbf{u}$ has bounded norm $\|\mathbf{u}\| \leq B$. Let the random variable $\mathbf{v} = \boldsymbol{\alpha}^\mathsf{T}\mathbf{u}$ for a fixed $\boldsymbol{\alpha}$. By Holder's inequality

$$\|\mathbf{v}\| \leq \|\boldsymbol{\alpha}\|_*\|\mathbf{u}\| \leq \|\boldsymbol{\alpha}\|_* B \leq C\|\boldsymbol{\alpha}\|_2 B$$

where $\|\cdot\|_*$ denotes the dual norm and is defined below for a general vector $\mathbf{a}$

$$\|\mathbf{b}\|_* = \sup\{\mathbf{b}^\mathsf{T}\mathbf{a} \mid \|\mathbf{a}\| \leq 1\}$$

and $C$ is related to the norm inequality $\|\boldsymbol{\alpha}\|_* \leq C\|\boldsymbol{\alpha}\|_2$. Now

$$\mathbb{E}[\exp(\boldsymbol{\alpha}^\mathsf{T}\mathbf{u})] = \mathbb{E}[\exp(\mathbf{v})] \leq \exp(C^2\|\boldsymbol{\alpha}\|_2^2 B^2/2)$$

which follows by boundedness of $\mathbf{v}$ and is the definition in (Hsu et al., 2012). □

## A.3  Huber model is a special case of our adversarial model

**Lemma 3.** The $\alpha$-Huber-model (Prasad et al., 2020; Diakonikolas et al., 2019) corrupting only $\alpha < 1$ fraction of samples is a special case of our adversarial model specified in Assumption 1.

*Proof.* In this section, we briefly discuss the Huber model or other existing models in prior literature and show that it is a special case of our adversarial model. Thus, the requirements of sub-Gaussianity (to allow for concentration inequalities), zero-mean (to avoid the issue described in Lemma 5) are not restrictive.

In our model, we assume noise-free measurements $\mathbf{x}^{\star(i)}$ and adversarial perturbation $\boldsymbol{\Delta}(\mathbf{x}^\star)^{(i)}$ for any sample $i \in [n]$ are generated from sub-Gaussian distribution, say $P$ and $N(\mathbf{x}^\star)$ respectively:

$$\mathbf{x}^\star \sim P, \qquad \boldsymbol{\Delta}(\mathbf{x}^\star) \sim N(\mathbf{x}^\star)$$

where we have dropped the superscript $i$ for brevity. Furthermore, $N(\mathbf{x}^\star)$ clarifies that the distribution N might depend on the value of $\mathbf{x}^\star$.

Further, we assume the adversarially corrupted data is given to the learner:

$$\mathbf{x} = \mathbf{x}^\star + \boldsymbol{\Delta}(\mathbf{x}^\star), \qquad y = \mathbf{w}^{\star\mathsf{T}}\mathbf{x}^\star + e \tag{21}$$

where $\mathbf{w}^\star$ is the true parameter vector. In Lemma 5, we have already shown that adversarial perturbation should be zero-mean otherwise support can be estimated trivially by computing the mean. Our further analysis relies on the population covariance matrix defined as:

$$\begin{bmatrix} \boldsymbol{\Sigma}^{\mathbf{x}^\star} & \boldsymbol{\Sigma}^{\mathbf{x}^\star\boldsymbol{\Delta}} \\ \boldsymbol{\Sigma}^{\boldsymbol{\Delta}\mathbf{x}^\star} & \boldsymbol{\Sigma}^{\boldsymbol{\Delta}} \end{bmatrix} = \mathbb{E}\left(\begin{bmatrix} \mathbf{x}^\star\mathbf{x}^{\star\mathsf{T}} & \mathbf{x}^\star\boldsymbol{\Delta}(\mathbf{x}^\star)^\mathsf{T} \\ \boldsymbol{\Delta}(\mathbf{x}^\star)\mathbf{x}^{\star\mathsf{T}} & \boldsymbol{\Delta}(\mathbf{x}^\star)\boldsymbol{\Delta}(\mathbf{x}^\star)^\mathsf{T} \end{bmatrix}\right) \tag{22}$$

In the $\alpha$-Huber-model (Prasad et al., 2020; Diakonikolas et al., 2019), only $\alpha < 1$ fraction of samples are corrupted and can be modeled as shown below:

$$z \sim B(1-\alpha), \qquad \mathbf{x}^\star \sim P, \qquad \Gamma(\mathbf{x}^\star) \sim Q(\mathbf{x}^\star)$$
$$\mathbf{x} = z\mathbf{x}^\star + (1-z)\Gamma(\mathbf{x}^\star), \qquad y = \mathbf{w}^{\star\mathsf{T}}\mathbf{x}^\star + e \tag{23}$$

where $B(1-\alpha)$ denotes a Bernoulli distribution (Uspensky et al., 1937) and $\Gamma(\mathbf{x}^\star)$ corresponds to corrupted samples following some arbitrary distribution $Q(\mathbf{x}^\star)$. The expected value can be computed as:

$$\mathbb{E}_z[\mathbf{x}] = (1-\alpha)\mathbf{x}^\star + \alpha\Gamma(\mathbf{x}^\star)$$

Further, the covariance matrix in this case will be:

$$\begin{bmatrix} \boldsymbol{\Sigma}^{\mathbf{x}^\star} & \boldsymbol{\Sigma}^{\mathbf{x}^\star\boldsymbol{\Delta}} \\ \boldsymbol{\Sigma}^{\boldsymbol{\Delta}\mathbf{x}^\star} & \boldsymbol{\Sigma}^{\boldsymbol{\Delta}} \end{bmatrix} = \mathbb{E}\left(\begin{bmatrix} (1-\alpha)^2\mathbf{x}^\star\mathbf{x}^{\star\mathsf{T}} & \alpha(1-\alpha)\mathbf{x}^\star\Gamma(\mathbf{x}^\star)^\mathsf{T} \\ \alpha(1-\alpha)\Gamma(\mathbf{x}^\star)\mathbf{x}^{\star\mathsf{T}} & \alpha^2\Gamma(\mathbf{x}^\star)\Gamma(\mathbf{x}^\star)^\mathsf{T} \end{bmatrix}\right) \tag{24}$$

Hence the covariance matrix in our case presented in Eq. (22) has very similar structure and mathematical properties as compared to the covariance matrix for the Huber model presented in Eq. (24). Therefore the analysis will follow exactly the same as ours.

In fact, we can show that the Huber model is a special case of our model by defining our adversarial perturbation, i.e., the distribution $N(\mathbf{x}^\star)$ as follows::

$$\mathbf{\Delta}(\mathbf{x}^\star) \sim \begin{cases} \mathbf{0} & \text{with probability } (1-\alpha) \\ -\mathbf{x}^\star + \Gamma(\mathbf{x}^\star), & \text{where } \Gamma(\mathbf{x}^\star) \sim Q(\mathbf{x}^\star) \quad \text{with probability } \alpha \end{cases}$$

Substituting the above particular of $\mathbf{\Delta}(\mathbf{x}^\star)$ in Eq. (21) will lead to the Huber model in Eq. (23). Hence our model is more general.

$\square$

### A.4 Attacking only support and non-support can be countered by computing mean

**Lemma 5.** If the adversary attacks only the support entries ($\mathbf{x}_{\mathcal{S}}$) or non-support entries ($\mathbf{x}_{\mathcal{S}^c}$) with non-zero-mean adversarial perturbation, then the learner can guess the support trivially with probability at least $1 - \mathcal{O}\left(\frac{1}{p}\right)$ if $n = \Omega\left(\log(p)\right)$.

*Proof.* Let the non-zero-mean adversarial perturbation have the form $\boldsymbol{\mu} = \begin{bmatrix} \mu_1 \mathbf{1}_k \\ \mu_2 \mathbf{1}_{p-k} \end{bmatrix}$. Also, for clarity, we assume that the first $k$ entries of $\mathbf{x}$ correspond to the support $\mathcal{S}$ and the rest correspond to the non-support $\mathcal{S}^c$. If this is not the case, the support and non-support entries will need to be properly interleaved. For the attack on $\mathcal{S}$ only, we consider the case $\mu_1 \neq 0$ and $\mu_2 = 0$. Similarly, for the attack on $\mathcal{S}^c$ only, we consider the case of $\mu_1 = 0$ and $\mu_2 \neq 0$.

We first analyze the sample mean of the entries in the support. For $n$ samples, that is $\mathbf{x}^{(j)} \in \mathbb{R}^p$ for $j \in [n]$, we can compute the sample mean. For the $i^{\text{th}}$ entry, denoted by $\mathbf{x}_i$, where $i \in \mathcal{S}$, we use the sub-Gaussian tail bound along with the union bound:

$$\mathbb{P}\left[(\exists i \in \mathcal{S}) \left| \frac{1}{n} \sum_{j=1}^n \mathbf{x}_i^{(j)} - \mu_1 \right| \geq t\right] \leq 2 \sum_{i \in \mathcal{S}} \exp\left\{\frac{-nt^2}{2\sigma_{x_i}^2}\right\} \tag{25}$$

where $\sigma_{x_i}^2$ denotes the variance proxy parameter. Similarly, the mean of the entries in the non-support can be analyzed as

$$\mathbb{P}\left[(\exists l \in \mathcal{S}^c) \left| \frac{1}{n} \sum_{j=1}^n \mathbf{x}_l^{(j)} - \mu_2 \right| \geq t\right] \leq 2 \sum_{l \in \mathcal{S}^c} \exp\left\{\frac{-nt^2}{2\sigma_{x_l}^2}\right\}$$

We substitute $t = \frac{\max(|\mu_1|, |\mu_2|)}{3}$ in the above equations. Now, if $\mu_1 = 0$ and $\mu_2 \neq 0$, we can claim:

$$\left| \frac{1}{n} \sum_{j=1}^n \mathbf{x}_i^{(j)} \right| \leq \frac{\max(|\mu_1|, |\mu_2|)}{3} = \frac{|\mu_2|}{3}$$

$$\left| \frac{1}{n} \sum_{j=1}^n \mathbf{x}_l^{(j)} \right| \geq |\mu_2| - \left| \frac{1}{n} \sum_{j=0}^n \mathbf{x}_l^{(j)} - \mu_2 \right| \geq |\mu_2| - \frac{\max(|\mu_1|, |\mu_2|)}{3} = 2\frac{|\mu_2|}{3}$$

with high probability of $1 - \mathcal{O}\left(\frac{1}{p}\right)$ if $n = \Omega\left(\log(p)\right)$. Note that the sample mean of the entries in the support is upper-bounded by $\frac{|\mu_2|}{3}$, whereas the sample mean of the entries in the non-support is lower bounded by $2\frac{|\mu_2|}{3}$. Hence the learner can guess the support easily by observing the concentration of the sample mean. Note that the case of $\mu_1 \neq 0$ and $\mu_2 = 0$ can be analyzed similarly.

$\square$

### A.5 Necessity of minimum eigenvalue assumption

**Lemma 17.** *If Assumption 7 is violated, meaning $C_{\min} = 0$, then the Lasso solution will not be unique.*

*Proof.* Consider the simple case of $\boldsymbol{\Delta}(\mathbf{x}^\star) = \mathbf{0}$ for all samples and having infinite samples, meaning $n \to \infty$. In such a case, the minimum eigenvalue of the Hessian matrix derived at the end of Section D.1 will be $C_{\min}$. It should be noted that $C_{\min} < 0$ is not possible as covariance matrices are known to be positive semidefinite. Hence, the mild violation of the assumption only happens if $C_{\min} = 0$. As the Hessian is not positive definite, the loss function is not strictly convex and hence the optimal solution is not unique. $\qquad\square$

### A.6 Our adversarial model is more challenging than existing adversarial models

**Lemma 6.** The support can be trivially estimated in $n = \Omega\left(\log(p)\right)$ with high probability of $1 - \mathcal{O}\left(\frac{1}{p}\right)$ under existing adversarial attacks (Goodfellow et al., 2015; Madry et al., 2018; Szegedy et al., 2014; Xing et al., 2021; Yin et al., 2019; Awasthi et al., 2020; Qin et al., 2021) aiming to maximize a per-sample loss function.

*Proof.* In the existing literature, the adversarial attack is usually derived for the particular constraint of Euclidean norm or $\ell_\infty$ norm or general $\ell_p$ norm. In the following derivation, we present the proof for any general norm.

The attack generated by FGSM and its variants will attack only the support for each sample because it focuses on maximizing the loss function only, rather than making the estimation of parameters tougher. Mathematically, $y^{\star(j)} = \mathbf{w}^{\star\mathsf{T}}\mathbf{x}^{\star(j)} = \mathbf{w}_{\mathcal{S}}^{\star\mathsf{T}}\mathbf{x}_{\mathcal{S}}^{\star(j)}$ for any sample $j \in [n]$. FGSM and its variants maximize the loss function with respect to adversarial perturbation $\boldsymbol{\Delta}$ within some $\epsilon$ ball to obtain the optimal:

$$\boldsymbol{\Delta}_{\text{opt}}\left(\mathbf{x}^{\star(j)}\right) = \underset{\|\boldsymbol{\Delta}\| \leq \epsilon}{\arg\max}\left(\mathbf{w}^{\star\mathsf{T}}\left(\mathbf{x}^{\star(j)} + \boldsymbol{\Delta}\right) - y^{\star(j)}\right)^2$$

$$= \underset{\|\boldsymbol{\Delta}\| \leq \epsilon}{\arg\max}\left(\mathbf{w}_{\mathcal{S}}^{\star\mathsf{T}}\left(\mathbf{x}^{\star(j)} + \boldsymbol{\Delta}\right) - y^{\star(j)}\right)^2$$

We solve the problem for two cases of $\mathbf{w}_{\mathcal{S}}^{\star\mathsf{T}}\mathbf{x}^{(i)} - y^{(i)} = 0$ and $\mathbf{w}_{\mathcal{S}}^{\star\mathsf{T}}\mathbf{x}^{(i)} - y^{(i)} \neq 0$.

1. For $\mathbf{w}_{\mathcal{S}}^{\star\mathsf{T}}\mathbf{x}^{(i)} - y^{(i)} = 0$, the problem reduces to dual norm problem as shown below:

$$\sup_{\|\boldsymbol{\Delta}\| \leq \epsilon} \mathbf{w}_{\mathcal{S}}^{\star\mathsf{T}}\boldsymbol{\Delta}$$

Using Holder's inequality, we can claim:

$$\mathbf{w}_{\mathcal{S}}^{\star\mathsf{T}}\boldsymbol{\Delta} \leq \|\mathbf{w}_{\mathcal{S}}^\star\|_* \|\boldsymbol{\Delta}\| \leq \epsilon \|\mathbf{w}_{\mathcal{S}}^\star\|_*$$

where $\|\cdot\|_*$ denotes the dual norm for any general norm. Therefore to compute $\boldsymbol{\Delta}_{\text{opt}}\left(\mathbf{x}^{\star(j)}\right)$, we need to find the solution for

$$\boldsymbol{\Delta}_{\text{opt}}\left(\mathbf{x}^{\star(j)}\right) = \{\boldsymbol{\Delta} : \mathbf{w}_{\mathcal{S}}^{\star\mathsf{T}}\boldsymbol{\Delta} = \|\mathbf{w}_{\mathcal{S}}^\star\|, \|\boldsymbol{\Delta}\|_* \leq 1\}$$

To compute the optimal point, we use the sub-differential of a norm defined as follows for any general vector $\mathbf{a}$:

$$\partial \|\mathbf{a}\| = \{\mathbf{v} : \mathbf{v}^\mathsf{T}\mathbf{a} = \|\mathbf{a}\|, \|\mathbf{a}\|_* \leq 1\}$$

where $\|\cdot\|_*$ is the dual norm to $\|.\|$. Using this, we claim $\boldsymbol{\Delta}_{\text{opt}}\left(\mathbf{x}^{\star(j)}\right) \in \epsilon\partial \|\mathbf{w}_{\mathcal{S}}^\star\|_*$. As the original objective function is quadratic, $-\boldsymbol{\Delta}_{\text{opt}}\left(\mathbf{x}^{\star(j)}\right)$ can also be a solution.

2. For $\mathbf{w}_{\mathcal{S}}^{\star\mathsf{T}}\mathbf{x}^{(i)} - y^{(i)} \neq 0$, we first consider the case of $\mathbf{w}_{\mathcal{S}}^{\star\mathsf{T}}\mathbf{x}^{(i)} - y^{(i)} > 0$. The objective function to maximize $\left(\mathbf{w}_{\mathcal{S}}^{\mathsf{T}}\mathbf{x}^{(i)} - y^{(i)} + \mathbf{w}_{\mathcal{S}}^{\mathsf{T}}\mathbf{\Delta}\right)^2$ can be expressed as maximizing $\mathbf{w}_{\mathcal{S}}^{\mathsf{T}}\mathbf{\Delta}$ because $\mathbf{w}_{\mathcal{S}}^{\mathsf{T}}\mathbf{x}^{(i)} - y^{(i)}$ is a positive constant and not a function of $\mathbf{\Delta}$. As discussed earlier, the optimal solution will be $\mathbf{\Delta}_{\mathrm{opt}}\left(\mathbf{x}^{\star(j)}\right) \in \epsilon\partial\left\|\mathbf{w}_{\mathcal{S}}^{\star}\right\|_*$. Using similar arguments for the case of $\mathbf{w}_{\mathcal{S}}^{\star\mathsf{T}}\mathbf{x}^{(i)} - y^{(i)} < 0$, the solution is given by $\mathbf{\Delta}_{\mathrm{opt}}\left(\mathbf{x}^{\star(j)}\right) \in -\epsilon\partial\left\|\mathbf{w}_{\mathcal{S}}^{\star}\right\|_*$.

Hence, the overall solution is given by:

$$\mathbf{\Delta}_{\mathrm{opt}}\left(\mathbf{x}^{\star(j)}\right) = \begin{cases} \pm\epsilon\frac{\mathbf{v}}{\|\mathbf{v}\|}, & \mathbf{w}_{\mathcal{S}}^{\star\mathsf{T}}\mathbf{x}^{(i)} - y^{(i)} = 0 \\ \mathrm{sign}(\mathbf{w}_{\mathcal{S}}^{\star\mathsf{T}}\mathbf{x}^{(i)} - y^{(i)})\epsilon\frac{\mathbf{v}}{\|\mathbf{v}\|} & \mathbf{w}_{\mathcal{S}}^{\star\mathsf{T}}\mathbf{x}^{(i)} - y^{(i)} \neq 0 \end{cases}$$

where $\mathbf{v} \in \partial\left\|\mathbf{w}_{\mathcal{S}}^{\star}\right\|_*$.

The important thing to note is that the optimal solution is a function of the support only. As the adversary is attacking only the support , the learner can guess the support in $n = \Omega\left(\log(p)\right)$ with a high probability of $1 - \mathcal{O}\left(1/p\right)$ as shown in Lemma 5. $\qquad\square$

### A.7 Necessity of mutual incoherence assumption

**Lemma 18.** *If for some $\gamma > 0$, if $\left\|\mathbf{\Sigma}_{\mathcal{S}^c\mathcal{S}}^{\mathbf{x}}\left(\mathbf{\Sigma}_{\mathcal{S}\mathcal{S}}^{\mathbf{x}}\right)^{-1}\mathrm{sign}(\mathbf{w}_{\mathcal{S}}^{\star})\right\|_\infty > 1+\gamma$, then support recovery is not possible, meaning $\mathrm{sign}(\hat{\mathbf{w}}_i) \neq \mathrm{sign}(\mathbf{w}_i^{\star})$ for some $i \in \mathcal{S}$, even if we have infinite samples.*

*Proof.* Consider the simple case with no adversarial attack, i.e., $\mathbf{\Delta}(\mathbf{x}^{\star}) = \mathbf{0}$ for all samples. We can directly use Theorem 2(a) from (Wainwright, 2009) to prove the claim. Theorem 2(a) from (Wainwright, 2009) shows that dual feasibility does not hold if the mutual incoherence condition is violated, i.e., if $\left\|\mathbf{\Sigma}_{\mathcal{S}^c\mathcal{S}}^{\mathbf{x}}\left(\mathbf{\Sigma}_{\mathcal{S}\mathcal{S}}^{\mathbf{x}}\right)^{-1}\mathrm{sign}(\mathbf{w}_{\mathcal{S}}^{\star})\right\|_\infty > 1 + \gamma$. Hence, $\mathbf{w}^{\star}$ is not optimal as one of the KKT conditions (i.e., dual feasibility) is not satisfied. Furthermore, since the solution $\hat{\mathbf{w}}$ fulfills dual feasibility, i.e., $\left\|\mathbf{\Sigma}_{\mathcal{S}^c\mathcal{S}}^{\mathbf{x}}\left(\mathbf{\Sigma}_{\mathcal{S}\mathcal{S}}^{\mathbf{x}}\right)^{-1}\mathrm{sign}(\hat{\mathbf{w}}_{\mathcal{S}})\right\|_\infty \leq 1$, we know that $\mathrm{sign}(\hat{\mathbf{w}}_i) \neq \mathrm{sign}(\mathbf{w}_i^{\star})$ for some $i \in \mathcal{S}$, which proves our claim.

$\qquad\square$

## B Necessary Condition: Proof of Lemma 9

**Lemma 9.** The condition $\frac{16b}{15\gamma}\left(1 + \frac{\gamma}{4}\right)\frac{3G_{\max}}{2} \leq \min_{i\in\mathcal{S}}|\mathbf{w}_i^{\star}|$ is necessary for support recovery, even in the population regime, where (intuitively speaking) the learner has access to an infinite number of samples.

*Proof.* For simplicity, assume $\mathbf{x}^{\star}$ follows a Gaussian distribution with zero mean. Similarly, assume $\mathbf{\Delta}(\mathbf{x}^{\star})$ follows a Gaussian distribution with zero mean but has certain correlation with $\mathbf{x}^{\star}$. Also, assume $e$ is zero-mean Gaussian independent of $\mathbf{x}^{\star}$ and $\mathbf{\Delta}(\mathbf{x}^{\star})$. Further, assume:

$$\mathbf{\Sigma}^{\mathbf{x}^{\star}} = \mathbb{E}\left[\mathbf{x}^{\star}\mathbf{x}^{\star\mathsf{T}}\right] = \mathbf{I}, \qquad \mathbf{\Sigma}^{\mathbf{x}} = \mathbb{E}\left[\mathbf{x}\mathbf{x}^{\mathsf{T}}\right] = 2\mathbf{I}$$

Most importantly we assume the learner does not know $\mathbf{\Sigma}^{\mathbf{x}^{\star}\mathbf{\Delta}}$, defined as:

$$\mathbf{\Sigma}^{\mathbf{x}^{\star}\mathbf{\Delta}} = \mathbb{E}\left[\mathbf{x}^{\star}\mathbf{\Delta}(\mathbf{x}^{\star})^{\mathsf{T}}\right]$$

The distribution of $(\mathbf{x}, y)$, since it is multivariate Gaussian, is fully identifiable from its first and second order moments:

$$\mathbb{E}\begin{bmatrix}\mathbf{x} \\ y\end{bmatrix} = \begin{bmatrix}\mathbb{E}[\mathbf{x}] \\ \mathbb{E}[y]\end{bmatrix}, \qquad \mathbb{E}\left[\begin{bmatrix}\mathbf{x} \\ y\end{bmatrix}\begin{bmatrix}\mathbf{x} \\ y\end{bmatrix}^{\mathsf{T}}\right] = \begin{bmatrix}2\mathbf{I} & \mathbb{E}[\mathbf{x}y] \\ \mathbb{E}[\mathbf{x}y]^{\mathsf{T}} & \mathbb{E}[y^2]\end{bmatrix}$$

It is easy to observe that $\mathbb{E}[\mathbf{x}] = \mathbb{E}[\mathbf{x}^\star] + \mathbb{E}[\mathbf{\Delta}(\mathbf{x}^\star)] = \mathbf{0}$ and $\mathbb{E}[y] = \mathbb{E}[\mathbf{x}^\star]^\intercal \mathbf{w}^\star + \mathbb{E}[e] = 0$. Further:

$$\begin{aligned}
\mathbb{E}\left[\mathbf{x}y\right] &= \mathbb{E}\left[\left(\mathbf{x}^\star + \mathbf{\Delta}(\mathbf{x}^\star)\right)\left(\mathbf{x}^{\star\intercal}\mathbf{w}^\star + e\right)\right] \\
&= \left(\mathbb{E}\left[\mathbf{x}^\star \mathbf{x}^{\star\intercal}\right] + \mathbb{E}\left[\mathbf{\Delta}(\mathbf{x}^\star)\mathbf{x}^{\star\intercal}\right]\right)\mathbf{w}^\star + \mathbb{E}\left[\mathbf{x}^\star + \mathbf{\Delta}(\mathbf{x}^\star)\right]\mathbb{E}\left[e\right] \\
&= \left(\mathbf{I} + \mathbf{\Sigma}^{\mathbf{x}^\star \mathbf{\Delta}}\right)\mathbf{w}^\star
\end{aligned}$$

Similarly, note that:

$$\begin{aligned}
\mathbb{E}[y^2] &= \mathbb{E}\left[\left(\mathbf{x}^{\star\intercal}\mathbf{w}^\star + e\right)^2\right] \\
&= \mathbf{w}^{\star\intercal}\mathbb{E}\left[\mathbf{x}^\star \mathbf{x}^{\star\intercal}\right]\mathbf{w}^\star + 2\mathbb{E}\left[\mathbf{x}^{\star\intercal}e\right] + \mathbb{E}\left[e^2\right] \\
&= \|\mathbf{w}^\star\|_2^2 + \mathbb{E}[e^2]
\end{aligned}$$

Note that the learner has to estimate the support $\mathcal{S}$ from two quantities $\mathbb{E}\left[\mathbf{x}y\right] = \left(\mathbf{I} + \mathbf{\Sigma}^{\mathbf{x}^\star \mathbf{\Delta}}\right)\mathbf{w}^\star$ and $\mathbb{E}[y^2] = \|\mathbf{w}^\star\|_2^2 + \mathbb{E}[e^2]$ without knowing $\mathbf{\Sigma}^{\mathbf{x}^\star \mathbf{\Delta}}$.

Now let us assume there exists a case such that $\frac{16b}{15\gamma}\left(1 + \frac{\gamma}{4}\right)\frac{3G_{\max}}{2} = \frac{16\left\|\mathbf{\Sigma}^{\mathbf{x}^\star \mathbf{\Delta}}_{[p]\mathcal{S}}\ \mathbf{w}^\star_{\mathcal{S}}\right\|_\infty}{15\gamma}\left(1 + \frac{\gamma}{4}\right)\frac{3G_{\max}}{2} \geq \min_{i\in\mathcal{S}}|\mathbf{w}^\star_i|$.
Since $\mathbf{\Sigma}^\mathbf{x} = 2\mathbf{I}$, we have $G_{\max} = \left\|\left|\left(\mathbf{\Sigma}^\mathbf{x}_{\mathcal{S}\mathcal{S}}\right)^{-1}\right|\right\|_\infty = 1/2$ and $\left\|\left|\mathbf{\Sigma}^\mathbf{x}_{\mathcal{S}^c\mathcal{S}}\left(\mathbf{\Sigma}^\mathbf{x}_{\mathcal{S}\mathcal{S}}\right)^{-1}\right|\right\|_\infty = 0 \leq 1 - \gamma$, which implies $\gamma = 1$. Substituting these values, we arrive at:

$$\frac{16\left\|\mathbf{\Sigma}^{\mathbf{x}^\star \mathbf{\Delta}}_{[p]\mathcal{S}}\ \mathbf{w}^\star_{\mathcal{S}}\right\|_\infty}{15\gamma}\left(1 + \frac{\gamma}{4}\right)\frac{3G_{\max}}{2} = \frac{16\left\|\mathbf{\Sigma}^{\mathbf{x}^\star \mathbf{\Delta}}_{[p]\mathcal{S}}\ \mathbf{w}^\star_{\mathcal{S}}\right\|_\infty}{15}\left(1 + \frac{1}{4}\right)\frac{3}{4} = \left\|\mathbf{\Sigma}^{\mathbf{x}^\star \mathbf{\Delta}}_{[p]\mathcal{S}}\ \mathbf{w}^\star_{\mathcal{S}}\right\|_\infty \geq \min_{i\in\mathcal{S}}|\mathbf{w}^\star_i|$$

Now we consider two cases discussed below for $\left\|\mathbf{\Sigma}^{\mathbf{x}^\star \mathbf{\Delta}}_{[p]\mathcal{S}}\ \mathbf{w}^\star_{\mathcal{S}}\right\|_\infty \geq \min_{i\in\mathcal{S}}|\mathbf{w}^\star_i|$:

1. If $\mathbf{w}^\star_i > 0$ and $\left(\mathbf{\Sigma}^{\mathbf{x}^\star \mathbf{\Delta}}\mathbf{w}^\star\right)_i \leq -\mathbf{w}^\star_i$, then $\mathbb{E}\left[\mathbf{x}y\right]_i = \left(\left(\mathbf{I} + \mathbf{\Sigma}^{\mathbf{x}^\star \mathbf{\Delta}}\right)\mathbf{w}^\star\right)_i = \mathbf{w}^\star_i + \left(\mathbf{\Sigma}^{\mathbf{x}^\star \mathbf{\Delta}}\mathbf{w}^\star\right)_i \leq 0$.

2. If $\mathbf{w}^\star_i < 0$ and $\left(\mathbf{\Sigma}^{\mathbf{x}^\star \mathbf{\Delta}}\mathbf{w}^\star\right)_i \geq -\mathbf{w}^\star_i$, then $\mathbb{E}\left[\mathbf{x}y\right]_i = \left(\left(\mathbf{I} + \mathbf{\Sigma}^{\mathbf{x}^\star \mathbf{\Delta}}\right)\mathbf{w}^\star\right)_i = \mathbf{w}^\star_i + \left(\mathbf{\Sigma}^{\mathbf{x}^\star \mathbf{\Delta}}\mathbf{w}^\star\right)_i \geq 0$.

In both the cases, $\text{sign}\left(\mathbb{E}\left[\mathbf{x}y\right]_i\right) \neq \text{sign}\left(\mathbf{w}^\star_i\right)$. Hence support recovery can not be done under a case with $\left\|\mathbf{\Sigma}^{\mathbf{x}^\star \mathbf{\Delta}}_{[p]\mathcal{S}}\ \mathbf{w}^\star_{\mathcal{S}}\right\|_\infty \geq \min_{i\in\mathcal{S}}|\mathbf{w}^\star_i|$.

The final step of the proof consists on showing that there exists such a case. The distribution of $(\mathbf{x}^\star, \mathbf{\Delta}(\mathbf{x}^\star))$ is a multivariate Gaussian, fully identifiable from its first and second order moments:

$$\mathbb{E}\begin{bmatrix}\mathbf{x}^\star \\ \mathbf{\Delta}(\mathbf{x}^\star)\end{bmatrix} = \begin{bmatrix}\mathbf{0} \\ \mathbf{0}\end{bmatrix}, \qquad \mathbf{\Sigma} = \mathbb{E}\left[\begin{bmatrix}\mathbf{x}^\star \\ \mathbf{\Delta}(\mathbf{x}^\star)\end{bmatrix}\begin{bmatrix}\mathbf{x}^\star \\ \mathbf{\Delta}(\mathbf{x}^\star)\end{bmatrix}^\intercal\right] = \begin{bmatrix}\mathbf{I} & \mathbf{\Sigma}^{\mathbf{x}^\star \mathbf{\Delta}} \\ \mathbf{\Sigma}^{\mathbf{\Delta}\mathbf{x}^\star} & \mathbf{\Sigma}^{\mathbf{\Delta}}\end{bmatrix}$$

**Example 1.** The adversary can choose to set $\mathbf{\Sigma}^{\mathbf{x}^\star \mathbf{\Delta}} = \mathbf{\Sigma}^{\mathbf{\Delta}\mathbf{x}^\star} = -2\mathbf{I}$ and $\mathbf{\Sigma}^{\mathbf{\Delta}} = 5\mathbf{I}$, which leads to a positive definite covariance matrix $\mathbf{\Sigma}$. First, note that $\mathbf{\Sigma}^\mathbf{x} = \mathbb{E}\left[\mathbf{x}\mathbf{x}^\intercal\right] = \mathbb{E}\left[(\mathbf{x}^\star + \mathbf{\Delta}(\mathbf{x}^\star))(\mathbf{x}^\star + \mathbf{\Delta}(\mathbf{x}^\star))^\intercal\right] = \mathbf{\Sigma}^{\mathbf{x}^\star} + \mathbf{\Sigma}^{\mathbf{x}^\star \mathbf{\Delta}} + \mathbf{\Sigma}^{\mathbf{\Delta}\mathbf{x}^\star} + \mathbf{\Sigma}^{\mathbf{\Delta}} = 2\mathbf{I}$. Second, note that $\mathbf{\Sigma}^{\mathbf{x}^\star \mathbf{\Delta}}\mathbf{w}^\star = -2\mathbf{w}^\star$ which fits cases 1 and 2 discussed above. Finally, to further illustrate how the adversary can perturb a sample $\mathbf{x}^\star$, by properties of conditional distributions for multivariate Gaussians, we have that $\mathbf{\Delta}(\mathbf{x}^\star)$ given $\mathbf{x}^\star$ follows a Gaussian distribution with mean $\mathbf{\Sigma}^{\mathbf{x}^\star \mathbf{\Delta}}(\mathbf{\Sigma}^{\mathbf{\Delta}})^{-1}\mathbf{x}^\star = -\frac{2}{5}\mathbf{x}^\star$ and covariance $\mathbf{\Sigma}^{\mathbf{\Delta}} - \mathbf{\Sigma}^{\mathbf{x}^\star \mathbf{\Delta}}\mathbf{\Sigma}^{\mathbf{\Delta}\mathbf{x}^\star} = \mathbf{I}$. That is, $\mathbf{\Delta}(\mathbf{x}^\star) \sim \mathcal{N}(-\frac{2}{5}\mathbf{x}^\star, \mathbf{I})$.

**Example 2.** To give a more challenging case, the adversary can choose to set $\mathbf{\Sigma}^{\mathbf{x}^\star \mathbf{\Delta}} = \mathbf{\Sigma}^{\mathbf{\Delta}\mathbf{x}^\star} = -\mathbf{I}$ and $\mathbf{\Sigma}^{\mathbf{\Delta}} = 3\mathbf{I}$, which leads to a positive definite covariance matrix $\mathbf{\Sigma}$. First, note that $\mathbf{\Sigma}^\mathbf{x} = \mathbb{E}\left[\mathbf{x}\mathbf{x}^\intercal\right] = \mathbb{E}\left[(\mathbf{x}^\star + \mathbf{\Delta}(\mathbf{x}^\star))(\mathbf{x}^\star + \mathbf{\Delta}(\mathbf{x}^\star))^\intercal\right] = \mathbf{\Sigma}^{\mathbf{x}^\star} + \mathbf{\Sigma}^{\mathbf{x}^\star \mathbf{\Delta}} + \mathbf{\Sigma}^{\mathbf{\Delta}\mathbf{x}^\star} + \mathbf{\Sigma}^{\mathbf{\Delta}} = 2\mathbf{I}$. Second, note that $\mathbb{E}\left[\mathbf{x}y\right] = \left(\mathbf{I} + \mathbf{\Sigma}^{\mathbf{x}^\star \mathbf{\Delta}}\right)\mathbf{w}^\star = \mathbf{0}$, thus, the learner is not getting any information regarding the support. Finally, to further illustrate how the adversary can perturb a sample $\mathbf{x}^\star$, by properties of conditional distributions for multivariate Gaussians, we have that $\mathbf{\Delta}(\mathbf{x}^\star)$ given $\mathbf{x}^\star$ follows a Gaussian distribution with mean $\mathbf{\Sigma}^{\mathbf{x}^\star \mathbf{\Delta}}(\mathbf{\Sigma}^{\mathbf{\Delta}})^{-1}\mathbf{x}^\star = -\frac{1}{3}\mathbf{x}^\star$ and covariance $\mathbf{\Sigma}^{\mathbf{\Delta}} - \mathbf{\Sigma}^{\mathbf{x}^\star \mathbf{\Delta}}\mathbf{\Sigma}^{\mathbf{\Delta}\mathbf{x}^\star} = 2\mathbf{I}$. That is, $\mathbf{\Delta}(\mathbf{x}^\star) \sim \mathcal{N}(-\frac{1}{3}\mathbf{x}^\star, 2\mathbf{I})$. $\qquad\square$

# C  Proof for Uniqueness and Upper bound on $\|\hat{\mathbf{w}}_{\mathcal{S}} - \mathbf{w}_{\mathcal{S}}^{\star}\|_{\infty}$

## C.1  Uniqueness of the solution

In this sub-section, we prove the uniqueness of the optimal solution $\hat{\mathbf{w}}_{\mathcal{S}}$. We need the second order derivative, $\left[\nabla^2 l((\mathbf{w}_{\mathcal{S}}, \mathbf{0}))\right]_{\mathcal{S}\mathcal{S}} = \frac{1}{n}\mathbf{X}_{\mathcal{S}}^{\mathsf{T}}\mathbf{X}_{\mathcal{S}}$ (computed in Appendix D.1) to be positive definite for the problem in Eq. (2) to be strictly convex in the support space (See Eq. (33) in the appendix for a formal definition). The positive definiteness of a submatrix of the sample covariance is proved in the following lemma.

**Lemma 19.** *If Assumption 7 holds and $n = \Omega\left(k\log(p)\right)$, then we have:*

$$\mathbb{P}\left[\lambda_{\min}\left(\frac{\mathbf{X}_{\mathcal{S}}^{\mathsf{T}}\mathbf{X}_{\mathcal{S}}}{n}\right) \geq \frac{(C_{\min} + 2F_{\min} + D_{\min})}{2}\right] \geq 1 - \mathcal{O}\left(\frac{1}{p}\right)$$

Hence $\left[\nabla^2 l((\mathbf{w}_{\mathcal{S}}, \mathbf{0}))\right]_{\mathcal{S}\mathcal{S}} = \frac{1}{n}\mathbf{X}_{\mathcal{S}}^{\mathsf{T}}\mathbf{X}_{\mathcal{S}}$ is positive definite. More importantly, as the Hessian matrix depends only on adversarial perturbation in the support $\mathcal{S}$, sample complexity in the above lemma is not impacted by perturbation in the non-support $\mathcal{S}^c$. But this does not imply allocating more budget to $\mathcal{S}$ to design perturbation is recommended from the adversary's perspective, as more budget to $\mathcal{S}$ may lead to increasing $D_{\min}$, which is advantageous for the learning algorithm. In a more formal way, we need to bound $\lambda_{\min}\left(\mathbf{\Delta}(\mathbf{X}_{\mathcal{S}}^{\star\mathsf{T}})\mathbf{\Delta}(\mathbf{X}_{\mathcal{S}}^{\star})/n\right)$ while proving Lemma 19, which requires $n = \Omega\left((k\log(p))/D_{\min}^2\right)$ samples (Eq. (65)). Hence, it is advisable for the adversary to design perturbations such that $D_{\min}$ is small.

With a brief discussion on uniqueness in this sub-section, we provide theoretical guarantees for the estimated regression parameter vector in the next subsection.

## C.2  Quality of estimated regression parameter vector

In this subsection, we prove the third claim made in Theorem 10 and discuss how the adversarial perturbation in the non-support $\mathcal{S}^c$ can affect the theoretical guarantees for $\hat{\mathbf{w}}_{\mathcal{S}}$ (in the support) indirectly through regularization parameter. We start with the computation of $\hat{\mathbf{w}}_{\mathcal{S}} - \mathbf{w}_{\mathcal{S}}^{\star}$ by using the first order stationary condition specified in Eq. (4). The algebraic steps are presented in Appendix D.1 and the simplified expression is:

$$\|\hat{\mathbf{w}}_{\mathcal{S}} - \mathbf{w}_{\mathcal{S}}^{\star}\|_{\infty} \leq \left|\left|\left|\mathbf{A}^{-1}\right|\right|\right|_{\infty}\left(\|\mathbf{w_1}\|_{\infty} + \|\mathbf{w_2}\|_{\infty}\right) + \lambda\left|\left|\left|\mathbf{A}^{-1}\right|\right|\right|_{\infty}\|\hat{\mathbf{z}}_{\mathcal{S}}\|_{\infty} \tag{26}$$

$$\mathbf{w_1} = \frac{\mathbf{X}_{\mathcal{S}}^{\mathsf{T}}\mathbf{e}}{n}, \quad \mathbf{w_2} = \frac{\mathbf{X}_{\mathcal{S}}^{\mathsf{T}}\mathbf{\Delta}\left(\mathbf{X}_{\mathcal{S}}^{\star}\right)\mathbf{w}_{\mathcal{S}}^{\star}}{n}, \quad \mathbf{A} = \frac{\mathbf{X}_{\mathcal{S}}^{\mathsf{T}}\mathbf{X}_{\mathcal{S}}}{n} \tag{27}$$

The last term in RHS of Eq. (26) can be easily bounded as $\|\hat{\mathbf{z}}_{\mathcal{S}}\|_{\infty} \leq 1$. To further bound $\left|\left|\left|\mathbf{A}^{-1}\right|\right|\right|_{\infty}$, we use the triangle inequality:

$$\left|\left|\left|\mathbf{A}^{-1}\right|\right|\right|_{\infty} \leq \left|\left|\left|\mathbf{A}^{-1} - (\mathbb{E}\left[\mathbf{A}\right])^{-1}\right|\right|\right|_{\infty} + \left|\left|\left|(\mathbb{E}\left[\mathbf{A}\right])^{-1}\right|\right|\right|_{\infty}$$

The first term in the RHS of the above equation can be bounded using Lemma 13. We can claim

$$\mathbb{P}\left[\left|\left|\left|\mathbf{A}^{-1} - (\mathbb{E}\left[\mathbf{A}\right])^{-1}\right|\right|\right|_{\infty} \geq \frac{G_{\max}}{2}\right] \leq \mathcal{O}\left(\frac{1}{p}\right) \tag{28}$$

by substituting $\delta = G_{\max}/2$ in Lemma 13 if $n = \Omega\left(k^2\log(p)/G_{\max}^2\right)$. Using this, we can claim $\left|\left|\left|\mathbf{A}^{-1}\right|\right|\right|_{\infty} \leq 3G_{\max}/2$. Further we proceed to bound $\|\mathbf{w}_1\|_{\infty}$ defined in Eq. (27) by using an approach similar to Lemma 15:

$$\|\mathbf{w}_1\|_{\infty} = \left\|\frac{\mathbf{X}_{\mathcal{S}}^{\mathsf{T}}\mathbf{e}}{n}\right\|_{\infty} \leq \frac{\lambda\gamma}{8} \tag{29}$$

It should be noted that there is lower bound constraint on $\lambda$ for the above statement to hold with high probability, as specified in Lemma 15. The lower bound value of $\lambda$ can be tightened slightly for this case specifically by changing the $\log(p)$ factor to $\log(k)$ as $\mathbf{w}_1$ is a $k-$dimensional vector, and we need to take

union bound over $k$ elements only, instead of $p - k$ as done in Lemma 15. But we take the $\lambda$ mentioned in Eq. (15), so that the strict dual feasibility is also verified.

Further, we proceed to bound $\|\mathbf{w}_2\|_\infty$ defined in Eq. (27) by using the approach similar to Lemma 24 presented in Appendix D. We claim:

$$\|\mathbf{w_2}\|_\infty = \left\| \frac{\mathbf{X}_\mathcal{S}^\mathsf{T} \boldsymbol{\Delta}\left(\mathbf{X}_\mathcal{S}^\star\right)\mathbf{w}_\mathcal{S}^\star}{n} \right\|_\infty \leq \frac{\lambda\gamma}{8} \tag{30}$$

where the lower bound on $\lambda$ is specified in Eq. (15). Substituting the bounds derived in Eq. (29) and Eq. (30) in Eq. (26), we obtain:

$$\|\hat{\mathbf{w}}_\mathcal{S} - \mathbf{w}_\mathcal{S}^\star\|_\infty \leq \lambda\left(1 + \frac{\gamma}{4}\right)\frac{3G_{\max}}{2} = f(\lambda) \tag{31}$$

This proves the third claim in Eq. (3) of Theorem 10. From the above equation, we observe that a large value regularization $\lambda$ is not desirable as it is directly proportional to the bound of $\|\hat{\mathbf{w}}_\mathcal{S} - \mathbf{w}_\mathcal{S}^\star\|_\infty$. But note that the lower bound of $\lambda$ can be controlled by the adversary due to the presence of constants $b$ and $q$ in Eq. (15), and hence the adversary can control the quality of the estimated regression parameter vector as demonstrated shortly. Before proceeding to that discussion, we need to prove the fourth claim of sign matching in Theorem 10, which can be seen as a direct consequence of Lemma 26 in the Appendix.

## D Proof for KKT conditions

### D.1 First Order Stationarity condition

Consider the loss function

$$l(\mathbf{w}) = \frac{1}{2n}\|\mathbf{y} - \mathbf{X}\mathbf{w}\|_2^2. \tag{32}$$

The Lasso problem is given by:

$$\hat{\mathbf{w}}_\mathcal{S} = \underset{\mathbf{w}_\mathcal{S} \in \mathbb{R}^k}{\arg\min}\, l((\mathbf{w}_\mathcal{S}, \mathbf{0})) + \lambda\|\mathbf{w}_\mathcal{S}\|_1. \tag{33}$$

We start with the first-order stationary condition. Taking the first order derivative of Eq. (32), we get:

$$\nabla l(\mathbf{w}) = \frac{1}{n}\mathbf{X}^\mathsf{T}\left(\mathbf{X}(\mathbf{w} - \mathbf{w}^\star) + (\boldsymbol{\Delta}\left(\mathbf{X}^\star\right)\mathbf{w}^\star - \mathbf{e})\right)$$

$$[\nabla l((\hat{\mathbf{w}}_\mathcal{S}, \mathbf{0}))]_\mathcal{S} = \frac{1}{n}\mathbf{X}_\mathcal{S}^\mathsf{T}\left(\mathbf{X}_\mathcal{S}(\hat{\mathbf{w}}_\mathcal{S} - \mathbf{w}_\mathcal{S}^\star) + (\boldsymbol{\Delta}\left(\mathbf{X}_\mathcal{S}^\star\right)\mathbf{w}_\mathcal{S}^\star - \mathbf{e})\right)$$

$$[\nabla l((\hat{\mathbf{w}}_\mathcal{S}, \mathbf{0}))]_{\mathcal{S}^c} = \frac{1}{n}\mathbf{X}_{\mathcal{S}^c}^\mathsf{T}\left(\mathbf{X}_\mathcal{S}(\hat{\mathbf{w}}_\mathcal{S} - \mathbf{w}_\mathcal{S}^\star) + (\boldsymbol{\Delta}\left(\mathbf{X}_\mathcal{S}^\star\right)\mathbf{w}_\mathcal{S}^\star - \mathbf{e})\right)$$

The stationarity condition of Eq. (33), after splitting into the support $\mathcal{S}$ and non-support $\mathcal{S}^c$, becomes:

$$[\nabla l((\hat{\mathbf{w}}_\mathcal{S}, \mathbf{0}))]_\mathcal{S} + \lambda\hat{\mathbf{z}}_\mathcal{S} = \mathbf{0}_k$$

$$[\nabla l((\hat{\mathbf{w}}_\mathcal{S}, \mathbf{0}))]_{\mathcal{S}^c} + \lambda\hat{\mathbf{z}}_{\mathcal{S}^c} = \mathbf{0}_{(p-k)}$$

Using these equations, we arrive at:

$$\frac{1}{n}\mathbf{X}_\mathcal{S}^\mathsf{T}\left(\mathbf{X}_\mathcal{S}(\hat{\mathbf{w}}_\mathcal{S} - \mathbf{w}_\mathcal{S}^\star) + (\boldsymbol{\Delta}\left(\mathbf{X}_\mathcal{S}^\star\right)\mathbf{w}_\mathcal{S}^\star - \mathbf{e})\right) + \lambda\hat{\mathbf{z}}_\mathcal{S} = \mathbf{0}$$

$$(\hat{\mathbf{w}}_\mathcal{S} - \mathbf{w}_\mathcal{S}^\star) = \left(\mathbf{X}_\mathcal{S}^\mathsf{T}\mathbf{X}_\mathcal{S}\right)^{-1}\left(\mathbf{X}_\mathcal{S}^\mathsf{T}\left(\mathbf{e} - \boldsymbol{\Delta}\left(\mathbf{X}_\mathcal{S}^\star\right)\mathbf{w}_\mathcal{S}^\star\right) - n\lambda\hat{\mathbf{z}}_\mathcal{S}\right) \tag{34}$$

Further, using triangle inequality and sub-multiplicative property of norms, we arrive at:

$$\|\hat{\mathbf{w}}_\mathcal{S} - \mathbf{w}_\mathcal{S}^\star\|_\infty \leq \left\|\left(\frac{\mathbf{X}_\mathcal{S}^\mathsf{T}\mathbf{X}_\mathcal{S}}{n}\right)^{-1}\right\|_\infty \left(\left\|\frac{\mathbf{X}_\mathcal{S}^\mathsf{T}\mathbf{e}}{n}\right\|_\infty + \left\|\frac{\mathbf{X}_\mathcal{S}^\mathsf{T}\boldsymbol{\Delta}\left(\mathbf{X}_\mathcal{S}^\star\right)\mathbf{w}_\mathcal{S}^\star}{n}\right\|_\infty + \lambda\|\hat{\mathbf{z}}_\mathcal{S}\|_\infty\right) \tag{35}$$

Now, $\hat{\mathbf{z}}_{\mathcal{S}^c}$ can be computed as

$$
\begin{aligned}
\hat{\mathbf{z}}_{\mathcal{S}^c} &= -\frac{1}{\lambda n}\mathbf{X}_{\mathcal{S}^c}^\intercal \left(\mathbf{X}_{\mathcal{S}}(\hat{\mathbf{w}}_{\mathcal{S}} - \mathbf{w}_{\mathcal{S}}^\star) + (\mathbf{\Delta}\,(\mathbf{X}_{\mathcal{S}}^\star)\mathbf{w}_{\mathcal{S}}^\star - \mathbf{e})\right) \\
&= -\frac{1}{\lambda n}\mathbf{X}_{\mathcal{S}^c}^\intercal \left(\mathbf{X}_{\mathcal{S}}\left((\mathbf{X}_{\mathcal{S}}^\intercal\mathbf{X}_{\mathcal{S}})^{-1}(\mathbf{X}_{\mathcal{S}}^\intercal\,(\mathbf{e} - \mathbf{\Delta}\,(\mathbf{X}_{\mathcal{S}}^\star)\mathbf{w}_{\mathcal{S}}^\star) - n\lambda\hat{\mathbf{z}}_{\mathcal{S}})\right) + (\mathbf{\Delta}\,(\mathbf{X}_{\mathcal{S}}^\star)\mathbf{w}_{\mathcal{S}}^\star - \mathbf{e})\right) \\
&= \mathbf{X}_{\mathcal{S}^c}^\intercal \left\{\mathbf{X}_{\mathcal{S}}\,(\mathbf{X}_{\mathcal{S}}^\intercal\mathbf{X}_{\mathcal{S}})^{-1}\,\hat{\mathbf{z}}_{\mathcal{S}} + \left(\mathbf{I}_n - \mathbf{X}_{\mathcal{S}}\,(\mathbf{X}_{\mathcal{S}}^\intercal\mathbf{X}_{\mathcal{S}})^{-1}\,\mathbf{X}_{\mathcal{S}}^\intercal\right)\frac{(\mathbf{e} - \mathbf{\Delta}\,(\mathbf{X}_{\mathcal{S}}^\star)\mathbf{w}_{\mathcal{S}}^\star)}{\lambda n}\right\}
\end{aligned}
\tag{36}
$$

where $\mathbf{I}_n$ denotes an identity matrix of dimension $n \times n$.

The second order derivative of Eq. (32) is:

$$
\nabla^2 l(\mathbf{w}) = \frac{1}{n}\mathbf{X}^\intercal\mathbf{X}
$$

$$
\left[\nabla^2 l((\mathbf{w}_{\mathcal{S}}, \mathbf{0}))\right]_{\mathcal{S},\mathcal{S}} = \frac{1}{n}\mathbf{X}_{\mathcal{S}}^\intercal\mathbf{X}_{\mathcal{S}}
$$

### D.2 Simplification of $\left|\left|\hat{\mathbf{z}}_{\mathcal{S}^c_{t_1}}\right|\right|_\infty$

In this sub-section, we present the simplification of the term $\left|\left|\hat{\mathbf{z}}_{\mathcal{S}^c_{t_1}}\right|\right|_\infty$, which uses the triangle inequality as shown below:

$$
\begin{aligned}
\left|\left|\left|\mathbf{X}_{\mathcal{S}^c}^\intercal\mathbf{X}_{\mathcal{S}}\,(\mathbf{X}_{\mathcal{S}}^\intercal\mathbf{X}_{\mathcal{S}})^{-1}\right|\right|\right|_\infty &\leq \left|\left|\left|\left(\frac{1}{n}\mathbf{X}_{\mathcal{S}^c}^\intercal\mathbf{X}_{\mathcal{S}} - \mathbf{\Sigma}_{\mathcal{S}^c\mathcal{S}}^{\mathbf{x}} + \mathbf{\Sigma}_{\mathcal{S}^c\mathcal{S}}^{\mathbf{x}}\right)\left(\frac{1}{n}\mathbf{X}_{\mathcal{S}}^\intercal\mathbf{X}_{\mathcal{S}}\right)^{-1}\right|\right|\right|_\infty \\
&\leq \left|\left|\left|\left(\frac{1}{n}\mathbf{X}_{\mathcal{S}^c}^\intercal\mathbf{X}_{\mathcal{S}} - \mathbf{\Sigma}_{\mathcal{S}^c\mathcal{S}}^{\mathbf{x}}\right)\left(\frac{1}{n}\mathbf{X}_{\mathcal{S}}^\intercal\mathbf{X}_{\mathcal{S}}\right)^{-1}\right|\right|\right|_\infty + \left|\left|\left|\mathbf{\Sigma}_{\mathcal{S}^c\mathcal{S}}^{\mathbf{x}}\left(\frac{1}{n}\mathbf{X}_{\mathcal{S}}^\intercal\mathbf{X}_{\mathcal{S}}\right)^{-1}\right|\right|\right|_\infty \\
&= \left|\left|\left|\left(\frac{1}{n}\mathbf{X}_{\mathcal{S}^c}^\intercal\mathbf{X}_{\mathcal{S}} - \mathbf{\Sigma}_{\mathcal{S}^c\mathcal{S}}^{\mathbf{x}}\right)\left(\left(\frac{1}{n}\mathbf{X}_{\mathcal{S}}^\intercal\mathbf{X}_{\mathcal{S}}\right)^{-1} - (\mathbf{\Sigma}_{\mathcal{S}\mathcal{S}}^{\mathbf{x}})^{-1} + (\mathbf{\Sigma}_{\mathcal{S}\mathcal{S}}^{\mathbf{x}})^{-1}\right)\right|\right|\right|_\infty \\
&\quad + \left|\left|\left|\mathbf{\Sigma}_{\mathcal{S}^c\mathcal{S}}^{\mathbf{x}}\left(\left(\frac{1}{n}\mathbf{X}_{\mathcal{S}}^\intercal\mathbf{X}_{\mathcal{S}}\right)^{-1} - (\mathbf{\Sigma}_{\mathcal{S}\mathcal{S}}^{\mathbf{x}})^{-1} + (\mathbf{\Sigma}_{\mathcal{S}\mathcal{S}}^{\mathbf{x}})^{-1}\right)\right|\right|\right|_\infty \\
&\leq \left|\left|\left|\left(\frac{1}{n}\mathbf{X}_{\mathcal{S}^c}^\intercal\mathbf{X}_{\mathcal{S}} - \mathbf{\Sigma}_{\mathcal{S}^c\mathcal{S}}^{\mathbf{x}}\right)\left(\left(\frac{1}{n}\mathbf{X}_{\mathcal{S}}^\intercal\mathbf{X}_{\mathcal{S}}\right)^{-1} - (\mathbf{\Sigma}_{\mathcal{S}\mathcal{S}}^{\mathbf{x}})^{-1}\right)\right|\right|\right|_\infty \\
&\quad + \left|\left|\left|\left(\frac{1}{n}\mathbf{X}_{\mathcal{S}^c}^\intercal\mathbf{X}_{\mathcal{S}} - \mathbf{\Sigma}_{\mathcal{S}^c\mathcal{S}}^{\mathbf{x}}\right)(\mathbf{\Sigma}_{\mathcal{S}\mathcal{S}}^{\mathbf{x}})^{-1}\right|\right|\right|_\infty \\
&\quad + \left|\left|\left|\mathbf{\Sigma}_{\mathcal{S}^c\mathcal{S}}^{\mathbf{x}}\left(\left(\frac{1}{n}\mathbf{X}_{\mathcal{S}}^\intercal\mathbf{X}_{\mathcal{S}}\right)^{-1} - (\mathbf{\Sigma}_{\mathcal{S}\mathcal{S}}^{\mathbf{x}})^{-1}\right)\right|\right|\right|_\infty + \left|\left|\left|\mathbf{\Sigma}_{\mathcal{S}^c\mathcal{S}}^{\mathbf{x}}(\mathbf{\Sigma}_{\mathcal{S}\mathcal{S}}^{\mathbf{x}})^{-1}\right|\right|\right|_\infty
\end{aligned}
$$

Let $\mathbf{R} = \frac{1}{n}\mathbf{X}_{\mathcal{S}^c}^\intercal\mathbf{X}_{\mathcal{S}}$ and $\mathbf{Q} = \frac{1}{n}\mathbf{X}_{\mathcal{S}}^\intercal\mathbf{X}_{\mathcal{S}}$, and hence $\mathbb{E}\,[\mathbf{R}] = \mathbf{\Sigma}_{\mathcal{S}^c\mathcal{S}}^{\mathbf{x}}$, $\mathbb{E}\,[\mathbf{Q}] = \mathbf{\Sigma}_{\mathcal{S}\mathcal{S}}^{\mathbf{x}}$. The above expression simplifies to:

$$
\begin{aligned}
\left|\left|\left|\mathbf{X}_{\mathcal{S}^c}^\intercal\mathbf{X}_{\mathcal{S}}\,(\mathbf{X}_{\mathcal{S}}^\intercal\mathbf{X}_{\mathcal{S}})^{-1}\right|\right|\right|_\infty &\leq \left|\left|\left|\mathbb{E}\,[\mathbf{R}]\,(\mathbb{E}\,[\mathbf{Q}])^{-1}\right|\right|\right|_\infty + \left|\left|\left|\mathbb{E}\,[\mathbf{R}]\left(\mathbf{Q}^{-1} - (\mathbb{E}\,[\mathbf{Q}])^{-1}\right)\right|\right|\right|_\infty \\
&\quad + \left|\left|\left|(\mathbf{R} - \mathbb{E}\,[\mathbf{R}])\left(\mathbf{Q}^{-1} - (\mathbb{E}\,[\mathbf{Q}])^{-1}\right)\right|\right|\right|_\infty + \left|\left|\left|(\mathbf{R} - \mathbb{E}\,[\mathbf{R}])\,(\mathbb{E}\,[\mathbf{Q}])^{-1}\right|\right|\right|_\infty
\end{aligned}
$$

### D.3 Proof of Lemma 12

**Lemma 12.** For $0 \leq \delta \leq 32\xi k$, where $\xi = \max_{i \in \mathcal{S}} \left( \sigma\sqrt{\mathbf{\Sigma_{ii}^x}} + r\sqrt{\mathbf{\Sigma_{ii}^\Delta}} \right) \max_{j \in \mathcal{S}^c} \left( \sigma\sqrt{\mathbf{\Sigma_{jj}^x}} + r\sqrt{\mathbf{\Sigma_{jj}^\Delta}} \right)$, if $n = \Omega\left( \frac{k^2\xi^2}{\delta^2} \log(p) \right)$, then

$$\mathbb{P}\left[ \left\| \left\| \frac{1}{n}\mathbf{X}_{\mathcal{S}^c}^\mathsf{T}\mathbf{X}_{\mathcal{S}} - \mathbf{\Sigma}_{\mathcal{S}^c\mathcal{S}}^x \right\| \right\|_\infty \leq \delta \right] \geq 1 - \mathcal{O}\left( \frac{1}{p} \right)$$

*Proof.* We start by analyzing each entry of $\frac{1}{n}\mathbf{X}_{\mathcal{S}^c}^\mathsf{T}\mathbf{X}_{\mathcal{S}}$. As $\mathbf{x} = \mathbf{x}^\star + \mathbf{\Delta}(\mathbf{x}^\star)$, we can claim $\mathbf{x}_i \sim SG(0, \sigma\sqrt{\mathbf{\Sigma_{ii}^x}} + r\sqrt{\mathbf{\Sigma_{ii}^\Delta}})$ using Lemma 27. Further as $\mathbf{X}_{ki}$ and $\mathbf{X}_{kj}$ are sub-Gaussian, its product is sub-exponentially distributed, denoted by $SE(8\sqrt{2}c_{ij}, 4c_{ij})$ using Lemma 28 where $c_{ij} = \left( \sigma\sqrt{\mathbf{\Sigma_{ii}^x}} + r\sqrt{\mathbf{\Sigma_{ii}^\Delta}} \right)\left( \sigma\sqrt{\mathbf{\Sigma_{jj}^x}} + r\sqrt{\mathbf{\Sigma_{jj}^\Delta}} \right)$. By using properties of sub-exponential distributions, we can further claim the following for entry $(i,j)$:

$$\left( \frac{1}{n}\mathbf{X}_{\mathcal{S}^c}^\mathsf{T}\mathbf{X}_{\mathcal{S}} \right)_{ij} \sim SE\left( \frac{8\sqrt{2}\xi}{\sqrt{n}}, \frac{4\xi}{n} \right)$$

where $\xi = \max_{i \in \mathcal{S}} \left( \sigma\sqrt{\mathbf{\Sigma_{ii}^x}} + r\sqrt{\mathbf{\Sigma_{ii}^\Delta}} \right) \max_{j \in \mathcal{S}^c} \left( \sigma\sqrt{\mathbf{\Sigma_{jj}^x}} + r\sqrt{\mathbf{\Sigma_{jj}^\Delta}} \right)$. Applying Lemma 23 for $\frac{1}{n}\mathbf{X}_{\mathcal{S}^c}^\mathsf{T}\mathbf{X}_{\mathcal{S}}$, we arrive at:

$$\mathbb{P}\left[ \left\| \left\| \frac{1}{n}\mathbf{X}_{\mathcal{S}^c}^\mathsf{T}\mathbf{X}_{\mathcal{S}} - \mathbf{\Sigma}_{\mathcal{S}^c\mathcal{S}}^x \right\| \right\|_\infty \geq \delta \right] \leq 2(p-k)k \exp\left\{ \frac{-n\delta^2}{256k^2\xi^2} \right\}$$

for $0 \leq \delta \leq 32\xi k$. If we choose $n = \Omega\left( \frac{k^2\xi^2}{\delta^2} \log(p) \right)$, then we may claim:

$$\mathbb{P}\left[ \left\| \left\| \frac{1}{n}\mathbf{X}_{\mathcal{S}^c}^\mathsf{T}\mathbf{X}_{\mathcal{S}} - \mathbf{\Sigma}_{\mathcal{S}^c\mathcal{S}}^x \right\| \right\|_\infty \leq \delta \right] \geq 1 - \mathcal{O}\left( \frac{1}{p} \right).$$

$\square$

### D.4 Proof of Lemma 13

**Lemma 13.** For any $\delta > 0$, if $n = \Omega\left( \frac{k^2}{\delta^2(C_{\min}+2F_{\min}+D_{\min})^4} \log(p) \right)$

$$\mathbb{P}\left[ \left\| \left\| \left( \frac{1}{n}\mathbf{X}_{\mathcal{S}}^\mathsf{T}\mathbf{X}_{\mathcal{S}} \right)^{-1} - (\mathbf{\Sigma}_{\mathcal{S}\mathcal{S}}^x)^{-1} \right\| \right\|_\infty \leq \delta \right] \geq 1 - \mathcal{O}\left( \frac{1}{p} \right)$$

*Proof.* We start by applying norm inequalities to arrive to the spectral norm:

$$\left\| \left\| \left( \frac{1}{n}\mathbf{X}_{\mathcal{S}}^\mathsf{T}\mathbf{X}_{\mathcal{S}} \right)^{-1} - (\mathbf{\Sigma}_{\mathcal{S}\mathcal{S}}^x)^{-1} \right\| \right\|_\infty \leq \sqrt{k} \left\| \left( \frac{1}{n}\mathbf{X}_{\mathcal{S}}^\mathsf{T}\mathbf{X}_{\mathcal{S}} \right)^{-1} - (\mathbf{\Sigma}_{\mathcal{S}\mathcal{S}}^x)^{-1} \right\|_2$$

$$\leq \sqrt{k} \left\| \left( \frac{1}{n}\mathbf{X}_{\mathcal{S}}^\mathsf{T}\mathbf{X}_{\mathcal{S}} \right)^{-1} \left( \mathbf{\Sigma}_{\mathcal{S}\mathcal{S}}^x - \frac{1}{n}\mathbf{X}_{\mathcal{S}}^\mathsf{T}\mathbf{X}_{\mathcal{S}} \right) (\mathbf{\Sigma}_{\mathcal{S}\mathcal{S}}^x)^{-1} \right\|_2$$

$$\leq \sqrt{k} \left\| \left( \frac{1}{n}\mathbf{X}_{\mathcal{S}}^\mathsf{T}\mathbf{X}_{\mathcal{S}} \right)^{-1} \right\|_2 \left\| \frac{1}{n}\mathbf{X}_{\mathcal{S}}^\mathsf{T}\mathbf{X}_{\mathcal{S}} - \mathbf{\Sigma}_{\mathcal{S}\mathcal{S}}^x \right\|_2 \left\| (\mathbf{\Sigma}_{\mathcal{S}\mathcal{S}}^x)^{-1} \right\|_2 \qquad (37)$$

The term $\left\|\left(\mathbf{\Sigma}_{\mathcal{SS}}^{\mathbf{x}}\right)^{-1}\right\|_2$ in the above equation can be bounded as shown below:

$$\lambda_{\min}\left(\mathbf{\Sigma}_{\mathcal{SS}}^{\mathbf{x}}\right) \geq \lambda_{\min}\left(\mathbf{\Sigma}_{\mathcal{SS}}^{\mathbf{x}^{\star}}\right) + 2\lambda_{\min}\left([\mathbf{\Sigma}^{\mathbf{\Delta x}^{\star}} + \mathbf{\Sigma}^{\mathbf{x}^{\star}\mathbf{\Delta}}]_{\mathcal{SS}}/2\right) + \lambda_{\min}\left(\mathbf{\Sigma}_{\mathcal{SS}}^{\mathbf{\Delta}}\right)$$
$$= C_{\min} + 2F_{\min} + D_{\min} \tag{38}$$

We use Lemma 19 to claim $\left\|\left(\frac{1}{n}\mathbf{X}_{\mathcal{S}}^{\mathsf{T}}\mathbf{X}_{\mathcal{S}}\right)^{-1}\right\|_2 \leq \frac{2}{C_{\min}+2F_{\min}+D_{\min}}$ with high probability of $1 - \mathcal{O}\left(\frac{1}{p}\right)$ if $n = \Omega\left(k\log(p)\right)$. Substituting this bound and Eq. (38) in Eq. (37):

$$\left\|\left\|\left(\frac{1}{n}\mathbf{X}_{\mathcal{S}}^{\mathsf{T}}\mathbf{X}_{\mathcal{S}}\right)^{-1} - \left(\mathbf{\Sigma}_{\mathcal{SS}}^{\mathbf{x}}\right)^{-1}\right\|\right\|_{\infty} \leq \sqrt{k}\frac{2}{(C_{\min} + 2F_{\min} + D_{\min})^2}\left\|\frac{1}{n}\mathbf{X}_{\mathcal{S}}^{\mathsf{T}}\mathbf{X}_{\mathcal{S}} - \mathbf{\Sigma}_{\mathcal{SS}}^{\mathbf{x}}\right\|_2 \tag{39}$$

We further proceed to bound $\left\|\frac{1}{n}\mathbf{X}_{\mathcal{S}}^{\mathsf{T}}\mathbf{X}_{\mathcal{S}} - \mathbf{\Sigma}_{\mathcal{SS}}^{\mathbf{x}}\right\|_2$ in Eq. (39):

$$\left\|\frac{1}{n}\mathbf{X}_{\mathcal{S}}^{\mathsf{T}}\mathbf{X}_{\mathcal{S}} - \mathbf{\Sigma}_{\mathcal{SS}}^{\mathbf{x}}\right\|_2 \leq \left\|\frac{1}{n}\mathbf{X}_{\mathcal{S}}^{\star\mathsf{T}}\mathbf{X}_{\mathcal{S}}^{\star} - \mathbf{\Sigma}_{\mathcal{SS}}^{\mathbf{x}^{\star}}\right\|_2 + 2\left\|\frac{1}{n}\mathbf{X}_{\mathcal{S}}^{\star\mathsf{T}}\mathbf{\Delta}\left(\mathbf{X}_{\mathcal{S}}^{\star}\right) - \mathbf{\Sigma}_{\mathcal{SS}}^{\mathbf{\Delta x}^{\star}}\right\|_2$$
$$+ \left\|\frac{1}{n}\mathbf{\Delta}\left(\mathbf{X}_{\mathcal{S}}^{\star\mathsf{T}}\right)\mathbf{\Delta}\left(\mathbf{X}_{\mathcal{S}}^{\star}\right) - \mathbf{\Sigma}_{\mathcal{SS}}^{\mathbf{\Delta}}\right\|_2 \tag{40}$$

The first term in the RHS of the above equation can be easily bounded by substituting $\delta = (C_{\min} + 2F_{\min} + D_{\min})^2\frac{\delta_1}{8\sqrt{k}}$ in Eq. (66) to claim

$$\mathbb{P}\left[\left\|\frac{1}{n}\mathbf{X}_{\mathcal{S}}^{\star\mathsf{T}}\mathbf{X}_{\mathcal{S}}^{\star} - \mathbf{\Sigma}_{\mathcal{SS}}^{\mathbf{x}^{\star}}\right\|_2 \leq \frac{(C_{\min} + 2F_{\min} + D_{\min})^2}{8}\frac{\delta_1}{\sqrt{k}}\right] \geq 1 - \mathcal{O}\left(\frac{1}{p}\right) \tag{41}$$

if $n = \Omega\left(\frac{k^2}{\delta_1^2(C_{\min}+F_{\min})^4} + \log(p)\right)$. The third term, $\left\|\frac{1}{n}\mathbf{\Delta}\left(\mathbf{X}_{\mathcal{S}}^{\star\mathsf{T}}\right)\mathbf{\Delta}\left(\mathbf{X}_{\mathcal{S}}^{\star}\right) - \mathbf{\Sigma}_{\mathcal{SS}}^{\mathbf{\Delta}}\right\|_2$ in the RHS of Eq. (40) can also be bounded in similar manner with same sample complexity. The second term in Eq. (40) can be bounded by substituting $\delta = \frac{(C_{\min}+2F_{\min}+D_{\min})^2}{8}\frac{\delta_1}{\sqrt{k}}$ in Theorem 14

$$\mathbb{P}\left[\left\|\frac{1}{n}\mathbf{X}_{\mathcal{S}}^{\star\mathsf{T}}\mathbf{\Delta}\left(\mathbf{X}_{\mathcal{S}}^{\star}\right) - \mathbf{\Sigma}_{\mathcal{SS}}^{\mathbf{\Delta x}^{\star}}\right\|_2 \leq \frac{(C_{\min} + 2F_{\min} + D_{\min})^2}{8}\frac{\delta_1}{\sqrt{k}}\right] \geq 1 - \mathcal{O}\left(\frac{1}{p}\right) \tag{42}$$

if $n = \Omega\left(\frac{k^2}{\delta_1^2(C_{\min}+2F_{\min}+D_{\min})^4}\log(p)\right)$. Further, we substitute Eq. (41) and Eq. (42) in Eq. (40) to claim the following

$$\left\|\frac{1}{n}\mathbf{X}_{\mathcal{S}}^{\mathsf{T}}\mathbf{X}_{\mathcal{S}} - \mathbf{\Sigma}_{\mathcal{SS}}^{\mathbf{x}}\right\|_2 \leq \frac{(C_{\min} + 2F_{\min} + D_{\min})^2}{2}\frac{\delta_1}{\sqrt{k}} \tag{43}$$

with probability $1 - \mathcal{O}\left(\frac{1}{p}\right)$. Substituting Eq. (43) in Eq. (39) and replacing the dummy variable $\delta_1$ with $\delta$, we arrive at the claimed result. $\square$

### D.5 Proof of Theorem 14

**Theorem 14.** For $0 < \delta < \frac{32r\sigma ab_k}{n}$, $\mathbf{\Delta}\left(\mathbf{X}_{\mathcal{S}}^{\star}\right), \mathbf{X}_{\mathcal{S}}^{\star} \in \mathbb{R}^{n \times k}$, $a^2 = \max_{j \in \mathcal{S}}\mathbf{\Sigma}_{jj}$, $b_k^2 = k\sum_{i \in \mathcal{S}}\left(\mathbf{\Sigma}_{ii}^{\mathbf{\Delta}}\right)^2$

$$\mathbb{P}\left[\left\|\frac{1}{n}\mathbf{X}_{\mathcal{S}}^{\star\mathsf{T}}\mathbf{\Delta}\left(\mathbf{X}_{\mathcal{S}}^{\star}\right) - \mathbf{\Sigma}_{\mathcal{SS}}^{\mathbf{\Delta x}^{\star}}\right\|_2 \geq \delta\right] \leq 4e^{\frac{-n\delta^2}{256r^2\sigma^2 ab_k}} \tag{44}$$

*Proof.* Let $\mathbf{B} = \mathbf{\Delta}\left(\mathbf{X}_{\mathcal{S}}^{\star\mathsf{T}}\right)\mathbf{X}_{\mathcal{S}}^{\star}$ and a matrix $\mathbf{Q}$ be defined as:

$$\mathbf{Q} = \begin{bmatrix} \mathbf{0}_{k \times k} & \mathbf{B} \\ \mathbf{B}^{\mathsf{T}} & \mathbf{0}_{k \times k} \end{bmatrix} \tag{45}$$

Using Lemma 20, $||\mathbf{B}||_2 = ||\mathbf{Q}||_2$. Hence we work with $\mathbf{Q}$ instead of $\mathbf{B}$.

Using Lemma 6.12 from (Wainwright, 2019), we have

$$\mathbb{P}\left[||\mathbf{Q} - \mathbb{E}\left[\mathbf{Q}\right]||_2 \geq \delta\right] \leq 2\mathtt{tr}\left(\Psi_{\mathbf{Q}}(\lambda)\right)e^{-\lambda\delta} \tag{46}$$

where $\Psi_{\mathbf{Q}}$ is the moment generating function of a random matrix $\mathbf{Q}$ and can be seen as a mapping $\Psi_{\mathbf{Q}} : \mathbb{R} \to \mathcal{S}^{d\times d}$ defined as:

$$\Psi_{\mathbf{Q}}(\lambda) = \mathbb{E}\left[e^{\lambda\mathbf{Q}}\right] = \sum_{k=0}^{\infty}\frac{\lambda^k}{k!}\mathbb{E}\left[\mathbf{Q}^k\right]$$

Therefore, we have to compute the moment generating function $\Psi_{\mathbf{Q}}(\lambda)$ or compute the bound for $\mathtt{tr}\left(\Psi_{\mathbf{Q}}(\lambda)\right)$ in Eq (46). To do that, we need to study the distribution of $\mathbf{B}$. Any entry $(i, j)$ of $\mathbf{B}$ can be expressed as the sum of products of pairs of sub-Gaussian random variables:

$$\mathbf{B}_{ij} = \frac{1}{n}\sum_{k=1}^{n}\mathbf{X}_{kj}^{\star}\mathbf{\Delta}\left(\mathbf{X}_{ki}^{\star}\right)$$

Since $\frac{\mathbf{E}_{ki}}{r\sqrt{\mathbf{\Sigma}_{ii}^{\Delta}}}$ and $\frac{\mathbf{X}_{kj}^{\star}}{\sigma\sqrt{\mathbf{\Sigma}_{jj}}}$ are zero-mean sub-Gaussian random variables with variance proxy 1, their product is a sub-exponential random variable with parameter $\left(8\sqrt{2}, 4\right)$ by using Lemma 28. Further, we define $q_{ij} = r\sigma\sqrt{\mathbf{\Sigma}_{ii}^{\Delta}}\sqrt{\mathbf{\Sigma}_{jj}}$ by using properties of sub-exponential distributions:

$$\mathbf{X}_{kj}^{\star}\mathbf{\Delta}\left(\mathbf{X}_{ki}^{\star}\right) \sim SE\left(8\sqrt{2}q_{ij}, 4q_{ij}\right)$$

$$\sum_{k=1}^{n}\mathbf{X}_{kj}^{\star}\mathbf{\Delta}\left(\mathbf{X}_{ki}^{\star}\right) \sim SE\left(8\sqrt{2}q_{ij}\sqrt{n}, 4q_{ij}\right)$$

$$\frac{1}{n}\sum_{k=1}^{n}\mathbf{X}_{kj}^{\star}\mathbf{\Delta}\left(\mathbf{X}_{ki}^{\star}\right) \sim SE\left(\frac{8\sqrt{2}q_{ij}}{\sqrt{n}}, \frac{4q_{ij}}{n}\right)$$

$$\frac{1}{n}\sum_{k=1}^{n}\mathbf{X}_{kj}^{\star}\mathbf{\Delta}\left(\mathbf{X}_{ki}^{\star}\right) \sim SE\left(\frac{8\sqrt{2}q_i}{\sqrt{n}}, \frac{4q_i}{n}\right)$$

where $q_i = r\sigma\sqrt{\mathbf{\Sigma}_{ii}^{\Delta}}\left(\max_{j\in\mathcal{S}}\mathbf{\Sigma}_{jj}\right)^{1/2}$. Therefore $\mathbf{Q}$ follows sub-exponential distribution with parameter $(\mathbf{V}_1, 4q)$, where $q = r\sigma\left(\max_{i\in\mathcal{S}}\mathbf{\Sigma}_{ii}^{\Delta}\right)^{1/2}\left(\max_{j\in\mathcal{S}}\mathbf{\Sigma}_{jj}\right)^{1/2}$, and $\mathbf{V}_1$ is a matrix of dimension $k \times k$:

$$\mathbf{V}_1 = \frac{128r^2\sigma^2\max_{j\in\mathcal{S}}\mathbf{\Sigma}_{jj}}{n}\begin{bmatrix}\mathbf{\Sigma}_{11}^{\Delta} & \mathbf{\Sigma}_{11}^{\Delta} & \dots & \mathbf{\Sigma}_{11}^{\Delta}\\ \mathbf{\Sigma}_{22}^{\Delta} & \mathbf{\Sigma}_{22}^{\Delta} & \dots & \mathbf{\Sigma}_{22}^{\Delta}\\ \vdots & \vdots & \ddots & \vdots\\ \mathbf{\Sigma}_{kk}^{\Delta} & \mathbf{\Sigma}_{kk}^{\Delta} & \dots & \mathbf{\Sigma}_{kk}^{\Delta}\end{bmatrix} \tag{47}$$

Further, it is easy to observe that the random matrix $\mathbf{Q}$ is sub-exponential with parameter $(\mathbf{V}, \frac{4q}{n})$, where $\mathbf{V}$ is described as

$$\mathbf{V} = \begin{bmatrix}\mathbf{0} & \mathbf{V}_1\\ \mathbf{V}_1^{\intercal} & \mathbf{0}\end{bmatrix} \tag{48}$$

The moment generating function $\Psi_{\mathbf{Q}}(\lambda)$ can be expressed as:

$$\Psi_{\mathbf{Q}}(\lambda) \preccurlyeq e^{\frac{\lambda^2\mathbf{V}}{2}}$$

Substituting the above in Eq. (46) and by replacing $\delta$ with $n\delta$, we get:

$$\mathbb{P}\left[||\mathbf{Q} - \mathbb{E}\left[\mathbf{Q}\right]||_2 \geq n\delta\right] \leq 2\mathtt{tr}\left(e^{\frac{\lambda^2 \mathbf{V}}{2}}\right)e^{-n\lambda\delta}$$

$$\mathbb{P}\left[\left|\left|\frac{1}{n}\mathbf{Q} - \mathbb{E}\left[\mathbf{Q}\right]\right|\right|_2 \geq \delta\right] \leq 2\mathtt{tr}\left(\sum_{i=0}^{\infty}\frac{\lambda^i}{2^i i!}\mathbf{V}^i\right)e^{-n\lambda\delta}$$

$$= 2\sum_{i=0}^{\infty}\mathtt{tr}\left(\frac{\lambda^i}{2^i i!}\left(\mathbf{D}\right)^i\right)e^{-n\lambda\delta}$$

$$= 2\mathtt{tr}\left(\sum_{i=0}^{\infty}\frac{\lambda^i}{2^i i!}\left(\mathbf{D}\right)^i\right)e^{-n\lambda\delta}$$

$$= 2e^{-n\lambda\delta}\sum_{i=1}^{2k}e^{\frac{\lambda^2}{2}d_i}$$

The next step is to compute the eigenvalues of the matrix $\mathbf{V}$ which is done in Lemma 22. It can be easily observed that $\mathbf{V}$ has only two non-zero eigenvalues equal to $\frac{c_2}{n}$, where

$$c_2 = 128r^2\sigma^2 \max_{j\in\mathcal{S}}\mathbf{\Sigma}_{jj}\sqrt{k\sum_{i\in\mathcal{S}}\left(\mathbf{\Sigma}_{ii}^{\mathbf{\Delta}}\right)^2} \tag{49}$$

If we use all the zero eigenvalues of $\mathbf{V}$ to compute $\sum_{i=1}^{2k}e^{\frac{\lambda^2}{2}d_i} = 2\exp\left\{n\frac{\lambda^2}{2}c_2\right\} + 2k - 2$, this would lead to ultimately non-optimal bounds. Hence the trick here is that the matrix $\mathbf{V}$ can be expressed as $\mathbf{V} = \mathbf{U}\mathbf{D}\mathbf{U}^{\mathsf{T}}$, where $\mathbf{U}$ is a $2k \times 2$ matrix instead of $2k \times 2k$ because we know $(2k - 2)$ eigenvalues of $\mathbf{V}$ are zero. If we use the first two columns of $\mathbf{U}$, then $\sum_{i=1}^{2k}e^{\frac{\lambda^2}{2}d_i} = 2\exp\left\{n\frac{\lambda^2}{2}c_2\right\}$. Substituting this in Eq. (46):

$$\mathbb{P}\left[\left|\left|\frac{1}{n}\mathbf{X}_{\mathcal{S}}^{\star\mathsf{T}}\mathbf{\Delta}\left(\mathbf{X}_{\mathcal{S}}^{\star}\right) - \mathbf{\Sigma}_{\mathcal{SS}}^{\mathbf{\Delta}\mathbf{x}^{\star}}\right|\right|_2 \geq \delta\right] \leq 2e^{-\lambda\delta} \times 2\exp\left\{\frac{\lambda^2 c_2}{2n}\right\} \qquad \forall \lambda < \frac{n}{4q}$$

Substituting the optimal $\lambda = \frac{n\delta}{c_2}$, we get:

$$\mathbb{P}\left[\left|\left|\frac{1}{n}\mathbf{X}_{\mathcal{S}}^{\star\mathsf{T}}\mathbf{\Delta}\left(\mathbf{X}_{\mathcal{S}}^{\star}\right) - \mathbf{\Sigma}_{\mathcal{SS}}^{\mathbf{\Delta}\mathbf{x}^{\star}}\right|\right|_2 \geq \delta\right] \leq 4e^{\frac{-n\delta^2}{2c_2}} = 4\exp\left\{\frac{-n\delta^2}{256r^2\sigma^2 \max_{j\in\mathcal{S}}\mathbf{\Sigma}_{jj}\sqrt{k\sum_{i\in\mathcal{S}}\left(\mathbf{\Sigma}_{ii}^{\mathbf{\Delta}}\right)^2}}\right\} \tag{50}$$

for $0 < \delta < 32\sqrt{k}r\sigma \max_{j\in\mathcal{S}}\mathbf{\Sigma}_{jj}\sqrt{\sum_{i\in\mathcal{S}}\left(\mathbf{\Sigma}_{ii}^{\mathbf{\Delta}}\right)^2}$. Hence, a slightly simplified version of $\delta$ can be $0 < \delta < 32\sqrt{k}r\sigma\sqrt{\max_{i\in\mathcal{S}}\mathbf{\Sigma}_{ii}^{\mathbf{\Delta}}}\sqrt{\max_{j\in\mathcal{S}}\mathbf{\Sigma}_{jj}}$.

$\square$

This lemma helps us to work with a symmetric matrix ($\mathbf{M}$) instead of non-symmetric matrix ($\mathbf{B}$).

**Lemma 20.** *For matrix* $\mathbf{B} \in \mathbb{R}^{k\times k}$, *we claim* $||\mathbf{B}||_2 = ||\mathbf{M}||_2$, *where* $\mathbf{M} = \begin{bmatrix} \mathbf{0}_{k\times k} & \mathbf{B} \\ \mathbf{B}^{\mathsf{T}} & \mathbf{0}_{k\times k} \end{bmatrix}$.

*Proof.* Using $\mathbf{M}$ defined as above, $\mathbf{M}^2$ can be computed as:

$$\mathbf{M}^2 = \begin{bmatrix} \mathbf{B}\mathbf{B}^{\mathsf{T}} & \mathbf{0}_{k\times k} \\ \mathbf{0}_{k\times k} & \mathbf{B}^{\mathsf{T}}\mathbf{B} \end{bmatrix}$$

The spectral norm of $\mathbf{M}^2$ can be computed as:

$$\left|\left|\mathbf{M}^2\right|\right|_2 = \max\left\{\lambda_{\max}\left(\mathbf{B}\mathbf{B}^{\mathsf{T}}\right), \lambda_{\max}\left(\mathbf{B}^{\mathsf{T}}\mathbf{B}\right)\right\}$$

From basic linear algebra properties, it is easy to observe that eigenvalues of $\mathbf{BB^\intercal}$ and $\mathbf{B^\intercal B}$ are the same:

$$\mathbf{BB^\intercal x} = \lambda \mathbf{x}$$
$$\mathbf{B^\intercal BB^\intercal x} = \lambda \mathbf{B^\intercal x}$$
$$\mathbf{B^\intercal B}\left(\mathbf{B^\intercal x}\right) = \lambda\left(\mathbf{B^\intercal x}\right)$$
$$\mathbf{B^\intercal By} = \lambda \mathbf{y}$$

Using the above, we can claim,

$$\left\|\mathbf{M}^2\right\|_2 = \lambda_{\max}\left(\mathbf{BB^\intercal}\right)$$

We also know that $\|\mathbf{M}\|_2^2 = \left\|\mathbf{M}^2\right\|_2$. Therefore $\|\mathbf{M}\|_2 = \sqrt{\lambda_{\max}\left(\mathbf{BB^\intercal}\right)} = \|\mathbf{B}\|_2$. $\qquad\square$

### D.6 Simplification of $\hat{\mathbf{z}}_{\mathcal{S}^c{}_{t_2}}$

$$\hat{\mathbf{z}}_{\mathcal{S}^c{}_{t_2}} = \mathbf{X}_{\mathcal{S}^c}^\intercal\left(\mathbf{P}/\lambda n\right)\left(\mathbf{e} - \mathbf{\Delta}\left(\mathbf{X}_{\mathcal{S}}^\star\right)\mathbf{w}_{\mathcal{S}}^\star\right), \qquad \text{where} \qquad \mathbf{P} = \left(\mathbf{I}_n - \mathbf{X}_{\mathcal{S}}\left(\mathbf{X}_{\mathcal{S}}^\intercal\mathbf{X}_{\mathcal{S}}\right)^{-1}\mathbf{X}_{\mathcal{S}}^\intercal\right) \tag{51}$$

Using triangle inequality and sub-multiplicative property of norms:

$$\left\|\hat{\mathbf{z}}_{\mathcal{S}^c{}_{t_2}}\right\|_\infty \leq \frac{1}{\lambda}\left\|\frac{1}{n}\mathbf{X}_{\mathcal{S}^c}^\intercal\mathbf{Pe}\right\|_\infty + \frac{1}{\lambda}\left\|\frac{1}{n}\mathbf{X}_{\mathcal{S}^c}^\intercal\mathbf{\Delta}\left(\mathbf{X}_{\mathcal{S}}^\star\right)\mathbf{w}_{\mathcal{S}}^\star\right\|_\infty + \frac{1}{\lambda}\left\|\mathbf{X}_{\mathcal{S}^c}^\intercal\mathbf{X}_{\mathcal{S}}\left(\mathbf{X}_{\mathcal{S}}^\intercal\mathbf{X}_{\mathcal{S}}\right)^{-1}\right\|_\infty\left\|\frac{1}{n}\mathbf{X}_{\mathcal{S}}^\intercal\mathbf{\Delta}\left(\mathbf{X}_{\mathcal{S}}^\star\right)\mathbf{w}_{\mathcal{S}}^\star\right\|_\infty$$

Further using the bound for $\left\|\mathbf{X}_{\mathcal{S}^c}^\intercal\mathbf{X}_{\mathcal{S}}\left(\mathbf{X}_{\mathcal{S}}^\intercal\mathbf{X}_{\mathcal{S}}\right)^{-1}\right\|_\infty$ derived in Section 3.1.1 or Eq. (12):

$$\left\|\hat{\mathbf{z}}_{\mathcal{S}^c{}_{t_2}}\right\|_\infty \leq \frac{1}{\lambda}\left\|\frac{1}{n}\mathbf{X}_{\mathcal{S}^c}^\intercal\mathbf{Pe}\right\|_\infty + \frac{1}{\lambda}\left\|\frac{1}{n}\mathbf{X}_{\mathcal{S}^c}^\intercal\mathbf{\Delta}\left(\mathbf{X}_{\mathcal{S}}^\star\right)\mathbf{w}_{\mathcal{S}}^\star\right\|_\infty + \frac{1}{\lambda}\left(1 - \frac{3\gamma}{4}\right)\left\|\frac{1}{n}\mathbf{X}_{\mathcal{S}}^\intercal\mathbf{\Delta}\left(\mathbf{X}_{\mathcal{S}}^\star\right)\mathbf{w}_{\mathcal{S}}^\star\right\|_\infty \tag{52}$$

### D.7 Proof of Lemma 15

**Lemma 15.** If the regularization parameter $\lambda = \lambda_1 \geq \frac{8q_1\sigma_e}{\gamma}\sqrt{\frac{4\log(p)}{n}}$, where constant $q_1^2 = 3\left(C_{\max} + 2F_{\max} + D_{\max}\right)$, then $\left\|\frac{\mathbf{X}_{\mathcal{S}^c}^\intercal\mathbf{Pe}}{n\lambda}\right\|_\infty \leq \frac{\gamma}{8}$ with probability of at least $1 - \mathcal{O}\left(\frac{1}{p}\right)$.

*Proof.* Consider the random vector of dimension $(p - k)$:

$$\mathbf{t}_1 = \frac{\mathbf{X}_{\mathcal{S}^c}^\intercal\mathbf{Pe}}{n\lambda} \tag{53}$$

whose each entry is zero-mean sub-Gaussian conditioned on $\mathbf{X}$. The variance parameter for each entry is given by:

$$\sigma_{t_1}^2 = \frac{1}{\lambda^2 n^2}\left\|\mathbf{X}_{\mathcal{S}^c}^\intercal\mathbf{P}\mathbb{E}\left[\mathbf{ee}_y^\intercal\right]\mathbf{PX}_{\mathcal{S}^c}\right\|_2 = \frac{\sigma_e^2}{\lambda^2 n}\left\|\frac{\mathbf{X}_{\mathcal{S}^c}^\intercal\mathbf{X}_{\mathcal{S}^c}}{n}\right\|_2$$
$$\leq \frac{\sigma_e^2}{\lambda^2 n}\left(\left\|\mathbf{\Sigma}_{\mathcal{S}^c\mathcal{S}^c}^{\mathbf{x}}\right\|_2 + \left\|\frac{\mathbf{X}_{\mathcal{S}^c}^\intercal\mathbf{X}_{\mathcal{S}^c}}{n} - \mathbf{\Sigma}_{\mathcal{S}^c\mathcal{S}^c}^{\mathbf{x}}\right\|_2\right)$$
$$\leq \frac{3\sigma_e^2}{2\lambda^2 n}\left(C_{\max} + 2F_{\max} + D_{\max}\right) \tag{54}$$

where we have used Lemma 21 in the first step. In the last step, we decompose $\left\|\frac{\mathbf{X}_{\mathcal{S}^c}^\intercal\mathbf{X}_{\mathcal{S}^c}}{n} - \mathbf{\Sigma}_{\mathcal{S}^c\mathcal{S}^c}^{\mathbf{x}}\right\|_2$ as done in Eq. (40) and further use Eq. (66) and Theorem 14 to claim the resulting bound with high probability of at least $1 - \mathcal{O}\left(\frac{1}{p}\right)$, if $n = k\log(p)$.

Further we use union-bound along with sub-Gaussian tail bounds to claim:

$$\mathbb{P}\left[\|\mathbf{t}_1\|_\infty \geq \delta\right] \leq 2\exp\left\{\frac{-\delta^2}{2\sigma_{t_1}^2} + \log(p-k)\right\} \tag{55}$$

Substituting $\delta = \frac{\gamma}{16}$, we can claim the above state with high probability of at least $1 - \mathcal{O}\left(\frac{1}{p}\right)$ if

$$\lambda \geq \frac{\sigma_e}{\gamma}\sqrt{3\left(C_{\max} + 2F_{\max} + D_{\max}\right)}\sqrt{\frac{2\log(p)}{n}} \tag{56}$$

which completes the proof. $\qquad\square$

The following lemma helps us to bound the spectral norm of $\mathbf{P}$.

**Lemma 21.** $\mathbf{P}$ *defined in Eq.* (6) *is a projection matrix and hence* $\|\mathbf{P}\|_2^2 = 1$.

*Proof.* We use the fact that $\mathbf{P}$ is a projection matrix iff $\mathbf{I} - \mathbf{P}$ is a projection matrix. Hence we focus only on $\mathbf{T} = \mathbf{I} - \mathbf{P} = \mathbf{X}_{\mathcal{S}}\left(\mathbf{X}_{\mathcal{S}}^\intercal\mathbf{X}_{\mathcal{S}}\right)^{-1}\mathbf{X}_{\mathcal{S}}^\intercal$

$$\mathbf{T} = \mathbf{X}_{\mathcal{S}}\left(\mathbf{X}_{\mathcal{S}}^\intercal\mathbf{X}_{\mathcal{S}}\right)^{-1}\mathbf{X}_{\mathcal{S}}^\intercal$$
$$\mathbf{T}^2 = \mathbf{X}_{\mathcal{S}}\left(\mathbf{X}_{\mathcal{S}}^\intercal\mathbf{X}_{\mathcal{S}}\right)^{-1}\mathbf{X}_{\mathcal{S}}^\intercal\mathbf{X}_{\mathcal{S}}\left(\mathbf{X}_{\mathcal{S}}^\intercal\mathbf{X}_{\mathcal{S}}\right)^{-1}\mathbf{X}_{\mathcal{S}}^\intercal = \mathbf{X}_{\mathcal{S}}\left(\mathbf{X}_{\mathcal{S}}^\intercal\mathbf{X}_{\mathcal{S}}\right)^{-1}\mathbf{X}_{\mathcal{S}}^\intercal = \mathbf{T}$$

Hence $\mathbf{P}$ defined in Eq. (6) is a valid projection matrix. $\qquad\square$

### D.8 Proof of Lemma 16

**Lemma 16.** If $\lambda = \lambda_2 \geq \frac{16}{\gamma}\max\left\{\left\|\mathbf{\Sigma}_{\mathcal{SS}}^{\mathbf{x}^\star\mathbf{\Delta}}\mathbf{w}_{\mathcal{S}}^\star + \mathbf{\Sigma}_{\mathcal{SS}}^{\mathbf{\Delta}}\mathbf{w}_{\mathcal{S}}^\star\right\|_\infty, q_2\sqrt{\frac{4\log(p)}{n}}\right\}$, then

$$\mathbb{P}\left[\left\|\frac{\mathbf{X}_{\mathcal{S}}^\intercal\mathbf{\Delta}\left(\mathbf{X}_{\mathcal{S}}^\star\right)\mathbf{w}_{\mathcal{S}}^\star}{n\lambda}\right\|_\infty \leq \frac{\gamma}{8}\right] \geq 1 - \mathcal{O}\left(\frac{1}{p}\right)$$

where $q_2 = r\sqrt{\mathbf{w}_{\mathcal{S}}^{\star\intercal}\mathbf{\Sigma}_{\mathcal{SS}}^{\mathbf{\Delta}}\mathbf{w}_{\mathcal{S}}^\star}\max\limits_{i\in\mathcal{S}^c}\left(\sigma\sqrt{\mathbf{\Sigma}_{ii}^{\mathbf{x}}} + r\sqrt{\mathbf{\Sigma}_{ii}^{\mathbf{\Delta}}}\right)$.

*Proof.* Consider $\mathbf{t}_2 = \frac{1}{n}\mathbf{X}_{\mathcal{S}^c}^\intercal\mathbf{\Delta}\left(\mathbf{X}_{\mathcal{S}}^\star\right)\mathbf{w}_{\mathcal{S}}^\star$ which is a $(p-k)\times 1$ random vector whose $i^{th}$ entry can be expressed as the mean of $n$ samples:

$$\mathbf{t}_{2i} = \frac{1}{n}\sum_{j=1}^n\mathbf{x}_i^{(j)}\left(\sum_{l\in\mathcal{S}}\mathbf{\Delta}\left(\mathbf{x}_l^{\star(j)}\right)\mathbf{w}_l^\star\right)$$

$$\mathbb{E}\left[\mathbf{t}_{2i}\right] = \frac{1}{n}\sum_{j=1}^n\sum_{l\in\mathcal{S}}\mathbb{E}\left[\mathbf{x}_i^{\star(j)}\mathbf{\Delta}\left(\mathbf{x}_l^{\star(j)}\right)\mathbf{w}_l^\star\right] + \frac{1}{n}\sum_{j=1}^n\sum_{l\in\mathcal{S}}\mathbb{E}\left[\mathbf{\Delta}\left(\mathbf{x}_i^{\star(j)}\right)\mathbf{\Delta}\left(\mathbf{x}_l^{\star(j)}\right)\mathbf{w}_l^\star\right]$$

$$= \mathbf{\Sigma}_{i\mathcal{S}}^{\mathbf{x}^\star\mathbf{\Delta}}\mathbf{w}_{\mathcal{S}}^\star + \mathbf{\Sigma}_{i\mathcal{S}}^{\mathbf{\Delta}}\mathbf{w}_{\mathcal{S}}^\star$$

where $i\in\mathcal{S}^c$. Since $\frac{\mathbf{x}_i^{(j)}}{\left(\sigma\sqrt{\mathbf{\Sigma}_{ii}^{\mathbf{x}}}+r\sqrt{\mathbf{\Sigma}_{ii}^{\mathbf{\Delta}}}\right)}$ and $\frac{\sum\limits_{l\in\mathcal{S}}\mathbf{\Delta}\left(\mathbf{x}_l^{\star(j)}\right)\mathbf{w}_l^\star}{\sigma_t}$, where $\sigma_t = r\sqrt{\mathbf{w}_{\mathcal{S}}^{\star\intercal}\mathbf{\Sigma}_{\mathcal{SS}}^{\mathbf{\Delta}}\mathbf{w}_{\mathcal{S}}^\star}$ are zero-mean sub-Gaussian random variables with variance proxy 1, their product is a sub-exponential random variable with parameter $\left(8\sqrt{2}, 4\right)$ by using Lemma 28. Therefore the sample mean will also be a sub-exponential random variable with the following parameters:

$$\frac{1}{n\lambda}\sum_{j=1}^n\mathbf{x}_i^{(j)}\sum_{l\in\mathcal{S}}\mathbf{\Delta}\left(\mathbf{x}_l^{\star(j)}\right)\mathbf{w}_l^\star \sim SE\left(\frac{8\sqrt{2}c_i}{\lambda\sqrt{n}}, \frac{4c_i}{\lambda n}\right)$$

where $c_i = \left( \sigma \sqrt{\boldsymbol{\Sigma}_{ii}^{\mathbf{x}}} + r \sqrt{\boldsymbol{\Sigma}_{ii}^{\boldsymbol{\Delta}}} \right) \sigma_t$. By using sub-exponential tail bounds and union bound, we further claim:

$$\mathbb{P}\left[ \frac{1}{\lambda} \left\| \mathbf{t}_2 - \mathbb{E}[\mathbf{t}_2] \right\|_\infty > \delta \right] \leq 2 \exp\left( -\frac{n\delta^2 \lambda^2}{2q_2^2} + \log(p-k) \right)$$

for $0 < \delta\lambda \leq 32q_2$, where $q_2 = \max\limits_{i \in \mathcal{S}^c} c_i$. Substituting $\delta = \frac{\gamma}{16}$ in the above equation, we arrive at:

$$\frac{1}{\lambda} \left\| \mathbf{t}_2 - \mathbb{E}[\mathbf{t}_2] \right\|_\infty \leq \frac{\gamma}{16} \tag{57}$$

with high probability of at least $1 - \mathcal{O}\left( \frac{1}{p} \right)$, if the regularization parameter satisfies:

$$\lambda \geq \frac{16q_2}{\gamma} \sqrt{\frac{2\log(p)}{n}} \tag{58}$$

Using triangle inequality, we can claim:

$$\frac{\left\| \mathbf{t}_2 \right\|_\infty}{\lambda} \leq \frac{1}{\lambda} \left\| \mathbf{t}_2 - \mathbb{E}[\mathbf{t}_2] \right\|_\infty + \frac{\left\| \mathbb{E}[\mathbf{t}_2] \right\|_\infty}{\lambda} \leq \frac{\gamma}{16} + \frac{\gamma}{16} = \frac{\gamma}{8}$$

with a high probability if the regularization parameter satisfies:

$$\lambda \geq \frac{16 \left\| \mathbb{E}[\mathbf{z}_1] \right\|_\infty}{\gamma} = \frac{16}{\gamma} \left( \left\| \boldsymbol{\Sigma}_{\mathcal{S}^c \mathcal{S}}^{\mathbf{x}^\star \boldsymbol{\Delta}} \mathbf{w}_{\mathcal{S}}^\star + \boldsymbol{\Sigma}_{\mathcal{S}^c \mathcal{S}}^{\boldsymbol{\Delta}} \mathbf{w}_{\mathcal{S}}^\star \right\|_\infty \right) \tag{59}$$

Combining Eq. (58) and Eq. (59) for the regularization parameter:

$$\lambda \geq \frac{16}{\gamma} \max\left\{ \left\| \boldsymbol{\Sigma}_{\mathcal{S}^c \mathcal{S}}^{\mathbf{x}^\star \boldsymbol{\Delta}} \mathbf{w}_{\mathcal{S}}^\star + \boldsymbol{\Sigma}_{\mathcal{S}^c \mathcal{S}}^{\boldsymbol{\Delta}} \mathbf{w}_{\mathcal{S}}^\star \right\|_\infty, q_2 \sqrt{\frac{4\log(p)}{n}} \right\}$$

$\square$

### D.9 Proof of Lemma 19

**Lemma 19.** If Assumption 7 holds and $n = \Omega\left( k\log(p) \right)$, then we have $\lambda_{\min}\left( \frac{1}{n} \mathbf{X}_{\mathcal{S}}^\mathsf{T} \mathbf{X}_{\mathcal{S}} \right) \geq \frac{C_{\min} + 2F_{\min} + D_{\min}}{2} > 0$ with probability at least $1 - \mathcal{O}\left( \frac{1}{p} \right)$.

*Proof.* The minimum eigenvalue of $\frac{1}{n} \mathbf{X}_{\mathcal{S}}^\mathsf{T} \mathbf{X}_{\mathcal{S}}$ can be expressed as:

$$\lambda_{\min}\left( \frac{1}{n} \mathbf{X}_{\mathcal{S}}^\mathsf{T} \mathbf{X}_{\mathcal{S}} \right) = \lambda_{\min}\left( \frac{1}{n} \mathbf{X}_{\mathcal{S}}^{\star\mathsf{T}} \mathbf{X}_{\mathcal{S}}^\star + \frac{1}{n} \boldsymbol{\Delta}\left( \mathbf{X}_{\mathcal{S}}^{\star\mathsf{T}} \right) \mathbf{X}_{\mathcal{S}}^\star + \frac{1}{n} \mathbf{X}_{\mathcal{S}}^{\star\mathsf{T}} \boldsymbol{\Delta}\left( \mathbf{X}_{\mathcal{S}}^\star \right) + \frac{1}{n} \boldsymbol{\Delta}\left( \mathbf{X}_{\mathcal{S}}^{\star\mathsf{T}} \right) \boldsymbol{\Delta}\left( \mathbf{X}_{\mathcal{S}}^\star \right) \right)$$

$$\geq \lambda_{\min}\left( \frac{1}{n} \mathbf{X}_{\mathcal{S}}^{\star\mathsf{T}} \mathbf{X}_{\mathcal{S}}^\star \right) + \lambda_{\min}\left( \frac{1}{n} \boldsymbol{\Delta}\left( \mathbf{X}_{\mathcal{S}}^{\star\mathsf{T}} \right) \mathbf{X}_{\mathcal{S}}^\star + \frac{1}{n} \mathbf{X}_{\mathcal{S}}^{\star\mathsf{T}} \boldsymbol{\Delta}\left( \mathbf{X}_{\mathcal{S}}^\star \right) \right) + \lambda_{\min}\left( \frac{1}{n} \boldsymbol{\Delta}\left( \mathbf{X}_{\mathcal{S}}^{\star\mathsf{T}} \right) \boldsymbol{\Delta}\left( \mathbf{X}_{\mathcal{S}}^\star \right) \right) \tag{60}$$

We need to further derive lower bounds for $\lambda_{\min}\left( \frac{1}{n} \mathbf{X}_{\mathcal{S}}^{\star\mathsf{T}} \mathbf{X}_{\mathcal{S}}^\star \right)$ and $\lambda_{\min}\left( \frac{1}{n} \boldsymbol{\Delta}\left( \mathbf{X}_{\mathcal{S}}^{\star\mathsf{T}} \right) \boldsymbol{\Delta}\left( \mathbf{X}_{\mathcal{S}}^\star \right) \right)$. Substituting $\delta = \frac{1}{2}$ in Eq. (63) and Eq. (65) of Lemma 25, we can claim $\lambda_{\min}\left( \frac{1}{n} \mathbf{X}_{\mathcal{S}}^{\star\mathsf{T}} \mathbf{X}_{\mathcal{S}}^\star \right) \geq \frac{C_{\min}}{2}$ and $\lambda_{\min}\left( \frac{1}{n} \boldsymbol{\Delta}\left( \mathbf{X}_{\mathcal{S}}^{\star\mathsf{T}} \right) \boldsymbol{\Delta}\left( \mathbf{X}_{\mathcal{S}}^\star \right) \right) \geq \frac{D_{\min}}{2}$ with probability $\left( 1 - 2\exp\left\{ \frac{-c_1 C_{\min}^2 n}{4} + k \right\} \right)$ and $\left( 1 - 2\exp\left\{ \frac{-c_2 D_{\min}^2 n}{4} + k \right\} \right)$ respectively. Using this information, we claim:

$$\lambda_{\min}\left( \frac{1}{n} \mathbf{X}_{\mathcal{S}}^\mathsf{T} \mathbf{X}_{\mathcal{S}} \right) \geq \frac{C_{\min}}{2} + \lambda_{\min}\left( \frac{1}{n} \boldsymbol{\Delta}\left( \mathbf{X}_{\mathcal{S}}^{\star\mathsf{T}} \right) \mathbf{X}_{\mathcal{S}}^\star + \frac{1}{n} \mathbf{X}_{\mathcal{S}}^{\star\mathsf{T}} \boldsymbol{\Delta}\left( \mathbf{X}_{\mathcal{S}}^\star \right) \right) + \frac{D_{\min}}{2}$$

For general square matrices $\mathbf{A}$ and $\mathbf{B}$, we have $\lambda_{\min}(\mathbf{A} + \mathbf{A}^\intercal) = \lambda_{\min}(\mathbf{B} + \mathbf{B}^\intercal + (\mathbf{A} - \mathbf{B}) + (\mathbf{A} - \mathbf{B})^\intercal) \geq \lambda_{\min}(\mathbf{B} + \mathbf{B}^\intercal) - \|(\mathbf{A} - \mathbf{B}) + (\mathbf{A} - \mathbf{B})^\intercal\|_2 \geq \lambda_{\min}(\mathbf{B} + \mathbf{B}^\intercal) - 2\|\mathbf{A} - \mathbf{B}\|_2$. Thus:

$$\lambda_{\min}\left(\frac{1}{n}\boldsymbol{\Delta}\left(\mathbf{X}_{\mathcal{S}}^{\star\intercal}\right)\mathbf{X}_{\mathcal{S}}^\star + \frac{1}{n}\mathbf{X}_{\mathcal{S}}^{\star\intercal}\boldsymbol{\Delta}\left(\mathbf{X}_{\mathcal{S}}^\star\right)\right) \geq \lambda_{\min}\left(\boldsymbol{\Sigma}_{\mathcal{S}\mathcal{S}}^{\boldsymbol{\Delta}\mathbf{x}^\star} + \boldsymbol{\Sigma}_{\mathcal{S}\mathcal{S}}^{\mathbf{x}^\star\boldsymbol{\Delta}}\right) - 2\left\|\frac{1}{n}\boldsymbol{\Delta}\left(\mathbf{X}_{\mathcal{S}}^{\star\intercal}\right)\mathbf{X}_{\mathcal{S}}^\star - \boldsymbol{\Sigma}_{\mathcal{S}\mathcal{S}}^{\boldsymbol{\Delta}\mathbf{x}^\star}\right\|_2$$

$$= 2\lambda_{\min}\left([\boldsymbol{\Sigma}^{\boldsymbol{\Delta}\mathbf{x}^\star} + \boldsymbol{\Sigma}^{\mathbf{x}^\star\boldsymbol{\Delta}}]_{\mathcal{S}\mathcal{S}}/2\right) - 2\left\|\frac{1}{n}\boldsymbol{\Delta}\left(\mathbf{X}_{\mathcal{S}}^{\star\intercal}\right)\mathbf{X}_{\mathcal{S}}^\star - \boldsymbol{\Sigma}_{\mathcal{S}\mathcal{S}}^{\boldsymbol{\Delta}\mathbf{x}^\star}\right\|_2$$

$$= 2F_{\min} - 2\left\|\frac{1}{n}\boldsymbol{\Delta}\left(\mathbf{X}_{\mathcal{S}}^{\star\intercal}\right)\mathbf{X}_{\mathcal{S}}^\star - \boldsymbol{\Sigma}_{\mathcal{S}\mathcal{S}}^{\boldsymbol{\Delta}\mathbf{x}^\star}\right\|_2$$

The next step is to bound $\left\|\frac{1}{n}\boldsymbol{\Delta}\left(\mathbf{X}_{\mathcal{S}}^{\star\intercal}\right)\mathbf{X}_{\mathcal{S}}^\star - \boldsymbol{\Sigma}_{\mathcal{S}\mathcal{S}}^{\boldsymbol{\Delta}\mathbf{x}^\star}\right\|_2$ which is done in Theorem 14. Substituting $\delta = \frac{F_{\min}}{2}$ in Eq. (9), we can claim the following with high probability

$$\lambda_{\min}\left(\frac{1}{n}\boldsymbol{\Delta}\left(\mathbf{X}_{\mathcal{S}}^{\star\intercal}\right)\mathbf{X}_{\mathcal{S}}^\star + \frac{1}{n}\mathbf{X}_{\mathcal{S}}^{\star\intercal}\boldsymbol{\Delta}\left(\mathbf{X}_{\mathcal{S}}^\star\right)\right) \geq 2F_{\min} - F_{\min} = F_{\min} \tag{61}$$

if $n = \Omega(k\log(p))$. Hence we claim

$$\lambda_{\min}\left(\frac{1}{n}\mathbf{X}_{\mathcal{S}}^\intercal\mathbf{X}_{\mathcal{S}}\right) \geq \frac{C_{\min} + 2F_{\min} + D_{\min}}{2} > 0$$

with probability $1 - \mathcal{O}\left(\frac{1}{p}\right)$ if $n = \Omega\left(k\log(p)\right)$.

$\square$

**Lemma 22.** *The two non-zero eigenvalues of the matrix $\mathbf{V}$ defined in Eq (48) are equal to $\frac{c_2}{n}$, where* $c_2 = 128r^2\sigma^2 \max_{j\in\mathcal{S}}\boldsymbol{\Sigma}_{jj}\sqrt{k\sum_{i\in\mathcal{S}}\left(\boldsymbol{\Sigma}_{ii}^{\boldsymbol{\Delta}}\right)^2}$. *The rest of the $2k-2$ eigenvalues are zero.*

*Proof.* We leave the multiplicative factor $\dfrac{128r^2\sigma^2 \max\limits_{j\in[p]}\boldsymbol{\Sigma}_{jj}}{n}$ aside and focus on the matrix structure now. Let $a_i = \boldsymbol{\Sigma}_{ii}^{\boldsymbol{\Delta}}$ for the ease of notation. Hence the transformed matrix $\mathbf{V}_1'$ has the following form:

$$\mathbf{V}_1' = \begin{bmatrix} a_1 & a_1 & \dots & a_1 \\ a_2 & a_2 & \dots & a_2 \\ \vdots & \vdots & \ddots & \vdots \\ a_k & a_k & \dots & a_k \end{bmatrix}$$

We use the idea used in Lemma 20 and compute the eigenvalues of $\mathbf{V}'^2$ instead of $\mathbf{V}'$ directly:

$$\mathbf{V}'^2 = \begin{bmatrix} \mathbf{V}_1'\mathbf{V}_1'^\intercal & \mathbf{0}_{k\times k} \\ \mathbf{0}_{k\times k} & \mathbf{V}_1'^\intercal\mathbf{V}_1' \end{bmatrix}$$

$$\mathbf{V}_1'\mathbf{V}_1'^\intercal = k\begin{bmatrix} a_1^2 & a_1a_2 & \dots & a_1a_k \\ a_1a_2 & a_2^2 & \dots & a_2a_k \\ \dots & \dots & \ddots & \vdots \\ a_ka_1 & a_ka_2 & \dots & a_k^2 \end{bmatrix}$$

To compute the eigenvalues of $\mathbf{V}'^2$, we focus on $\mathbf{V}_1'\mathbf{V}_1'^\intercal$ and $\mathbf{V}_1'^\intercal\mathbf{V}_1'$ separately. To compute the eigenvalues of $\mathbf{V}_1'\mathbf{V}_1'^\intercal$, we first determine its rank by using some elementary row operations: $R_i \to R_i - R_1\frac{a_i}{a_1}$ for $i \in [2, 3, \dots, k]$. The resulting matrix becomes:

$$k\begin{bmatrix} a_1^2 & a_1a_2 & \dots & a_1a_k \\ 0 & 0 & \dots & 0 \\ \vdots & \vdots & \ddots & \vdots \\ 0 & 0 & \dots & 0 \end{bmatrix}$$

Therefore, $\mathbf{V}_1^{'} \mathbf{V}_1^{'\mathsf{T}}$ is a rank 1 matrix and hence the one non-zero eigenvalue can be computed using the trace of the matrix, which is $\sqrt{k \sum_{i \in \mathcal{S}} a_i^2}$. By using Lemma 20, we can claim that the eigenvalues of $\mathbf{V}_1^{'} \mathbf{V}_1^{'\mathsf{T}}$ and $\mathbf{V}_1^{'\mathsf{T}} \mathbf{V}_1^{'}$ are the same, and hence the two non-zero eigenvalues of $\mathbf{V}^{'}$ can be derived as:

$$\lambda(\mathbf{V}^{'}) = \sqrt{k \sum_{i \in \mathcal{S}} a_i^2} = \sqrt{k \sum_{i \in \mathcal{S}} \left(\boldsymbol{\Sigma}_{ii}^{\boldsymbol{\Delta}}\right)^2}$$

Accounting for the scaling factor that was kept aside in the first step:

$$\lambda(\mathbf{V}) = \frac{128 r^2 \sigma^2 \max\limits_{j \in \mathcal{S}} \boldsymbol{\Sigma}_{jj}}{n} \sqrt{k \sum_{i \in \mathcal{S}} \left(\boldsymbol{\Sigma}_{ii}^{\boldsymbol{\Delta}}\right)^2}$$

$\square$

**Lemma 23.** *Let each entry of $\mathbf{X} \in \mathbb{R}^{k_1 \times k_2}$ be sub-exponentially distributed, denoted by $SE(\nu, \alpha)$, then for any $0 \le \delta \le k_2 \frac{\nu^2}{\alpha}$.*

$$\mathbb{P}\left[|||X - \mathbb{E}[\mathbf{X}]|||_\infty > \delta\right] \le 2 k_1 k_2 \exp\left\{-\frac{\delta^2}{2 k_2^2 \nu^2}\right\}.$$

*Proof.* We start with the use of basic norm inequalities and further use a union bound.

$$\mathbb{P}\left[|||X - \mathbb{E}[\mathbf{X}]|||_\infty > \delta\right] \le \mathbb{P}\left[k_2 \|\mathbf{X} - \mathbb{E}[\mathbf{X}]\|_\infty > \delta\right]$$

$$\le \mathbb{P}\left[(\forall i \in [k_1], j \in [k_2]) \quad |\mathbf{X}_{ij} - \mathbb{E}[\mathbf{X}_{ij}]| > \frac{\delta}{k_2}\right]$$

$$\le k_1 k_2 \mathbb{P}\left[|\mathbf{X}_{ij} - \mathbb{E}[\mathbf{X}_{ij}]| > \frac{\delta}{k_2}\right]$$

$$\le 2 k_1 k_2 \exp\left\{-\frac{\delta^2}{2 k_2^2 \nu^2}\right\}$$

for $0 \le \delta \le k_2 \frac{\nu^2}{\alpha}$, where we have used sub-exponential tail bounds in the last step. $\square$

**Lemma 24.** *If $\lambda = \lambda_3 \ge \frac{16}{\gamma}\left(1 - \frac{3\gamma}{4}\right) \max\left\{\left\|\boldsymbol{\Sigma}_{\mathcal{S}\mathcal{S}}^{\mathbf{x}^\star \boldsymbol{\Delta}} \mathbf{w}_{\mathcal{S}}^\star + \boldsymbol{\Sigma}_{\mathcal{S}\mathcal{S}}^{\boldsymbol{\Delta}} \mathbf{w}_{\mathcal{S}}^\star\right\|_\infty, q_3 \sqrt{\frac{4 \log(p)}{n}}\right\}$, then*

$$\mathbb{P}\left[\left\|\frac{\mathbf{X}_{\mathcal{S}}^\mathsf{T} \boldsymbol{\Delta}\left(\mathbf{X}_{\mathcal{S}}^\star\right) \mathbf{w}_{\mathcal{S}}^\star}{n\lambda}\right\|_\infty \le \frac{\gamma}{8} \frac{1}{\left(1 - \frac{3\gamma}{4}\right)}\right] \ge 1 - \mathcal{O}\left(\frac{1}{p}\right) \tag{62}$$

*where $q_3 = r \sqrt{\mathbf{w}_{\mathcal{S}}^{\star\mathsf{T}} \boldsymbol{\Sigma}_{\mathcal{S}\mathcal{S}}^{\boldsymbol{\Delta}} \mathbf{w}_{\mathcal{S}}^\star} \max\limits_{i \in \mathcal{S}}\left(\sigma \sqrt{\boldsymbol{\Sigma}_{ii}^{\mathbf{x}}} + r \sqrt{\boldsymbol{\Sigma}_{ii}^{\boldsymbol{\Delta}}}\right).$*

*Proof.* The proof of this lemma is analogous to proof of Lemma 16. We need to take union bound over $k$ terms only, as we are working with $\mathcal{S}$. Also, we substitute $\delta = \frac{\gamma}{16} \frac{1}{\left(1 - \frac{3\gamma}{4}\right)}$ in Eq. (57), which is the reason we see the scaling factor of $\left(1 - \frac{3\gamma}{4}\right)$. $\square$

**Lemma 25.** *If Assumption 7 holds, then for some $0 \le \delta < 1$, we have:*

$$\mathbb{P}\left[\lambda_{\min}\left(\frac{1}{n}\mathbf{X}_{\mathcal{S}}^{\star\mathsf{T}} \mathbf{X}_{\mathcal{S}}^\star\right) \le (1 - \delta) C_{\min}\right] \le 2 \exp\left\{-c_1 C_{\min}^2 \delta^2 n + k\right\} \tag{63}$$

$$\text{or equivalently} \quad \mathbb{P}\left[\left\|\left(\frac{1}{n}\mathbf{X}_{\mathcal{S}}^{\star\mathsf{T}} \mathbf{X}_{\mathcal{S}}^\star\right)^{-1}\right\|_2 \ge \frac{1}{(1 - \delta) C_{\min}}\right] \le 2 \exp\left\{-c_1 C_{\min}^2 \delta^2 n + k\right\} \tag{64}$$

$$\text{and independently,} \quad \mathbb{P}\left[\lambda_{\min}\left(\frac{1}{n}\boldsymbol{\Delta}\left(\mathbf{X}_{\mathcal{S}}^{\star\mathsf{T}}\right) \boldsymbol{\Delta}\left(\mathbf{X}_{\mathcal{S}}^\star\right)\right) \le (1 - \delta) D_{\min}\right] \le 2 \exp\left\{-c_2 D_{\min}^2 \delta^2 n + k\right\} \tag{65}$$

where $c_1$, $c_2$ are some positive constants. If $n = \Omega\left(k + \log(p)\right)$, the probability bound $1 - 2\exp\left\{-c_1 C_{\min}^2 \delta^2 n + k\right\}$ and $1 - 2\exp\left\{-c_1 D_{\min}^2 \delta^2 n + k\right\}$ simplify to $1 - \mathcal{O}\left(\frac{1}{p}\right)$.

*Proof.* Let $\mathbf{A} = \frac{1}{n}\mathbf{X}_{\mathcal{S}}^{\star\mathsf{T}}\mathbf{X}_{\mathcal{S}}^{\star}$. To derive an upper bound on the maximum eigenvalue of $\mathbf{A}^{-1}$, we derive a lower bound on the minimum eigenvalue of $\mathbf{A}$:

$$
\begin{aligned}
\lambda_{\min}\left(\mathbf{A}\right) &= \lambda_{\min}\left(\mathbf{A} - \boldsymbol{\Sigma}_{\mathcal{S}\mathcal{S}} + \boldsymbol{\Sigma}_{\mathcal{S}\mathcal{S}}\right) \\
&\geq \lambda_{\min}\left(\boldsymbol{\Sigma}_{\mathcal{S}\mathcal{S}}\right) - \max\left(\lambda_{\max}\left(\mathbf{A} - \boldsymbol{\Sigma}_{\mathcal{S}\mathcal{S}}\right), -\lambda_{\min}\left(\mathbf{A} - \boldsymbol{\Sigma}_{\mathcal{S}\mathcal{S}}\right)\right) \\
&= C_{\min} - \left|\left|\mathbf{A} - \boldsymbol{\Sigma}_{\mathcal{S}\mathcal{S}}\right|\right|_2
\end{aligned}
$$

Using Proposition 2.1 of (Vershynin, 2012), we can bound $\left|\left|\mathbf{A} - \boldsymbol{\Sigma}_{\mathcal{S}\mathcal{S}}\right|\right|_2$ as follows:

$$
\mathbb{P}\left[\left|\left|\mathbf{A} - \boldsymbol{\Sigma}_{\mathcal{S}\mathcal{S}}\right|\right|_2 \geq \epsilon\right] \leq 2\exp\left\{-c\epsilon^2 n + k\right\} \tag{66}
$$

where $c$ is a constant. Substituting $\epsilon = \delta C_{\min}$ in the above equation, we get

$$
\mathbb{P}\left[\left|\left|\mathbf{A} - \boldsymbol{\Sigma}_{\mathcal{S}\mathcal{S}}\right|\right|_2 \geq \delta C_{\min}\right] \leq 2\exp\left\{-c\delta^2 C_{\min}^2 n + k\right\}
$$

Hence, we can claim $\lambda_{\min}\left(\mathbf{A}\right) \geq (1 - \delta)C_{\min}$ with probability $1 - 2\exp\left\{-cC_{\min}^2\delta^2 n + k\right\}$. If $n > C(k + \log(p))$, then we claim $\lambda_{\min}\left(\mathbf{A}\right) \geq \frac{C_{\min}}{2}$ with probability $1 - \mathcal{O}\left(\frac{1}{p}\right)$. Therefore $\left|\left|\mathbf{A}^{-1}\right|\right|_2 \leq \frac{2}{C_{\min}}$.

The bound on $\lambda_{\min}\left(\frac{1}{n}\boldsymbol{\Delta}\left(\mathbf{X}_{\mathcal{S}}^{\star\mathsf{T}}\right)\boldsymbol{\Delta}\left(\mathbf{X}_{\mathcal{S}}^{\star}\right)\right)$ can be proved using the same approach. $\qquad\square$

**Lemma 26.** *For any $a, b \in \mathbb{R}$, fix $\epsilon > 0$. If we have $|a - b| \leq \epsilon \wedge |b| > 2\epsilon$, then $\mathrm{sign}(a) = \mathrm{sign}(b)$*

*Proof.* Consider the two cases for $|b| > 2\epsilon$

Case 1: if $b > 2\epsilon$ and $|a - b| \leq \epsilon$, then $a \geq \epsilon$. This implies $a$ and $b$ are both positive and have the same sign.

Case 2: if $b < -2\epsilon$ and $|a - b| \leq \epsilon$, then $a \leq -\epsilon$. This implies $a$ and $b$ are both negative and have the same sign. $\qquad\square$

**Lemma 27.** *Let $X \sim SG(0, \sigma_x)$ and $Y \sim SG(0, \sigma_y)$, then*

1. $X + Y \sim SG\left(0, \left(\sigma_x^2 + \sigma_y^2\right)^{1/2}\right)$ *if $X$ and $Y$ are mutually independent.*

2. $X + Y \sim SG\left(0, (\sigma_x + \sigma_y)\right)$ *if $X$ and $Y$ are dependent.*

*where $SG(\mu, \sigma_z)$ denotes a sub-Gaussian distribution with mean $\mu$ and parameter $\sigma_z$.*

*Proof.* We start with the easier case of $X$ and $Y$ being independent. We compute the moment generating function for $X + Y$:

$$
\begin{aligned}
\mathbb{E}\left[e^{\lambda(X+Y)}\right] &= \mathbb{E}\left[e^{\lambda X}e^{\lambda Y}\right] = \mathbb{E}\left[e^{\lambda X}\right]\mathbb{E}\left[e^{\lambda Y}\right] \\
&\leq \exp\left(\frac{\lambda^2\sigma_x^2}{2}\right)\exp\left(\frac{\lambda^2\sigma_y^2}{2}\right) = \exp\left(\frac{\lambda^2(\sigma_x^2 + \sigma_y^2)}{2}\right)
\end{aligned}
$$

which completes the proof for mutually independent random variables $X$ and $Y$.

Further proceeding to the general case and writing the moment generating function:

$$
\begin{aligned}
\mathbb{E}\left[e^{\lambda(X+Y)}\right] &= \mathbb{E}\left[e^{\lambda X}e^{\lambda Y}\right] \overset{(i)}{\leq} \left(\mathbb{E}\left[e^{\lambda p X}\right]\right)^{1/p}\left(\mathbb{E}\left[e^{\lambda q X}\right]\right)^{1/q} \\
&\leq \exp\left(\frac{\lambda^2\sigma_x^2 p^2}{2}\frac{1}{p}\right)\exp\left(\frac{\lambda^2\sigma_y^2 q^2}{2}\frac{1}{q}\right) = \exp\left(\frac{\lambda^2\left(\sigma_x^2 p + \sigma_y^2 q\right)}{2}\right)
\end{aligned} \tag{67}
$$

where (i) uses Hölder's inequality where $\frac{1}{p} + \frac{1}{q} = 1$. To upper bound the above, we optimize with respect to variable $p$ and solve:

$$\max f(p) = \max \left( \sigma_x^2 p + \sigma_y^2 q \right) = \max \left( \sigma_x^2 p + \sigma_y^2 \frac{p}{p-1} \right)$$

Taking the first order derivative:

$$\frac{df(p)}{dp} = \sigma_x^2 - \sigma_y^2 \frac{1}{(p-1)^2} = 0$$

which gives $p = 1 + \frac{\sigma_y}{\sigma_x}$, and therefore $q = 1 + \frac{\sigma_x}{\sigma_y}$. Substituting this in Eq. (67), we arrive at:

$$\mathbb{E}\left[ e^{\lambda(X+Y)} \right] \leq \exp\left( \frac{\lambda^2 \left( \sigma_x^2 + \sigma_x \sigma_y + \sigma_y^2 + \sigma_x \sigma_y \right)}{2} \right) = \exp\left( \frac{\lambda^2 \left( \sigma_x + \sigma_y \right)^2}{2} \right)$$

which completes the proof for the general case. $\qquad\square$

**Lemma 28.** *Let $X \sim SG(0,1)$ and $Y \sim SG(0,1)$, then*

1. *$XY \sim SE(4\sqrt{2}, 4)$ if $X$ and $Y$ are independent*

2. *$XY \sim SE(8\sqrt{2}, 4)$ if $X$ and $Y$ are dependent.*

*where $SG(\mu, \sigma_z)$ denotes a sub-Gaussian distribution with mean $\mu$ and parameter $\sigma_z$, and $SE(\nu, \alpha)$ denotes a sub-exponential distribution with parameters $\nu, \alpha$.*

*Proof.* We first start with the case of mutually independent $X$ and $Y$. Their product can be expressed as:

$$XY = \frac{(X+Y)^2 - (X-Y)^2}{4} \tag{68}$$

So, we derive the distribution of $(X+Y)^2$ and $(X-Y)^2$. We use Lemma 27 to derive the distribution for the sum of a pair of independent random variables

$$X + Y \sim SG(0, \sqrt{2})$$

Further, by scaling of sub-Gaussian random variables, we claim:

$$\frac{X+Y}{\sqrt{2}} \sim SG(0,1)$$

In the next step, we use Lemma 8 from (Barik & Honorio, 2023) to derive the distribution of the square of a sub-Gaussian random variable:

$$\left( \frac{X+Y}{\sqrt{2}} \right)^2 \sim SE(4\sqrt{2}, 4)$$

In a similar manner, we can claim the following for the difference of two sub-Gaussian random variables:

$$\left( \frac{X-Y}{\sqrt{2}} \right)^2 \sim SE(4\sqrt{2}, 4)$$

By scaling of sub-exponential random variables, we claim:

$$(X+Y)^2 \sim SE(8\sqrt{2}, 8)$$
$$(X-Y)^2 \sim SE(8\sqrt{2}, 8)$$

To derive the distribution of the sum of $(X + Y)^2$ and $(X - Y)^2$, we use Lemma 27 for dependent variables:

$$(X + Y)^2 - (X - Y)^2 \sim SE(16\sqrt{2}, 8)$$

Further, by scaling of sub-exponential random variables:

$$\frac{(X + Y)^2 - (X - Y)^2}{4} \sim SE(4\sqrt{2}, 2)$$

This completes the proof for the first claim of the lemma. Proceeding in a similar manner for the general case, we use Lemma 27 for dependent variables to claim the following:

$$X + Y \sim SG(0, 2)$$

Proceeding in a similar manner as done for the case of independent random variables, but now for dependent random variables, we arrive at:

$$\frac{(X + Y)^2 - (X - Y)^2}{4} \sim SE(8\sqrt{2}, 4)$$

$\square$

## D.10 Gaussian Adversarial Error

In this section, we prove that the sample complexity for Gaussian adversarial perturbation improves to $\Omega(k \log(p))$ as compared to the sub-Gaussian case where it is $\Omega(k^2 \log(p))$ as presented in Theorem 10. Since $\mathbf{x}^\star \sim \mathcal{N}(\mathbf{0}, \boldsymbol{\Sigma})$ and $\boldsymbol{\Delta}(\mathbf{x}^\star) \sim \mathcal{N}(\mathbf{0}, \boldsymbol{\Sigma}^{\boldsymbol{\Delta}})$, we can claim that $\mathbf{x} \sim \mathcal{N}(\mathbf{0}, \boldsymbol{\Sigma}^a)$, where

$$\boldsymbol{\Sigma}^a = \boldsymbol{\Sigma} + \boldsymbol{\Sigma}^{\boldsymbol{\Delta}}$$

The first step is to verify the strict dual feasibility condition by bounding the infinity norm of $\hat{\mathbf{z}}_{\mathcal{S}^c}$ defined in Eq. (5). In the case of the Gaussian distribution, we can express $\mathbf{X}_{\mathcal{S}^c}$ in Eq. (5) in terms of $\mathbf{X}_{\mathcal{S}}$ using the conditional expectation of jointly normal distribution:

$$\mathbf{X}_{\mathcal{S}^c}^\mathsf{T} = \boldsymbol{\Sigma}_{\mathcal{S}^c\mathcal{S}}^a \left(\boldsymbol{\Sigma}_{\mathcal{S}\mathcal{S}}^a\right)^{-1} \mathbf{X}_{\mathcal{S}}^\mathsf{T} + \boldsymbol{\Delta}\left(\mathbf{X}_{\mathcal{S}^c}^{\star\mathsf{T}}\right) \tag{69}$$

where $\boldsymbol{\Delta}\left(\mathbf{X}_{\mathcal{S}^c}^{\star\mathsf{T}}(i, j)\right) \sim \mathcal{N}(0, [\boldsymbol{\Sigma}_{\mathcal{S}^c|\mathcal{S}}^a]_{jj})$ and

$$\boldsymbol{\Sigma}_{\mathcal{S}^c|\mathcal{S}}^a = \boldsymbol{\Sigma}_{\mathcal{S}^c\mathcal{S}^c}^a - \boldsymbol{\Sigma}_{\mathcal{S}^c\mathcal{S}}^a \left(\boldsymbol{\Sigma}_{\mathcal{S}\mathcal{S}}^a\right)^{-1} \boldsymbol{\Sigma}_{\mathcal{S}\mathcal{S}^c}^a \tag{70}$$

This simplifies the expression of $\hat{\mathbf{z}}_{\mathcal{S}^c}$ to:

$$\hat{\mathbf{z}}_{\mathcal{S}^c} = \boldsymbol{\Sigma}_{\mathcal{S}^c\mathcal{S}}^a \left(\boldsymbol{\Sigma}_{\mathcal{S}\mathcal{S}}^a\right)^{-1} \hat{\mathbf{z}}_{\mathcal{S}} + \boldsymbol{\Delta}\left(\mathbf{X}_{\mathcal{S}^c}^{\star\mathsf{T}}\right) \left\{ \mathbf{X}_{\mathcal{S}} \left(\mathbf{X}_{\mathcal{S}}^\mathsf{T}\mathbf{X}_{\mathcal{S}}\right)^{-1} \hat{\mathbf{z}}_{\mathcal{S}} + \mathbf{P}\frac{(\mathbf{e} - \boldsymbol{\Delta}\left(\mathbf{X}_{\mathcal{S}}^\star\right)\mathbf{w}_{\mathcal{S}}^\star)}{\lambda n} \right\} \tag{71}$$

The first term can be bounded using mutual incoherence assumption. The second term is similar to Eq. 37(a) in (Wainwright, 2009) and can be bounded with $\mathcal{O}(k \log(p))$ samples using the same approach Gaussian tail bounds and $\chi^2$ tail bounds (Appendix J in (Wainwright, 2009)). This will ensure strict dual feasibility. Similarly, the uniqueness of the solution can be claimed with $\mathcal{O}(k \log(p))$ samples by using Lemma 9 from (Wainwright, 2009).

For bounding $\|\hat{\mathbf{w}}_{\mathcal{S}} - \mathbf{w}_{\mathcal{S}}^\star\|_\infty$, we need to bound $\left\|\left\|\mathbf{A}^{-1}\right\|\right\|_\infty$ which requires $\mathcal{O}\left(k^2 \log(p)\right)$ samples according to Lemma 13 for the sub-Gaussian case, where $\mathbf{A}$ is defined in Eq. (27). The sample complexity can be shown to be of order $\mathcal{O}(k \log(p))$ for the Gaussian case by using Lemma 5 of (Wainwright, 2009). Bounds for $\|\mathbf{w_1}\|_\infty$ and $\|\mathbf{w_2}\|_\infty$ can be guaranteed with high probability by choosing an appropriate value of $\lambda$. Hence the sample complexity is $\mathcal{O}(k \log(p))$.

# E  Experiments

## E.1  Synthetic data

Continuing the discussion in Section 4 of the main manuscript, we present the experimental settings in more detail here.

First we discuss the settings used for generating Figure 1 shown in the main manuscript. We start with the data generation process:

1. We randomly generate the support $\mathcal{S}$ of size $k = 20$, and hence $\mathcal{S}^c = [p] \setminus \mathcal{S}$.

2. We generate a random regression parameter vector, $\mathbf{w}_{\mathcal{S}}^{\star}$. We generate a random regression parameter vector by choosing $\mathbf{w}_i$ uniformly over $[-1, -0.1] \cup [0.1, 1]$ for $i \in \mathcal{S}$ and $\mathbf{w}_j = 0, \forall j \in \mathcal{S}^c$.

3. We generate the noise-free features, denoted by $\mathbf{x}^{\star} \in \mathbb{R}^p$. For the ease of analysis, we chose $\mathbf{x}_i^{\star} = \mathcal{N}(0,1), \forall i \in [p]$ and generate $n$ independent samples. The next step is to generate $y^{\star(j)}$ by using $y^{\star(j)} = \mathbf{w}^{\star\mathsf{T}} \mathbf{x}^{\star(j)}$ for $j \in [n]$.

4. We corrupt the measurements using Eq. (1), where $e \sim \mathcal{N}(0, \sigma_1^2)$ and $\boldsymbol{\Delta}(\mathbf{x}^{\star}) \sim \mathcal{N}(\mathbf{0}, \sigma_2^2 \mathbf{I})$. We chose the values of $\sigma_1 = 0.05$ and $\sigma_2 = 0.1$.

5. Further we estimate the parameter vector, denoted by $\hat{\mathbf{w}}$ using Lasso and check if $\mathcal{S}(\hat{\mathbf{w}}) = \mathcal{S}(\mathbf{w}^{\star})$ by setting $\lambda$ twice of the lower bound derived in Eq. (15).

6. We repeat the above five steps 200 times and count the number of success for $\mathcal{S}(\hat{\mathbf{w}}) = \mathcal{S}(\mathbf{w}^{\star})$ in step 5, which helps to compute the probability of success.

7. We repeat the above six steps for different values of $n$ for a given value of $p$. We consider a rescaled sample size $\frac{n}{\log(p)}$.

8. We repeat all the seven steps for different values of $p \in \{128, 256, 512\}$.

Figure 1 shows that the probability of support recovery increases as we increase the number of samples. Note that the probability reaches 1 when the rescaled sample size $\frac{n}{\log(p)} = 1150$. More importantly, the plot for each value of $p$ overlaps which confirms the hypothesis of sample complexity being logarithmic in the dimension of the regression vector.

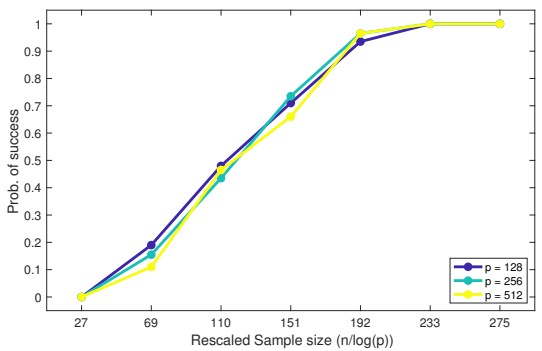
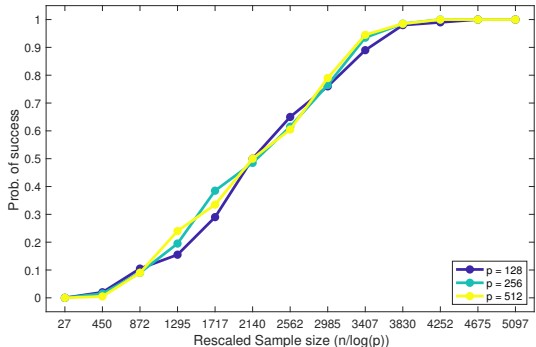

Figure 2: Probability of support recovery vs rescaled sample size for $\boldsymbol{\Delta}(\mathbf{x}^{\star}) = \mathbf{0}$

Figure 3: Probability of support recovery vs rescaled sample size for $\sigma_2 = 0.2$

We compare our results to the classical support recovery problem with no adversarial attack (Wainwright, 2009) by making $\mathbf{\Delta}(\mathbf{x}^\star) = \mathbf{0}$ via $\sigma_2 = 0$ in our experiments. We repeat the above experiment for the same value $k$ and $p$ with different values of sample size. The results are presented in Figure 2, which shows a similar trajectory as in Figure 1. The key difference is that we reach the probability of 1 in Figure 2 when the rescaled sample size $\frac{n}{\log(p)} = 275$ which was 1150 for Figure 1. Comparison of Figure 1 and Figure 2 helps us to understand the effect of an adversary.

In order to understand the effect of $\mathbf{\Delta}(\mathbf{x}^\star)$, we increase $\sigma_2 = 0.2$ in the step 4 of the procedure mentioned above. Note that we have doubled $\sigma_2$ as compared to our default analysis. From Eq. (15), we can observe that for a constant lower bound on $\lambda$, the sample size $(n)$ has to increase linearly with $\left\|\mathbf{\Sigma}_{\mathcal{SS}}^{\mathbf{\Delta}}\right\|_2$. We are discussing the case of constant lower bound because $\min_{i \in \mathcal{S}} |\mathbf{w}_i^\star| \geq 2f(\lambda)$ as per the fourth claim of Theorem 10. As we double $\sigma_2$, $\left\|\mathbf{\Sigma}_{\mathcal{SS}}^{\mathbf{\Delta}}\right\|_2$ will increase 4 times and hence the minimum the number of samples required should also increase 4 times.

We actually observe this phenomenon in our experiments. The results for $\sigma_2 = 0.2$ are presented in Figure 3. Note that the probability of success reaches 1 when the rescaled sample size $\frac{n}{\log(p)} = 4600 \approx 4 \times 1150$ in Figure 3 which is four times the rescaled sample size needed (1150) for success probability one for the case of $\sigma_2 = 0.1$ presented in Figure 1. Hence our theoretical claim is justified empirically.

We further conduct experiments with more complicated forms of adversarial perturbation ($\mathbf{\Delta}(\mathbf{x}^\star)$). These cases are discussed below.

### E.2 Mixture of two distributions

The adversarial perturbation for $j^{\text{th}}$ sample is chosen as a combination of Bernoulli distribution and Gaussian distribution as shown below:

$$\mathbf{\Delta}(\mathbf{x}^{\star(j)}) = \frac{r\mathbf{v}^{(j)}}{\left\|\mathbf{v}^{(j)}\right\|_2}, \quad \text{where} \quad \mathbf{v}^{(j)} \sim \begin{cases} \mathbf{v}_i^{(j)} \sim 2\text{Bernoulli}(0.5) - 1 & \text{with probability 0.5 for } i \in [p] \\ \mathcal{N}(\mathbf{0}, \mathbf{I}) & \text{with probability 0.5} \end{cases} \quad (72)$$

where $r$ denotes the per sample budget for adversarial perturbation and $j \in [n]$. Compared to the previous case of all adversarial samples being drawn from Gaussian distribution, now 50% of the samples will be drawn from scaled Bernoulli distribution such that each entry is $+1$ or $-1$ with equal probability. As Bernoulli distribution is bounded, we can claim it is sub-Gaussian, and the final distribution of $\mathbf{\Delta}(\mathbf{x}^\star)$ is sub-Gaussian. Note that $\mathbf{\Delta}(\mathbf{x}^\star)$ is designed in such a way that $\left\|\mathbf{\Delta}(\mathbf{x}^{\star(j)})\right\|_2 = \epsilon$ for all $j \in [n]$ to respect the budget constraint. We chose $\epsilon = 0.1$ in our simulations.

After generating the adversarial perturbation, we repeat the same exercise as described previously and present the plot for the probability of support recovery in Figure 4. The plot confirms that Lasso performs successful support recovery and also confirms that the sample complexity is logarithmic with respect to the size of the regression parameter vector. Further, we move to another method for adversarial perturbation generation.

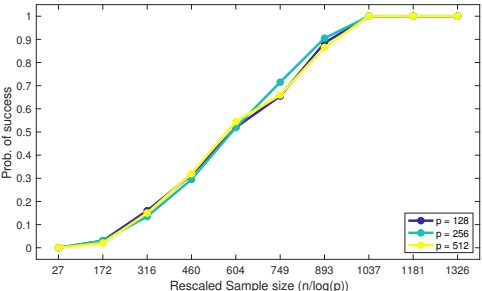

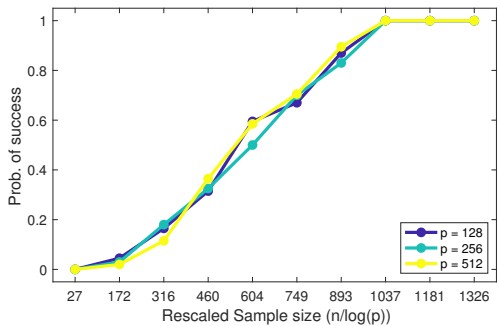

Figure 4: Probability of support recovery vs rescaled sample size when adversarial perturbations are drawn from a mixture of Bernoulli distribution and Gaussian distribution

Figure 5: Probability of support recovery vs rescaled sample size when adversarial perturbations are correlated with uncorrupted regressors

### E.3    Adversary uses uncorrupted data

In this approach, we design the adversarial perturbation in such a way that it is correlated with uncorrupted regressors ($\mathbf{x}^\star$) in 50% of the samples as shown below:

$$\boldsymbol{\Delta}(\mathbf{x}^{\star(j)}) = \frac{r\mathbf{v}^{(j)}}{\left\|\mathbf{v}^{(j)}\right\|_2}, \quad \text{where} \quad \mathbf{v}^{(j)} \sim \begin{cases} (2\text{Bernoulli}(0.5) - 1)\,\mathbf{x}^\star & \text{with probability } 0.5 \\ \mathcal{N}(\mathbf{0}, \mathbf{I}) & \text{with probability } 0.5 \end{cases} \tag{73}$$

where $j \in [n]$. The above equation indicates the adversarial perturbation may be positively or negatively correlated with uncorrupted regressors with the probability of 0.5. We further repeat the experiment as discussed at the beginning of Section 4 and present the support recovery plot in Figure 5. The plot verifies that the algorithm can successfully recover the support even when the adversarial perturbation is correlated with the uncorrupted features.

### E.4    Different cases of regularization parameter

In Theorem 10, we saw the regularization parameter is given by:

$$\lambda \geq \max\left\{\lambda_1, \lambda_2, \lambda_3\right\}$$
$$= \max\left\{\frac{16b}{\gamma}, \frac{q_1\sigma_e}{\gamma}\sqrt{\frac{2\log(p)}{n}}, \frac{16q}{\gamma}\sqrt{\frac{4\log(p)}{n}}\right\} \tag{74}$$

In this section, we analyze the support recovery by simulating the data for three cases such that regularization parameter:

1. $b \neq 0$ and $\lambda = \lambda_1 = \frac{16b}{\gamma}$

2. $b = 0$ and $\lambda = \lambda_2 = \frac{q_1\sigma_e}{\gamma}\sqrt{\frac{2\log(p)}{n}}$

3. $b = 0$ and $\lambda = \lambda_3 = \frac{16q}{\gamma}\sqrt{\frac{4\log(p)}{n}}$

For all the three cases, we present the plots of three probabilities computed empirically:

1. $\mathbb{P}\left[\mathcal{S}\left(\hat{\mathbf{w}}\right) = \mathcal{S}\right]$, which denotes exact support recovery.

2. $\mathbb{P}\left[\forall i \in \mathcal{S} : \hat{\mathbf{w}}_i \neq 0\right]$, which denotes correct recovery of support.

3. $\mathbb{P}\left[\forall i \notin \mathcal{S} : \hat{\mathbf{w}}_i = 0\right]$, which denotes correct recovery of non-support.

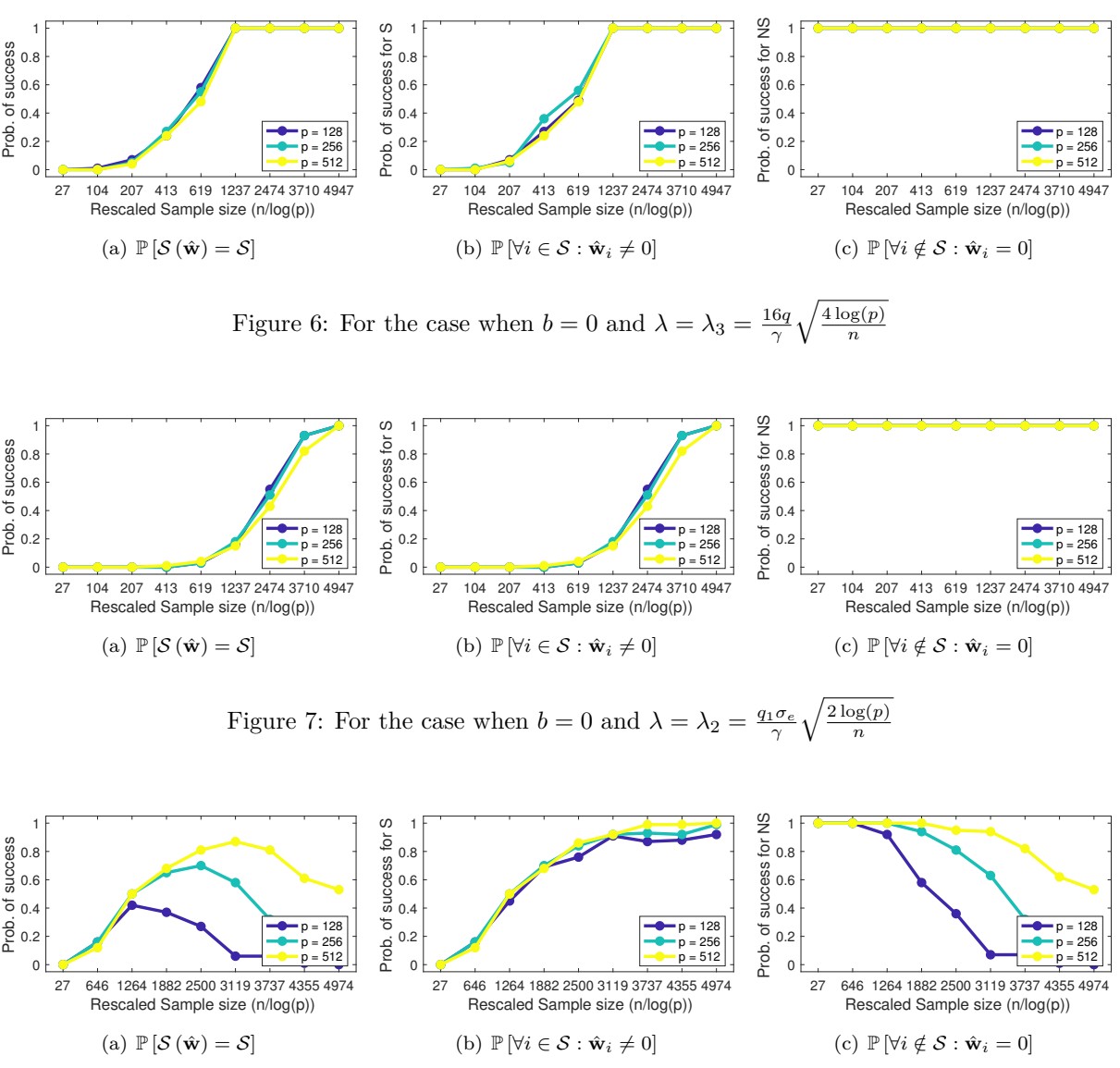

Figure 6: For the case when $b = 0$ and $\lambda = \lambda_3 = \frac{16q}{\gamma}\sqrt{\frac{4\log(p)}{n}}$

Figure 7: For the case when $b = 0$ and $\lambda = \lambda_2 = \frac{q_1\sigma_e}{\gamma}\sqrt{\frac{2\log(p)}{n}}$

Figure 8: For the case when $b \neq 0$ and $\lambda = \lambda_1 = \frac{16b}{\gamma}$

Note that $\mathcal{S}$ denotes the true support. We start from case 3, when $b = 0$ and $\lambda = \lambda_3$. The support recovery plots presented in Figure 6 shows that the support is recovered successfully with high probability when the rescaled sample size (presented in x-axis) is around 1200. Further, we move onto case 2, when $b = 0$ and $\lambda = \lambda_2$. We increase $\sigma_e$ by 4 times, so that $\lambda_2 > \lambda_3$. The plots presented in Figure 7 shows that we can still do successful support recovery when the rescaled sample size is around 4800. This indicates that the learner now requires more number of samples for support recovery.

For the above two cases, we chose $\boldsymbol{\Sigma}^{\mathbf{x}^\star \boldsymbol{\Delta}}_{[p]\mathcal{S}} = \mathbf{0}_{p \times k}$ so that $b = \left\| \boldsymbol{\Sigma}^{\mathbf{x}^\star \boldsymbol{\Delta}}_{[p]\mathcal{S}} \mathbf{w}^\star_{\mathcal{S}} \right\|_\infty = 0$, and we can choose $\lambda = \lambda_2$ or $\lambda_3$. To study case 1, when $b \neq 0$ and $\lambda = \lambda_1$, we generate the data with $\boldsymbol{\Sigma}^{\mathbf{x}^\star \boldsymbol{\Delta}}_{[p]\mathcal{S}} \neq \mathbf{0}_{p \times k}$. The plots presented in Figure 8 shows that 100% exact support recovery is not possible even for a large sample size. Figure 8 (b) shows that we can recover the support successfully but not the non-support as shown in Figure 8 (c). This presents a case in which the adversary can dominate over Lasso.

But, it should be noted that it might be possible to obtain the correct support even in the case of $\mathbf{\Sigma}_{[p]\mathcal{S}}^{\mathbf{x}^\star \mathbf{\Delta}} \neq \mathbf{0}_{p \times k}$ as shown in the Section E.3 or Figure 5. This can happen when the entries in $\mathbf{w}^\star$ are large enough or $b$ is small such that Eq. (17) holds as shown in our theoretical analysis. Hence, the practical use of Lasso is still viable.

### E.5 Real-world data

We used the BlogFeedback dataset (Buza, 2014) which contains 52397 samples and 276 features extracted from blog posts and the task is to predict how many comments a post will receive using these features.

First, the "true" support is obtained by solving Lasso on the original Blogfeedback dataset (Buza, 2014). Let the "perturbed" support be defined as the support obtained by solving Lasso on the perturbed Blog-Feedback dataset. To construct the perturbed dataset, we add zero-mean Gaussian white noise in each feature. The variance of Gaussian noise is chosen in proportion to the feature variance of the original data. After obtaining the "true" and "perturbed" support, we compute the standard F1-score defined below:

$$\text{Recall} = \frac{\text{Number of elements in the "true" support that are in the "perturbed" support}}{\text{Number of elements in the "perturbed" support}} \tag{75}$$

$$\text{Precision} = \frac{\text{Number of elements in the "true" support that are in the "perturbed" support}}{\text{Number of elements in the "true" support}} \tag{76}$$

$$\text{F1-score} = 2\frac{\text{Recall} \times \text{Precision}}{\text{Recall} + \text{Precision}} \tag{77}$$

The F1-score of the recovered support from the perturbed data is 0.9462, which effectively implies that Lasso is able to recover most of the support in real-world data as well. Further, we test the algorithm against other approaches for generating adversarial perturbations.

We modify the approach of a mixture of two distributions in Eq. (72) by scaling with standard deviations in regressors to handle large variations in regressors as shown below:

$$\mathbf{\Delta}(\mathbf{x}^{\star(j)}) = \frac{r\mathbf{v}^{(j)}}{\left\|\mathbf{v}^{(j)}\right\|_2}, \quad \text{where} \quad \mathbf{v}^{(j)} \sim \begin{cases} \mathbf{v}_i^{(j)} \sim (2\text{Bernoulli}(0.5) - 1)\text{std}(\mathbf{x}_i^\star) & \text{with probability 0.5 for } i \in [p] \\ \mathcal{N}(\mathbf{0}, \mathbf{\Sigma}) & \text{with probability 0.5} \end{cases} \tag{78}$$

where $j \in [n]$. The F1-score is reported to be 0.9393 for $r = 1000$, proving that the algorithm can recover the support.

We further test the algorithm against the correlated adversarial perturbation by modifying Eq. (73) to handle large variations in regressors as shown below:

$$\mathbf{\Delta}(\mathbf{x}^{\star(j)}) = \frac{r\mathbf{v}^{(j)}}{\left\|\mathbf{v}^{(j)}\right\|_2}, \quad \text{where} \quad \mathbf{v}^{(j)} \sim \begin{cases} (2\text{Bernoulli}(0.5) - 1)\,\mathbf{x}^\star & \text{with probability 0.5} \\ \mathcal{N}(\mathbf{0}, \mathbf{\Sigma}) & \text{with probability 0.5} \end{cases} \tag{79}$$

where $j \in [n]$. We repeat the experiment in the same procedure and report the F1-score to be 0.9485, which confirms that Lasso performs successful support recovery even when the adversarial perturbation is correlated with the uncorrupted regressors. Note that F1-score is reported to be 1 in all the cases if we do not use the standard deviation scaling to normalize the adversarial perturbation. Hence, by modifying the procedure of adversarial perturbation introduction, we are solving a more challenging problem.

Note that we do not need to verify the assumptions mentioned in Section 2.3 to run the algorithm. They are only needed for theoretical analysis to derive the sample complexity for support recovery.