# OpenReview forum: "A Theoretical Study of The Effects of Adversarial Attacks on Sparse Regression"
_TMLR — Accepted by TMLR_

### Review · Reviewer_7qn5 · 2024-09-03

**Summary Of Contributions:**

This paper addresses theoretical aspects of Lasso regression in the context of an adversarial attack. In particular, it focuses on the support recovery problem, i.e. recovering and estimating all non-zero regression coefficients. The type of attack is chosen to be more general and more challenging than previous results in the literature, as it disturbs both features of the support and outside.
The paper claims multiple theoretical properties to allow for the support to be recovered and these coefficient to be suitably estimated with Lasso, including necessary conditions on the adversarial attacks and a lower bound on the regularlization term. The results are also corroborated with numerical experiments on both a synthetic and real-world dataset.

**Audience:**

Yes

**Claims And Evidence:**

Yes

**Requested Changes:**

See weaknesses section.

**Strengths And Weaknesses:**

Strengths:
- Theoretical results of the paper seem solid, and well corroborated in experiment
- Paper is comprehensive and each result is justified
- Interesting insights are given when interpreting the theoretical findings

Weaknesses:
- I believe the paper is not citing several works done on Robust Linear Regression addressing this exact type adversarial noise, see for instance Robust and Adaptive Optimization by D. Bertsimas and D. den Hertog, 2022. It would be good to integrate this part of the literature and see how they complement the findings
- Typos: several typos in the paper (too many to keep track of), making reading hard to follow at times
- Clarity and rigor: reasonings in proofs are not always easy to follow and not completely rigorous, for instance:
  - Appendix A section 7: proof unclear, please explicitly explain how the result from the paper cited allows to prove the lemma
  - Appendix B: assumption that there exists such a case is not proven
  - Organization of the paper: lemmas introduced are only proven much later

---

> ### Author Response · Authors · 2024-11-03
>
> > ...The type of attack is chosen to be more general and more challenging than previous results in the literature, as it disturbs both features of the support and outside...
>
> > - Theoretical results of the paper seem solid, and well corroborated in experiment
>
> > - Paper is comprehensive and each result is justified
>
> > - Interesting insights are given when interpreting the theoretical findings
>
> We thank the reviewer for the appreciation of our work.
>
> > - ... not citing several works done on Robust Linear Regression addressing this exact type adversarial noise, see for instance Robust and Adaptive Optimization by D. Bertsimas and D. den Hertog, 2022
>
> Many thanks for bringing this up. We have added the following text in the introduction: "... In addition, there is prior literature in robust optimization (Bertsimas & Den Hertog, 2020), where there is uncertainty in the input coefficients of an optimization problem. Their goal is to provide guarantees that the unperturbed optimal solution is still feasible (although not optimal) in the problem with perturbed inputs. Bertsimas & Den Hertog (2020) treats canonical problems such as linear programming, for instance. While we also focus on an optimization problem, Bertsimas & Den Hertog (2020) provides a general treatment and lacks the focus on machine learning aspects, e.g., support recovery and adversarial perturbations."
>
> > - Typos: several typos in the paper
>
> We have made several corrections (typos and grammar) and changed some \cite to \citep.
>
> > - Appendix A section 7: proof unclear
>
> Thanks for pointing this out. We have clarified our proof in the revision.
>
> > - Appendix B: assumption that there exists such a case is not proven
>
> Many thanks for bringing this up. We have improved our proof in the revision.
>
> > Organization of the paper: lemmas introduced are only proven much later
>
> We have included explanations/proof-sketches after Lemmas 2, 3, 6 and 9. We preferred this style in order to not confuse the reader with technical details, and to keep the focus on the higher level messages. While we are not the only ones who use this writing style, we are open to any specific change that the reviewer might consider to improve clarity.

---

### Review · Reviewer_yjy2 · 2024-10-12

**Summary Of Contributions:**

This paper provides a theoretical study on support recovery in Lasso regression under adversarial attacks, focusing on how adversarial perturbations affect the recovery of non-zero coefficients in a sparse regression model. The authors critique existing deterministic adversarial attack models like FGSM, which primarily maximize the loss function and can be countered with few samples, and propose a more challenging adversarial model involving zero-mean sub-Gaussian perturbations that depend on uncorrupted data. The paper derives a necessary condition for support recovery that applies to any algorithm, not just Lasso, identifying key parameters like mutual incoherence and the covariance between adversarial perturbations and uncorrupted data as critical factors. The authors show that sample complexity for support recovery under adversarial attacks is O(k^2log(p)), where k is the size of the support and p is the number of features, and prove that when adversarial perturbations align with the support (i.e., b!=0), perfect support recovery becomes impossible. Empirical validation confirms these findings, demonstrating that when b=0, support recovery can be achieved with a sample complexity that scales logarithmically with p, but with b!=0, recovery is far more challenging. The paper contributes to the understanding of adversarial robustness in sparse models and, while primarily focused on Lasso regression, provides insights that could inspire further research in adversarial machine learning and robustness in more complex models like neural networks.

**Audience:**

Yes

**Broader Impact Concerns:**

There is no impact concern regarding the ethical implications of the paper.

**Claims And Evidence:**

Yes

**Requested Changes:**

Practical Relevance and Generalization to Neural Networks: Given that adversarial attacks are often discussed in the context of neural networks, providing more analysis or discussion on how the proposed adversarial model and results can be generalized to these settings would significantly improve the paper’s broader relevance.

Clarify Assumptions on Perturbations: The assumptions regarding sub-Gaussian and zero-mean perturbations should be justified further, or alternative scenarios should be discussed to cover more realistic adversarial conditions.

Provide More Empirical Details: Including more details about how adversarial perturbations are generated and whether the method can be generalized to different datasets would strengthen the empirical validation and improve reproducibility.

**Strengths And Weaknesses:**

Strengths:
Novel Adversarial Model: The submission introduces a more sophisticated adversarial model that is conditionally dependent on uncorrupted data. This novelty is important because it demonstrates how adversarial perturbations that correlate with the true data structure can significantly increase the difficulty of support recovery. The introduction of zero-mean sub-Gaussian perturbations adds complexity and realism to the attack model compared to simpler methods like FGSM, which only focus on maximizing the loss function.

Rigorous Theoretical Contributions: The paper provides a detailed theoretical analysis of support recovery under adversarial attacks, deriving necessary conditions and sample complexity bounds. The result that the new adversarial model demands O(k^2log(p)) samples for support recovery, where k is the support size and p is the feature dimension, adds valuable insights into the robustness of sparse models under attack. This shows a clear improvement over simpler attacks that do not exploit the correlation with uncorrupted data.

Challenging the Effectiveness of FGSM: By showing how attacks that consider correlations with uncorrupted data outperform simple gradient-based methods like FGSM, the paper provides a strong critique of existing methods and suggests avenues for improving the robustness of models. This insight could influence future research in adversarial machine learning.

Empirical Validation: The authors back their theoretical findings with experiments on both synthetic and real-world data, which is crucial for validating the theoretical results. The demonstration that perfect support recovery is not achievable when the adversarial perturbations correlate with uncorrupted data (i.e., when b!=0) strengthens the claim that these new attacks are more effective.

Weakness:
Support Recovery as a Metric: While relevant for sparse models, support recovery may not fully capture adversarial effectiveness, especially in neural networks where the concept of strict support doesn’t apply. The paper could discuss how its findings generalize to broader models.

Limited Practical Relevance: The study focuses on Lasso regression, which may limit its applicability to models like neural networks where all weights contribute. A discussion on generalizing to other model types would strengthen the paper.

Assumptions on Perturbations: The reliance on sub-Gaussian, zero-mean perturbations may not always align with real-world adversarial attacks, which might behave differently.

Empirical Clarity: More detailed explanations of the empirical setup and generalizability of the attack method could improve the paper’s practical applicability and reproducibility.

---

> ### Author Response · Authors · 2024-11-01
> **Response 1/2**
>
> > ... contributes to the understanding of adversarial robustness in sparse models ... provides insights that could inspire further research ... in more complex models like neural networks.
>
> > Novel Adversarial Model ... important because it demonstrates how adversarial perturbations that correlate with the true data structure can significantly increase the difficulty ... zero-mean sub-Gaussian perturbations adds complexity and realism ...
>
> > Rigorous Theoretical Contributions ... adds valuable insights into the robustness of sparse models under attack.
>
> > ... attacks that consider correlations with uncorrupted data outperform simple gradient-based methods like FGSM ... provides a strong critique of existing methods ... This insight could influence future research in adversarial machine learning.
>
> > Empirical Validation ... crucial for validating the theoretical results ... perfect support recovery is not achievable when the adversarial perturbations correlate with uncorrupted data ... strengthens the claim that these new attacks are more effective.
>
> We thank the reviewer for the appreciation of our work.
>
> > ... While relevant for sparse models, support recovery may not fully capture adversarial effectiveness, especially in neural networks where the concept of strict support doesn't apply. The paper could discuss how its findings generalize to broader models.
>
> > ... The study focuses on Lasso regression, which may limit its applicability to models like neural networks ... A discussion on generalizing to other model types would strengthen the paper.
>
> > ... adversarial attacks are often discussed in the context of neural networks ... discussion on how the proposed adversarial model and results can be generalized to these settings would significantly improve the paper's broader relevance.
>
> Many thanks for bringing this up. We have added a Concluding Remarks section with the following three paragraphs:
>
> "Sparsity, support recovery and weight recovery are tightly related concepts. In our paper, support relates to the zero/nonzero weights in the sparse regression vector (nonzero entries corresponding to relevant features for prediction) and weight recovery relates to estimating a vector that is close (in $\ell_\infty$ distance) to the true regression vector (See e.g., Theorem 10, Eq. (3))."
>
> "Our initial analysis for sparse regression already highlights some fundamental issues that not only pertain to sample complexity, e.g., impossibility. Our initial results could be later extended to other machine learning problems where sparsity as well as support and weight recovery is relevant. This includes for instance, Gaussian graphical models (Ravikumar et al., 2011), Ising models (Ravikumar et al., 2010), nonparametric regression (Ravikumar et al., 2009) and diffusion networks (Daneshmand et al., 2014)."
>
> "Besides the aforementioned models, we believe our results could potentially motivate future work on neural networks. Indeed, there has been a recent interest on sparse neural networks (Liu & Wang, 2023) as well as weight recovery for two-layer neural networks (Bakshi et al., 2019)."
>
> Besides the above, we would like to point out that there was an ICLR 2023 Workshop on
> Sparsity in Neural Networks.
>
> > ... The reliance on sub-Gaussian, zero-mean perturbations may not always align with real-world adversarial attacks, which might behave differently.
>
> > ... The assumptions regarding sub-Gaussian and zero-mean perturbations should be justified further ...
>
> In the already existing Remark 4, we have clarified that "The class of sub-Gaussian variates includes for instance some unbounded random variables (e.g., Gaussian variables), any bounded random variable (e.g. Bernoulli, multinomial, uniform), any random variable with strictly log-concave density, and any finite mixture of sub-Gaussian variables."
>
> To provide some important real-world example, note that when using images for instance, the pixel values are bounded (e.g., from 0 to 255). Thus perturbations must be bounded and therefore perturbations are sub-Gaussian.
>
> The reason behind the assumption of zero mean is described in Lemma 5 (followed by an implication in Lemma 6). In short, if the adversary attacks some feature with non-zero mean, the learner can easily detect such an attack.

---

> ### Author Response · Authors · 2024-11-01
> **Response 2/2**
>
> > ... More detailed explanations of the empirical setup and generalizability of the attack method could improve the paper's practical applicability and reproducibility.
>
> > ... more details about how adversarial perturbations are generated and whether the method can be generalized to different datasets would strengthen the empirical validation and improve reproducibility.
>
> Regarding empirical setup and reproducibility, we have added a note "See Appendix E.1 for details regarding the data and adversarial perturbation generation."
>
> Regarding generalizabity and practical applicability, please note that:
>
> - Lemma 2 shows that existing adversarial attacks (Goodfellow et al., 2015; Madry et al., 2018; Szegedy et al., 2014; Xing et al., 2021; Yin et al., 2019; Awasthi et al., 2020; Qin et al., 2021) are a special case of our adversarial model.
>
> - Lemma 3 shows that the Huber-model (Prasad et al., 2020; Diakonikolas et al., 2019) corrupting only a fraction of samples is a special case of our adversarial model.

---

### Review · Reviewer_xbhc · 2024-10-18

**Summary Of Contributions:**

### Summary

In this paper:
* the authors introduce a new theoretical framework for studying adversarial attacks on sparse recovery algorithms (such as the Lasso).
* under this framework, they prove that adversarial learning requires more data: under adversarial perturbations of the data, the sample complexity of support estimation of a $k$-sparse length-$p$ vector using the Lasso is $O(k^2 \log p)$, as opposed to the standard $O(k \log p)$ bound for regular recovery,
* they show that this quadratic dependence on $k$ can be reduced to linear dependence under certain additional assumptions.
* they support their theoretical results with some basic simulations.

**Audience:**

Yes

**Claims And Evidence:**

No

**Requested Changes:**

* Better discussion of the relevant literature.
* Better intuition of the theoretical analysis (especially the proofs, and the source of the various parameter scalings) in the main paper.

**Strengths And Weaknesses:**

### Strengths
* Studying concrete toy cases is a useful way to gain theoretical insights into adversarial learning. Such insights can subsequently inform practice.
* The results and techniques are nice (provided they are correct --- see comments below on this point).

### Weaknesses

* The paper treads a lot of familiar ground that was explored in earlier results. While novelty is not a criterion for TMLR, the paper probably needs to engage with the literature better before it can be published. In particular, it misses discussion/connections with at least two major lines of work that I am aware of.

    The first line of work is from the robust sparse recovery literature. (I am aware of the distinction between generic robustness and adversarial perturbations, but in the authors' model of stochastic perturbations the two settings seem tightly connected.) I would start with discussing Herman/Strohmer's 2010 paper, "General Deviants: Analysis of Perturbations in Compressed Sensing", and also revisit any relevant follow-up papers.

    The second line of work is from the (theoretical) adversarial learning literature. I would start with discussing Schmidt et al (2018), "Adversarial learning requires more data", which shows an analogous gap between standard and adversarial sample complexity in the classification setting, and revisit any relevant follow-up papers.

* The paper is not mathematically precise. I would encourage the authors to do a thorough review of the manuscript with a critical eye in order to ensure mathematical correctness.

* The paper can benefit from intuition surrounding the main results, proof techniques, and implications. I see that there is a good amount of discussion in the appendix, but found the main paper somewhat lacking in this aspect. Eg. Lemmas 2,3,6,9 are presented without a proof sketch. If space is an issue then the last parts of Section 3 can be moved to the appendix. Also, a bit more discussion about the discussion between the paper's techniques and classical sparse recovery results (e.g. the support recovery work of Wainwright '09) may help.

### Other questions

I am a bit confused by a few points but these are likely due to my own lack of understanding:
* Where does the noise variance $\sigma_e$ appear in any of the sample complexity bounds? (alternately, is there a assumption that this is bounded?)
* Where does the $k^2$ dependence come from? As I understand, it comes about due to Lemma 11 and Theorem 14, but more intuition would be helpful.

---

> ### Author Response · Authors · 2024-11-03
> **Response 1/2**
>
> > - Studying concrete toy cases is a useful way to gain theoretical insights into adversarial learning. Such insights can subsequently inform practice.
>
> > - The results and techniques are nice...
>
> We thank the reviewer for the appreciation of our work.
>
> > ... discussing Herman/Strohmer's 2010 paper, "General Deviants: Analysis of Perturbations in Compressed Sensing"
>
> Many thanks for bringing this up. We have added the following text in the introduction: "Some initial attempts have been made in the sparse regression literature (Herman & Strohmer, 2010), but we believe that the particular theoretical framework did not allow prior work to make important findings such as noting that when adversarial perturbations correlate with uncorrupted data, then support recovery is impossible. ..."
>
> This impossibility has been appreciated by reviewer yjy2.
>
> > ... discussing Schmidt et al (2018), "Adversarial learning requires more data"
>
> There were already closely-related literature to what the reviewer suggested, see e.g., Yin et al. (2019).
>
> We have added this reference to the already existing paragraph: "The attack in existing methods is just one possible attack model that fits our assumptions; hence, our adversarial model is more general. For instance, approaches by y Goodfellow et al. (2014); Madry et al. (2017); Szegedy et al. (2013); Xing et al. (2021); Yin et al. (2019); Schmidt et al. (2018); Awasthi et al. (2020); Qin et al. (2021) consider deterministic attacks within an $\epsilon$-ball, which is sub-Gaussian since any bounded random variable is known to be sub-Gaussian."
>
> > - ... intuition surrounding the main results, proof techniques, and implications ... Eg. Lemmas 2,3,6,9 are presented without a proof sketch ... discussion between the paper's techniques and classical sparse recovery results (e.g. ... Wainwright '09)
>
> > - Better intuition of the theoretical analysis (... the source of the various parameter scalings)
>
> We have already discussed implications in "Our contributions" as well as provided section titles with clear messages of implications, for instance:
>
> - 2.1 "Existing attacks are special cases of our adversarial model"
>
> - 2.2 "Our adversarial attack is more challenging than existing attacks"
>
> - 2.4 "Necessary condition for support recovery"
>
> - 3.2 "Adversary vs Lasso Territory"
>
> Section 3.2 contains some discussion about implications. We have included explanations/proof-sketches after Lemmas 2, 3, 6 and 9. Besides this, we also identified five places in Section 3 where we have already discussed implications, and labeled them with "Intuition".
>
> We have added an additional contribution in the introduction "Theoretical tools: Our contribution can be seen as a first step towards the study of learning from adversarial training data. As a byproduct, we also obtain several technical results related to a new concentration inequality (Theorem 14), necessary condition (Lemma 9) which could be useful for other problems."
>
> Discussing all the different parameter scalings might remove the focus from the higher level messages. We decided to choose some of the parameters as the most important ones (e.g., b), but we are open to any specific change aimed to highlight any other parameter that the reviewer might consider interesting.
>
> > - The paper is not mathematically precise. I would encourage the authors to do a thorough review of the manuscript with a critical eye in order to ensure mathematical correctness.
>
> Right after the reviewer submitted the review, we stated "we kindly ask you to be more specific regarding the statement", but unfortunately the reviewer did not reply to us. Given this, we have focused on the comments from reviewer 7qn5.
>
> > - Where does the noise variance $\sigma_e$ appear in any of the sample complexity bounds? (alternately, is there a assumption that this is bounded?)
>
> $\sigma_e$ is used in Lemma 15, which relates to the expression seen in Theorem 10. $\sigma_e$ should be bounded, i.e., $\sigma_e$ should not tend to infinity.

---

> ### Author Response · Authors · 2024-11-03
> **Response 2/2**
>
> > - under adversarial perturbations of the data, the sample complexity ... is $O(k^2 \log p)$, as opposed to the standard $O(k \log p)$ bound for regular recovery
>
> > - ... this quadratic dependence on $k$ can be reduced to linear dependence under certain additional assumptions.
>
> > - Where does the $k^2$ dependence come from?
>
> We would like to highlight three facts to provide the proper context before answering the question:
>
> - Our analysis highlights some important difference beyond sample complexity, e.g., impossibility under some considerations as appreciated by reviewer yjy2.
>
> - Regarding sample complexity, we mentioned in "Our Contributions" that "if we assume the adversarial perturbation to be Gaussian, the sample complexity improves to $\Omega(k \log p)$ (Appendix D.10)."
>
> - In the introduction, we made what we believe is a more proper comparison "... our adversary demands $\Omega(k^2 \log p)$ samples, whereas existing attacks only need $\Omega(\log p)$ samples (in Section 2.2)"
>
> The reason why Gaussian distributions allow for better concentration inequalities (and thus $O(k \log p)$ sample complexity) is because Gaussian distributions are symmetric and have nice conditional distribution equations (Both of these properties are unavailable for general sub-Gaussian distributions). Consider for instance a Gaussian distribution with zero-mean and identity covariance. If you take a slice of the probability density function passing through the origin, you would visualize exactly the same Gaussian bell shape. This fact is exploited in Lemma 5 of Wainwright '09 for instance. (Please see Appendix D.10 for the full explanation.)
>
> Finally, we noticed that the reviewer chose "Claims And Evidence: No". Please let us know if there is anything else that we are missing.

---

### Decision · Action_Editor_Tbgz · 2024-11-28

**Recommendation:** Accept as is

**Comment:**

The reviewers acknowledge the theoretical insights provided by the manuscript as well as the empirical validation. They note that the results may be of limited practical relevance with a small problem scope.

**Audience:**

Yes.

**Claims And Evidence:**

This is mostly a theoretical paper. Claims are supported by proofs. Reviewers note that the theoretical results are solid and supported by experiments.